# A non-canonical vitamin K cycle is a potent ferroptosis suppressor

Eikan Mishima[1,2 ✉], Junya Ito[3], Zijun Wu[4], Toshitaka Nakamura[1], Adam Wahida[1], Sebastian Doll[1], Wulf Tonnus[5], Palina Nepachalovich[6,14], Elke Eggenhofer[7], Maceler Aldrovandi[1], Bernhard Henkelmann[1], Ken-ichi Yamada[8], Jonas Wanninger[1], Omkar Zilka[4], Emiko Sato[9], Regina Feederle[10], Daniela Hass[11], Adriano Maida[11], André Santos Dias Mourão[12], Andreas Linkermann[5], Edward K. Geissler[7], Kiyotaka Nakagawa[3], Takaaki Abe[2,13], Maria Fedorova[6,14], Bettina Proneth[1], Derek A. Pratt[4] & Marcus Conrad[1 ✉]

Ferroptosis, a non-apoptotic form of cell death marked by iron-dependent lipid peroxidation[1], has a key role in organ injury, degenerative disease and vulnerability of therapy-resistant cancers[2]. Although substantial progress has been made in understanding the molecular processes relevant to ferroptosis, additional cell-extrinsic and cell-intrinsic processes that determine cell sensitivity toward ferroptosis remain unknown. Here we show that the fully reduced forms of vitamin K—a group of naphthoquinones that includes menaquinone and phylloquinone[3]—confer a strong anti-ferroptotic function, in addition to the conventional function linked to blood clotting by acting as a cofactor for γ-glutamyl carboxylase. Ferroptosis suppressor protein 1 (FSP1), a NAD(P)H-ubiquinone reductase and the second mainstay of ferroptosis control after glutathione peroxidase-4[4,5], was found to efficiently reduce vitamin K to its hydroquinone, a potent radical-trapping antioxidant and inhibitor of (phospho)lipid peroxidation. The FSP1-mediated reduction of vitamin K was also responsible for the antidotal effect of vitamin K against warfarin poisoning. It follows that FSP1 is the enzyme mediating warfarin-resistant vitamin K reduction in the canonical vitamin K cycle[6]. The FSP1-dependent non-canonical vitamin K cycle can act to protect cells against detrimental lipid peroxidation and ferroptosis.

Unrestrained iron-dependent lipid peroxidation is the common downstream cellular event leading to rupture of cellular membranes and ferroptosis[7]. Cells have evolved a number of highly efficient redox systems that counteract uncontrolled lipid peroxidation, such as selenium-dependent glutathione peroxidase-4 (GPX4), the FSP1-ubiquinone pathway, and the biopterin-dihydrofolate reductase system[4,8–12]. In addition, cells and tissues harness vitamin E (comprising both tocopherols and tocotrienols), Nature's premier lipophilic radical-trapping antioxidants[13] (RTA), to protect them from overwhelming lipid peroxidation and ferroptosis[7]. Vitamin E has also been shown to rescue certain tissues, including liver, endothelium, CD8[+] T cells and hematopoietic stem cells, from the deleterious consequences induced by the tissue-specific disruption of the key ferroptosis regulator GPX4[14–16].

## Vitamin K is a potent anti-ferroptotic compound

To interrogate whether there are other systems besides the aforementioned intrinsic and extrinsic mechanisms that efficiently prevent ferroptosis, we systematically studied a number of naturally available vitamin compounds in mouse embryonic fibroblasts with tamoxifen (TAM)-inducible deletion of *Gpx4* (referred to as Pfa1 cells[8]) (Extended Data Fig. 1a). Notably, besides α-tocopherol (α-TOH), the most biologically active form of vitamin E, only the three forms of vitamin K, phylloquinone, menaquinone-4 (MK4) and menadione, rescued cells from ferroptosis induced by *Gpx4* deletion (Fig. 1a,b). Phylloquinone is obtained mostly from leafy green vegetables, and can be converted to MK4 in the body, whereas menadione is a synthetic variant. The anti-ferroptotic activity of vitamin K was not only limited to mouse

[1]Institute of Metabolism and Cell Death, Helmholtz Zentrum München, Neuherberg, Germany. [2]Division of Nephrology, Rheumatology and Endocrinology, Tohoku University Graduate School of Medicine, Sendai, Japan. [3]Laboratory of Food Function Analysis, Tohoku University, Sendai, Japan. [4]Department of Chemistry and Biomolecular Science, University of Ottawa, Ottawa, Ontario, Canada. [5]Universitätsklinikum Carl Gustav Carus Dresden, Technische Universität Dresden, Dresden, Germany. [6]Institute of Bioanalytical Chemistry, Faculty of Chemistry and Mineralogy, Leipzig University, Leipzig, Germany. [7]Department of Surgery, University Hospital Regensburg, University of Regensburg, Regensburg, Germany. [8]Physical Chemistry for Life Science Laboratory, Faculty of Pharmaceutical Sciences, Kyushu University, Fukuoka, Japan. [9]Division of Clinical Pharmacology and Therapeutics, Tohoku University Graduate School of Pharmaceutical Sciences and Faculty of Pharmaceutical Sciences, Sendai, Japan. [10]Monoclonal Antibody Core Facility, Helmholtz Zentrum München, Neuherberg, Germany. [11]Institute for Diabetes and Cancer, Helmholtz Zentrum München, Neuherberg, Germany. [12]Institute of Structural Biology, Helmholtz Zentrum München, Neuherberg, Germany. [13]Division of Medical Science, Tohoku University Graduate School of Biomedical Engineering, Sendai, Japan. [14]Present address: Zentrum Membranbiochemie und Lipidforschung, Medizinische Fakultät Carl Gustav Carus, Technical University, Dresden, Germany. ✉e-mail: eikan@med.tohoku.ac.jp; marcus.conrad@helmholtz-muenchen.de

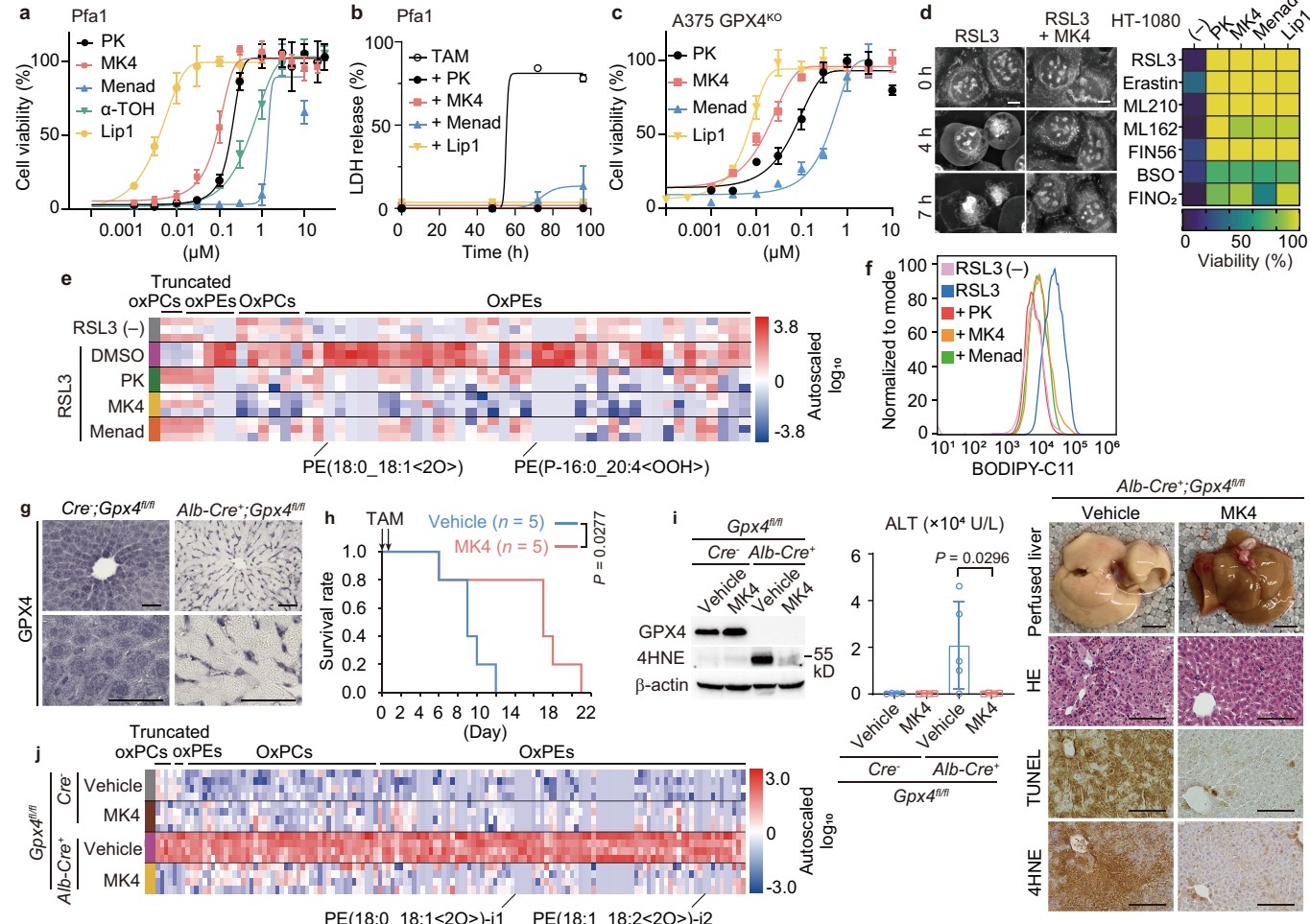

**Fig. 1 | Vitamin K is a potent class of anti-ferroptotic compounds. a**, Viability of 4-hydroxytamoxifen (4-OH-TAM)-induced GPX4-KO mouse embryonic fibroblasts (Pfa1 cells) treated with phylloquinone (PK), MK4, menadione (Menad), α-TOH or liproxstatin-1 (Lip1), a well-established ferroptosis inhibitor. **b**, Lactate dehydrogenase (LDH) release by Pfa1 cells at the indicated time after addition of 4-OH-TAM. Phylloquinone (1 μM), MK4 (1 μM), menadione (3 μM) or Lip1 (0.5 μM). **c**, Viability of A375 GPX4-KO cells treated with the indicated compound five days after withdrawal of Lip1, which is required to maintain the GPX4-KO cells. **d**, Left, images of HT-1080 cells treated with RSL3 (0.5 μM) with or without MK4 (3 μM) during live imaging (see also Supplementary Videos 1 and 2). Scale bars, 10 μm. Right, heat map showing the viability of HT-1080 cells co-treated with the indicated compounds and ferroptosis inducers. Phylloquinone (3 μM), MK4 (3 μM), menadione (3 μM) or Lip1 (0.5 μM) were added 1 h before the addition of the inducers. **e**, Heat map showing relative amounts of oxidized phospholipids in Pfa1 cells treated with RSL3 (0.5 μM) for 8 h. Phylloquinone, MK4 and menadione (3 μM) were added 6 h before the addition of RSL3. oxPE, oxidized phosphatidylethanolamines; oxPC, oxidized phosphatidylcholines. **f**, Lipid peroxidation evaluated by BODIPY 581/591 C11 staining of Pfa1 cells. RSL3 (−), cells without RSL3 treatment. **g**, Immunohistochemistry of GPX4 in the liver of TAM-induced hepatocyte-specific GPX4-KO (*Alb-creER[T2];Gpx4[fl/fl]*) mice and *cre[−]* mice after TAM injection. **h**, Survival rate of *Alb-creER[T2];Gpx4[fl/fl]* mice fed a diet with low vitamin E at the indicated time of MK4 treatment (200 mg kg[−1] day[−1], intraperitoneal injection) or vehicle. *n* = 5 per group. **i**, Evaluation of liver injury in *Alb-creER[T2];Gpx4[fl/fl]* mice analysed seven days after TAM injection. Left, immunoblotting for 4-hydroxynonenal (4HNE) and GPX4. Middle, serum levels of alanine transaminase (ALT). Right, images of liver and histology (haematoxylin and eosin staining (HE), TUNEL staining and immunohistochemistry for 4HNE). Scale bars, 5 mm (whole-liver images) and 100 μm (histology). *n* = 4 *cre[−]* and *Alb-cre[+]* with MK4; *n* = 5 *Alb-cre[+]* with vehicle. **j**, Heat map showing relative amounts of oxidized phospholipids in the liver collected seven days after TAM injection. Data are mean ± s.d. of *n* = 3 (**a**–**d**,**i**). Log-rank test (**h**); one-way ANOVA with Dunnett's test (**i**).

fibroblasts, as it also prevented ferroptosis in the human cancer cell lines A375 and 786-O that lack GPX4 expression (Fig. 1c and Extended Data Fig. 1b). Phylloquinone, MK4 and menadione also efficiently rescued cells from ferroptosis triggered by well-established ferroptosis inducers[17] including a GPX4 inhibitor (*1S,3R*)-RSL3 (RSL3) in fibrosarcoma HT-1080 cells (Fig. 1d and Supplementary Videos 1 and 2), as well as in other cancer and non-cancer cell lines (Extended Data Fig. 1c,d). In addition, all three vitamin K forms prevented glutamate-induced neuronal ferroptosis, whereas they failed to protect against other types of cell death, except protection of pyroptosis by menadione[18] (Extended Data Fig. 1e,f). Phylloquinone and MK4 showed no cellular toxicity up to 100 μM, although high doses of menadione (over 10 μM)

showed signs of toxicity, probably owing to the generation of reactive oxygen species, as reported[19] (Extended Data Fig. 1g). Menadione, which lacks an aliphatic sidechain, also rescued cells from ferroptosis−albeit with lower efficacy−whereas dimethylmenadione, a redox-inactive form, did not prevent ferroptosis (Extended Data Fig. 1h). This suggests that the quinone head group is a structural requirement for the anti-ferroptotic function.

As iron-dependent lipid peroxidation is the hallmark of ferroptosis, we evaluated the levels of lipid peroxidation by performing high-resolution liquid chromatography–mass spectrometry (LC–MS)-based epilipidomics analysis, staining cells with BODIPY 581/591 C11 and Liperfluo, and determining malondialdehyde concentrations

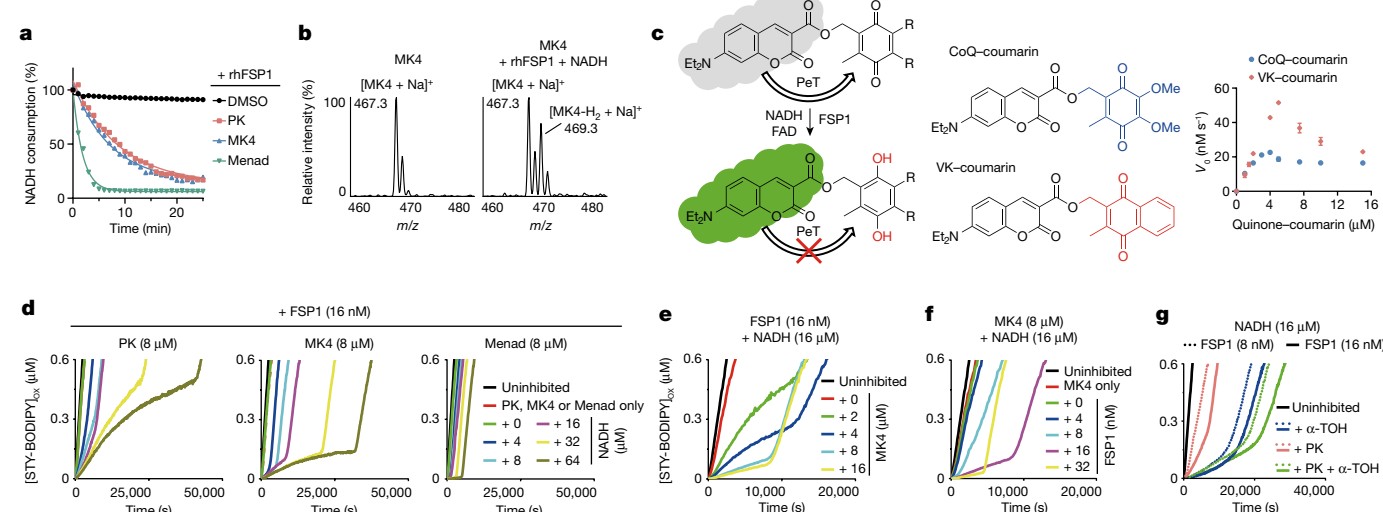

**Fig. 2 | FSP1 maintains vitamin K hydroquinone to act as a radical-trapping antioxidant. a,** NADH consumption assay using rhFSP1 in combination with the indicated form of vitamin K. **b,** Mass spectra of MK4 incubated with or without rhFSP1 and NADH. **c,** Coumarin conjugates of the naphthoquinone of vitamin K (VK) and quinone of coenzyme Q (CoQ) exhibit increased fluorescence upon reduction, enabling monitoring of quinone reduction by FSP1. Initial rates were determined for the reduction of quinone–coumarins by rhFSP1 in the presence of NADH. $n$ = 3. Et, ethyl group; PeT, photoinduced electron transfer. **d–g,** FENIX assay to determine lipid radical-trapping activity by FSP1-mediated vitamin K reduction. **d,** Inhibition of lipid peroxidation by phylloquinone, MK4 and menadione (8 µM) was evaluated in the presence of rhFSP1 (16 nM) and varying NADH concentrations (0–64 µM). **e,** Lipid peroxidation as a function of MK4 (0–16 µM) in the presence of FSP1 (16 nM) and NADH (16 µM). **f,** Lipid peroxidation as a function of FSP1 concentration (0–32 nM) with MK4 (8 µM) and NADH (16 µM). **g,** Lipid peroxidation in the presence of α-TOH (8 µM) with FSP1 (8 and 16 nM), NADH (16 µM) and phylloquinone (8 µM). Data are representative of three independent experiments (**a–g**).

(Fig. 1e,f and Extended Data Fig. 2a–c). These studies showed that all three forms of vitamin K efficiently suppressed lipid peroxidation, and through a mechanism independent of any iron-chelating effect (Extended Data Fig. 2d), clearly prevented the RSL3-induced formation of oxidized lipid species, of which the most prominent were long-chain oxidized phosphatidylethanolamine and phosphatidylethanolamine plasmalogens (Fig. 1e and Extended Data Fig. 2a). These data demonstrate that the vitamin K family of compounds acts as potent anti-ferroptotic agents.

## Vitamin K protects tissues from ferroptosis

We then tested whether vitamin K can also prevent ferroptosis in vivo using mice with genetic deletion of *Gpx4* and pathological models, in which ferroptosis results in tissue injury. We focused on MK4 as it was the most efficacious derivative (Fig. 1a,c and Extended Data Fig. 1b). First, we treated mice with TAM-inducible deletion of *Gpx4* in hepatocytes (referred to as *Alb-creER^T2^;Gpx4^fl/fl^* mice) (Fig. 1g). As reported previously, hepatocyte-specific *Gpx4*-knockout (KO) mice died soon after treatment as a result of widespread liver necrosis when the standard diet was switched to a vitamin E-deficient diet[15] (Extended Data Fig. 3a). Of note, treatment with a supra-nutritional level of MK4 extended the survival time of *Alb-creER^T2^;Gpx4^fl/fl^* mice under vitamin E-deficient conditions and robustly protected against related pathologic changes and lipid peroxidation in the liver (Fig. 1h–j and Extended Data Fig. 3b–d).

To address whether MK4 might also be protective in a model of ischaemia–reperfusion injury[10,20], we treated C57BL/6 mice with MK4 before liver or kidney ischaemia–reperfusion injury. Pre-treatment with MK4 in the mouse liver ischaemia–reperfusion injury model ameliorated liver injury with reduced lipid peroxidation, decreased hepatocyte cell death and diminished infiltration of inflammatory cells (Extended Data Fig. 4a). In the kidney ischaemia–reperfusion injury model, pre-treatment of MK4 also conferred protection against tissue damage reflected by a reduced number of terminal deoxynucleotidyl

transferase dUTP nick end labelling (TUNEL)-positive tubular cells, reduced expression of the tubular damage marker kidney injury molecule-1 (KIM-1) and improved kidney function (Extended Data Fig. 4b). We thus conclude that a pharmacological dose of vitamin K has a potent anti-ferroptotic effect preventing against cell death in relevant in vivo models of ferroptosis.

## FSP1 maintains VKH₂ to act as an RTA

Vitamin K is a redox-active naphthoquinone, which is converted to its corresponding hydroquinone (VKH₂) in the well-established vitamin K cycle[3] (Extended Data Fig. 4c,d). VKH₂ is reported to be a potent RTA[21] preventing lipid peroxidation[22] in addition to its canonical function as a cofactor for γ-glutamyl carboxylase (GGCX), which catalyses the carboxylation of vitamin K-dependent proteins, including coagulation factors. During the GGCX-mediated reaction, VKH₂ is oxidized to vitamin K epoxide, and then converted to vitamin K quinone by vitamin K epoxide reductase (VKOR), whose activity is inhibited by warfarin. The reduction of vitamin K to VKH₂ is also mediated by VKOR or an alternative warfarin-resistant pathway, catalysed by NAD(P)H-dependent vitamin K reductase activity[3,23] (Extended Data Fig. 4d). The identity of the warfarin-resistant vitamin K reductase remains unknown despite it was first described more than half a century ago[6,24].

FSP1 (encoded by the *AIFM2* gene), a NAD(P)H-dependent ubiquinone oxidoreductase, is a major means of ferroptosis control, acting independently of GPX4 by catalysing the reduction of ubiquinone to its hydroquinone form, ubiquinol, consuming NAD(P)H[4,5]. Since vitamin K and ubiquinone share structural properties (Extended Data Fig. 4c), we tested whether FSP1 could act as a vitamin K reductase, producing VKH₂ to inhibit lipid peroxidation. We interrogated this possibility using in vitro assays with recombinant human FSP1 (rhFSP1). When any of the three vitamin K forms were co-incubated with rhFSP1 and NADH, NADH was consumed (Fig. 2a). In addition, when MK4 was incubated with rhFSP1 and NADH, MK4-hydroquinone was generated (Fig. 2b and Extended Data Fig. 5a), demonstrating that FSP1 catalyses the reduction

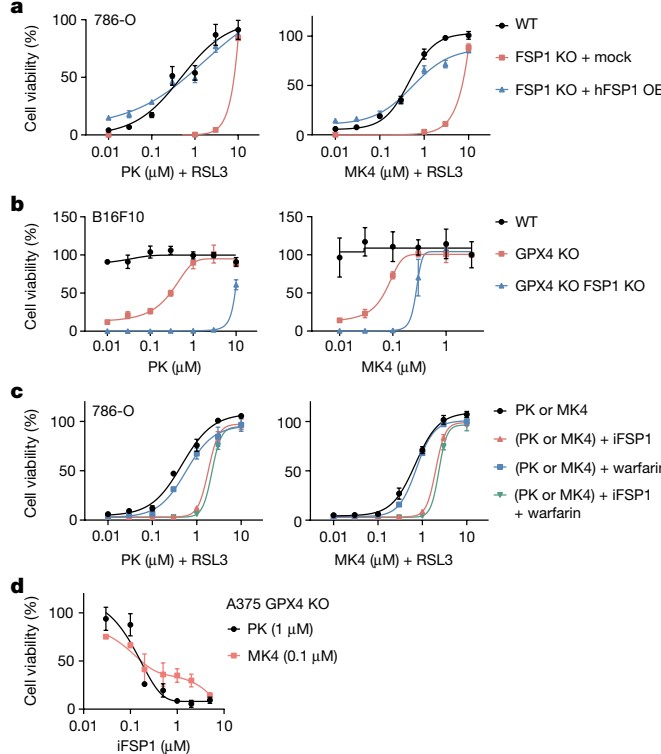

**Fig. 3 | FSP1-mediated anti-ferroptotic effect of vitamin K. a**, Protective effects of phylloquinone and MK4 against RSL3 (1 μM)-induced ferroptosis in wild-type and FSP1-KO 786-O cells overexpressing (OE) human FSP1 (hFSP1) or with mock transfection. Phylloquinone and MK4 were added 1 h before the addition of RSL3. **b**, Protective effects of phylloquinone and MK4 in GPX4-KO and GPX4 and FSP1 double-knockout B16F10 cells. Viability was assessed four days after withdrawal of Lip1. Viability was normalized to that of cells treated with Lip1 (1 μM; 100%). **c**, After pre-treatment of iFSP1 (10 μM) and/or warfarin (5 μM) with indicated concentrations of phylloquinone and MK4, 786-O cells were treated with RSL3 (1 μM). **d**, The effect of iFSP1 on the viability of GPX4-KO A375 cells treated with phylloquinone (1 μM) and MK4 (0.1 μM) for 3 days after withdrawal of Lip1. Data are mean ± s.d., *n* = 3 experiments (**a–d**).

of vitamin K. Given the instability of VKH₂ to autoxidation (Extended Data Fig. 5b,c), which complicates quantification and kinetic characterization of the enzyme, we synthesized a pro-fluorescent vitamin K analogue to enable direct monitoring of FSP1 activity using fluorescence (Fig. 2c and Extended Data Fig. 6a–d). This vitamin K analogue proved to be a good substrate for FSP1 with similar Michaelis constant ($K_M$) and maximum reaction velocity ($v_{max}$) than a pro-fluorescent ubiquinone analogue. Of note, FSP1-mediated enzymatic activity could be prevented by the FSP1 inhibitor iFSP1[4], but was insensitive to warfarin (Extended Data Fig. 6e).

We evaluated lipid radical-trapping activity by FSP1-mediated vitamin K reduction using the fluorescence-enabled inhibited autoxidation (FENIX) assay[25], in which liposomal lipid peroxidation is monitored fluorometrically by the competitive oxidation of STY-BODIPY (Extended Data Fig. 7a). Although phylloquinone, MK4 and menadione themselves are not inhibitors of lipid peroxidation, in the presence of rhFSP1 and NADH, the three vitamin K forms efficiently suppressed lipid peroxidation (Fig. 2d). NADH itself did not suppress lipid peroxidation (Extended Data Fig. 7b), but the supply of NADH clearly extended the duration of the inhibited period of each of the three forms of vitamin K with FSP1, indicating that NADH functioned as the stoichiometric reductant (Fig. 2d). The oxidation rate depended on the concentrations of NADH, vitamin K and FSP1 (Fig. 2d–f and Extended Data Fig. 7c,d), supporting

the notion that VKH₂ produced by the FSP1-catalysed reduction of vitamin K is the active RTA preventing lipid peroxidation. MK4 and phylloquinone, which both possess side chains, showed higher initial rates of oxidation, but inhibited the oxidation far longer in the presence of FSP1 and NADH (inhibition rate constant ($k_i$) = $5.4 \times 10^3$ and $k_i = 1.5 \times 10^3$ M⁻¹ s⁻¹, respectively); by contrast, menadione, the least lipophilic form, displayed the fastest radical-trapping kinetics, as indicated by the most suppressed initial rate combined with the shortest inhibition period ($k_i = 1.1 \times 10^4$ M⁻¹ s⁻¹ Extended Data Fig. 7i). Thus, poorer dynamics of phylloquinone and MK4 in the lipid bilayer owing to the lengthy side chains may give rise to their localization within the lipid membrane, suppressing their autoxidation and the consumption of reducing equivalents from NADH or NADPH (Extended Data Fig. 7e). Notably, in the presence of FSP1, the RTA activity of phylloquinone was similar to that of ubiquinone, whereas MK4 was much more efficient (compare Fig. 2d–f and Extended Data Fig. 7f). Unlike menadione, dimethylmenadione did not show any RTA activity (Extended Data Fig. 7g). It follows that the naphthoquinone head group confers RTA function to vitamin K, which is sufficient to prevent lipid peroxidation and subsequent ferroptosis.

Given the fact that ubiquinol can work in concert with α-TOH to suppress lipid peroxidation, and that we observed such a synergy enabled by FSP1 in our previous work[4], we also investigated the combination of vitamin K derivatives with α-TOH in the presence of FSP1. In each case, the inhibition period was extended, but the oxidation rate was the same as in the presence of α-TOH alone (Fig. 2g and Extended Data Fig. 7h,i), implying that it is the reactive RTA that is regenerated from the α-tocopheroxyl radical (α-TO•) by VKH₂ as is the case for ubiquinone and ubiquinol (Extended Data Fig. 8a). We additionally confirmed the RTA activity of VKH₂ reduced chemically or by FSP1-mediated reaction using LipiRADICALGreen (previously called NBD-Pen[26,27]), a fluorescence probe for lipid-derived radicals (Extended Data Fig. 8b–f).

## Vitamin K blocks ferroptosis via FSP1

Consistent with the observation that FSP1-mediated vitamin K reduction is responsible for RTA activity, phylloquinone and MK4 showed a diminished anti-ferroptotic effect against RSL3 in FSP1-KO cells similar to α-TOH, which can also be regenerated by FSP1[4] (Fig. 3a and Extended Data Fig. 9a–c). Reconstitution of FSP1 expression recovered the anti-ferroptotic function of phylloquinone and MK4, whereas expression of the myristoylation-defective G2A mutant of FSP1[4] did not rescue the protective effects of phylloquinone and MK4 in FSP1-KO cells (Fig. 3a and Extended Data Fig. 9d). In line with these findings, GPX4 and FSP1 double-KO cells required higher concentrations of phylloquinone and MK4 to prevent ferroptosis than GPX4 single-KO cells (Fig. 3b). Pharmacological inhibition of FSP1 by iFSP1[4] also diminished the protective effects of phylloquinone and MK4 against ferroptosis induced by RSL3 and by genetic deletion of *GPX4* (Fig. 3c,d). Treatment with warfarin and dicoumarol (an NAD(P)H quinone dehydrogenase 1 (NQO1) inhibitor) did not significantly influence the anti-ferroptotic activity of phylloquinone and MK4 (Fig. 3c and Extended Data Fig. 9e), although warfarin suppressed the protective effect of MK4 epoxide by inhibiting the conversion to MK4 (Extended Data Fig. 9f). These findings indicate that the reduction of vitamin K by FSP1 is responsible for the anti-ferroptotic action of phylloquinone and MK4. However, even in FSP1-KO cells, high doses of phylloquinone and MK4 still prevented ferroptosis (Fig. 3c), suggesting that other mechanisms (although less efficient) may contribute to the reduction of vitamin K. Indeed, menadione can be reduced non-enzymatically by glutathione (Extended Data Fig. 5c), and enzymatically by NQO1 and/or thioredoxin reductase in addition to FSP1[28,29]. Thus, genetic deletion and pharmacological inhibition of FSP1 did not significantly influence the anti-ferroptotic effect of menadione, similar to other FSP1-independent ferroptosis inhibitors (Extended Data Fig. 9c,g).

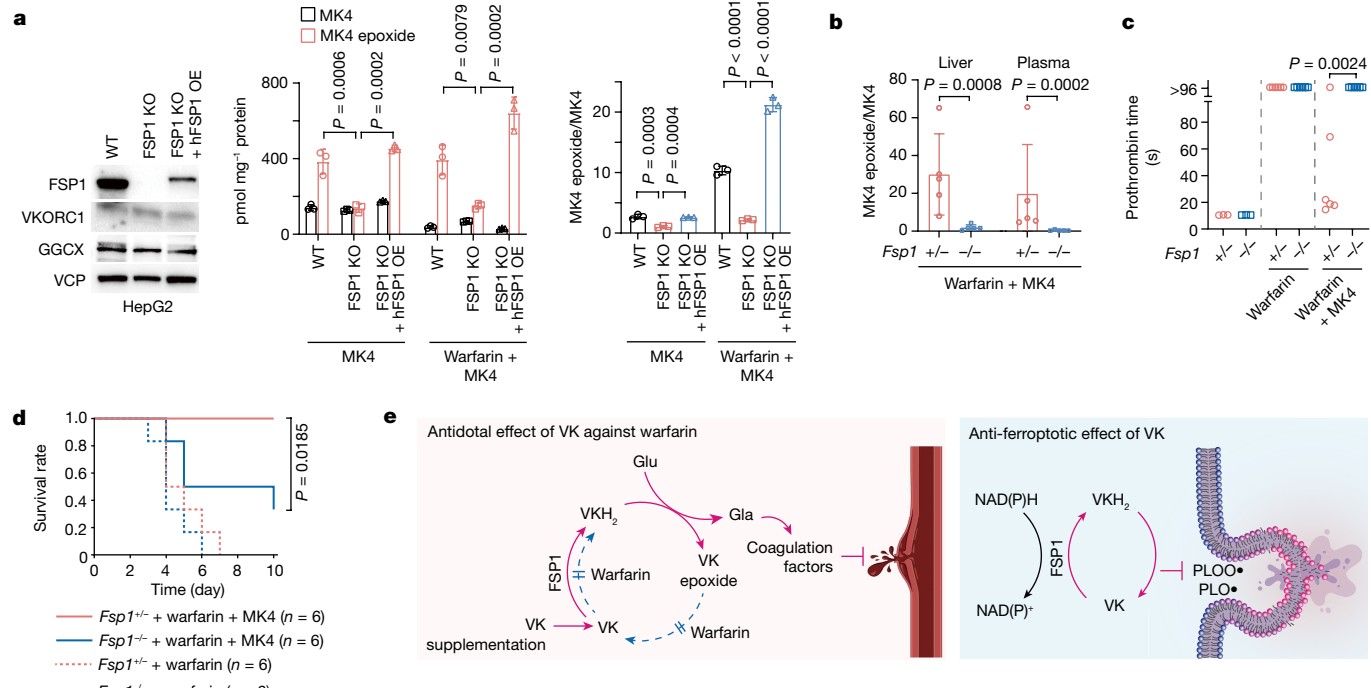

**Fig. 4 | FSP1 is the vitamin K reductase that overcomes warfarin poisoning.**
**a**, Left, immunoblotting of lysates of FSP1-KO and hFSP1-overexpressing
FSP1-KO HepG2 cells. Levels of MK4 and MK4 epoxide (middle) and the ratio of
MK4 epoxide to MK4 (right) in HepG2 cells treated with MK4 (3 μM) for 7 h in the
absence or presence of warfarin (10 μM). $n = 3$. **b**, Ratio of MK4 epoxide to MK4
in the liver and plasma of $Fsp1^{+/-}$ and $Fsp1^{-/-}$ mice treated with high-dose warfarin
(0.33 mg ml$^{-1}$ in drinking water) with MK4 (20 mg kg$^{-1}$, subcutaneous injection).
$n = 5$. **c**, Prothrombin time of $Fsp1^{+/-}$ and $Fsp1^{-/-}$ mice treated with high-dose
warfarin with or without MK4 (20 mg kg$^{-1}$, subcutaneous injection). $n = 3$
(no treatment), $n = 6$ (warfarin and warfarin + MK4). **d**, Survival rate of $Fsp1^{+/-}$
and $Fsp1^{-/-}$ mice treated with high-dose warfarin. The mice were injected with
MK4 (10 mg kg$^{-1}$ day$^{-1}$, subcutaneous injection) or vehicle during warfarin
administration. $n = 6$ in each group. Data are mean ± s.d. (**a**,**b**). One-way
ANOVA with Dunnett's test (**a**), two-tailed $t$-test using log-transformed values
(**b**), two-tailed $t$-test (**c**) and log-rank test (**d**). **e**, Right, graphical abstract
depicting the anti-ferroptotic function of vitamin K via FSP1-mediated
reduction and lipid radical-trapping activity, thus constituting a non-canonical
vitamin K redox cycle. Left, FSP1 also functions as the warfarin-resistant
vitamin K reduction pathway overcoming warfarin poisoning in the canonical
cycle. VK, vitamin K; VKH$_2$, vitamin K hydroquinone; Glu, glutamate;
Gla, γ-carboxyglutamate; PLOO•, phospholipid peroxyl radical; PLO•,
phospholipid alkoxyl radical. The illustration of the vessel was created using
BioRender.com.

## Antidotal FSP1 averts warfarin poisoning

Warfarin is one of the most widely prescribed anticoagulant drugs
worldwide. High-dose vitamin K is an effective antidote for warfarin
poisoning[30,31] because sufficient input of vitamin K can provide VKH$_2$ for
GGCX through the alternative vitamin K reduction pathway, bypassing
dysfunctional VKOR by the unidentified antidotal enzyme[32] (Extended
Data Fig. 4d). Since FSP1 was capable of reducing vitamin K, we tested
whether FSP1 is the enzyme responsible for the warfarin-resistant alter-
native vitamin K reduction pathway in the canonical GGCX–VKOR–
mediated cycle[3]. When human liver HepG2 cells were treated with MK4,
it was immediately metabolized to the MK4 epoxide via the hydroqui-
none and then converted to MK4 again[33]. However, the conversion effi-
ciency to the MK4 epoxide was significantly lower in the FSP1-deficient
cells, especially when cells were treated with warfarin (Fig. 4a), indicat-
ing that FSP1 is responsible for vitamin K reduction, in addition to VKOR,
in the canonical cycle. We next examined the function of FSP1 in this
pathway using $Fsp1^{+/-}$ and $Fsp1^{-/-}$ mice subjected to warfarin overdose in
the presence or absence of MK4 treatment (Extended Data Fig. 10a–c).
When a high dose of warfarin was administered, the $Fsp1^{-/-}$ mice treated
with MK4 showed much less conversion of MK4 to MK4 epoxide (Fig. 4b
and Extended Data Fig. 10d,e) and still showed extremely prolonged
prothrombin times (a parameter of vitamin K-dependent coagula-
tion factors) in contrast to $Fsp1^{+/-}$ mice (Fig. 4c). Whereas almost all
warfarin-treated groups had to be euthanized, mainly owing to cerebral

bleeding, one $Fsp1$ allele was sufficient to enable complete rescue by
high-dose vitamin K treatment (Fig. 4d and Extended Data Fig. 10f,g),
corroborating that FSP1 is the warfarin-resistant vitamin K reductase
in the canonical vitamin K cycle.

## Discussion

Long before the term ferroptosis was introduced in 2012[1], an anti-
oxidative effect of vitamin K was reported[22,34], although its mecha-
nism remained obscure. Here, we show that vitamin K confers robust
anti-ferroptotic activity via its reduced form, VKH$_2$. We further demon-
strate that the previously recognized ferroptosis suppressor FSP1 is the
vitamin K reductase that sustains a warfarin-insensitive non-canonical
vitamin K cycle that suppresses ferroptosis by maintaining VKH$_2$ at
the expense of NAD(P)H to prevent lipid peroxidation (Fig. 4e). Fur-
thermore, our data unveil FSP1 as the antidotal enzyme overcoming
warfarin poisoning. Phylloquinone and menaquinone are electron car-
riers used in plants and bacteria, respectively, whereas eukaryotes use
ubiquinone. FSP1 thus reduces both of the electron transfer quinones,
generating RTAs. Considering the evolution of life, when environmental
oxygen concentrations increased after the great oxidation event in
primordial Earth, it appears that menaquinone was substituted by
ubiquinone as an electron carrier owing to its higher redox potential
and increased abundance compared with vitamin K[35]. Since ferroptosis
is an evolutionarily conserved cell death mechanism in diverse species

ranging from prokaryotes and plants to mammals[36], our findings suggest that vitamin K might be the most ancient type of naturally occurring anti-ferroptotic quinones.

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

## Methods

### Chemicals

Menaquinone-4 (V9378), phylloquinone (V3501), menadione (M5625), MK4 epoxide (75618), vitamin K3 epoxide (51455), α-TOC (T3251), warfarin (A2250), warfarin sodium (PHR1435), dicoumarol (M1390), L-buthionine sulfoximine (BSO; B2515) N-acetyl-L-cysteine (A7250), lipopolysaccharide (LPS; L2880) and MCC950 (5381200001) were purchased from Sigma-Aldrich. Ferrostatin-1 (Fer1; 17729), RSL3 (19288), FINO$_2$ (25096), ML162 (20455), ML160 (23282) and staurosporine (81590) were purchased from Cayman. The following chemicals were obtained as indicated: erastin (329600, Merck Millipore), 17-AAG (A10010, Adqoo), L-glutamate (16911-22, Nacalai tesque), menadiol (M323135, TRC), iFSP1 (8009-2626, ChemDiv), liproxstatin-1 (Lip1, S7699, Selleckchem), BV-6 (S7597, Selleckchem), Trolox (56510, Fluka), recombinant mouse TNF (PMC3014, Thermo Fishier), nigericin (N1495, Thermo Fisher), zVAD-FMK (ALX-260-02, Enzo Life Sciences) and Nec 1s (2263, BioVision).

### Cell lines

4-OH-TAM-inducible $Gpx4^{-/-}$ mouse immortalized fibroblasts (Pfa1) were described previously[8]. HT-1080 (CCL-121), 786-O (CRL-1932), A375 (CRL-1619), B16F10 (CRL-6475), H9C2 (CRL-1446), NRK49F (CRL-1570), C2C12 (CRL-1772), HepG2 (HB-8065), Jurkat (TIB-152), L929 (CCL-1) and HEK293T (CRL-3216) cells were obtained from ATCC. Panc-1 cells were obtained from Cell Resource Center for Biomedical Research, Institute of Development, Aging and Cancer, Tohoku University (Sendai, Japan). THP-1 cells were obtained from DSMZ (Germany). HT-22 cells were purchased from Millipore. All cell lines, except Jurkat and THP-1, were maintained in DMEM high glucose (4.5 g l$^{-1}$ glucose, 21969-035, Gibco) supplemented with 10% fetal bovine serum (FBS), 2 mM L-glutamine, and 1% penicillin/streptomycin at 37 °C with 5% CO$_2$, unless stated otherwise. Jurkat and THP-1 cells were maintained in RPMI1640 Glutamax medium (61870010, Gibco) supplemented with 10% heat-inactivated FBS. GPX4-KO cells were maintained in medium containing Lip1 (1 μM) to prevent ferroptosis. All cells were regularly tested for mycoplasma contamination.

### Cell viability assays

Cells were seeded on 96-well plates and allowed to adhere overnight. On the next day, cells were treated with cell death inducers. Vitamin K compounds were added to the medium 1 h prior to the treatment with the ferroptosis inducing agents. Warfarin, iFSP1 and dicoumarol were added during cell seeding on 96-well plates. Cell viability was assessed 24 h (RSL3, erastin, FINO$_2$, ML210, ML162, glutamate and staurosporine), 48 h (17AAG and FIN56) and 60 h (BSO) after the treatment using AquaBluer (MultiTarget Pharmaceuticals) as an indicator of viable cells. The cell viability was expressed as relative values compared to the control sample, which was defined as 100%. Pfa1 cells were seeded on 96-well plates (500 cells per well) and treated in a dilution series of the compounds and 1 μM 4-OH-TAM to induce the KO of $Gpx4$. Cell viability of Pfa1 cells was assessed 72 h after TAM treatment. To directly monitor cell death, LDH release was used, whereby LDH activity in medium was measured using LDH Cytotoxity Detection kit (Takara Bio). For induction of ferroptosis in HT-1080 cells (seeded on 96-well plates at 2,000 cells per well), RSL3 (0.3 μM), erastin (2 μM), ML210 (1 μM), ML162 (1 μM), FIN56 (1 μM), BSO (500 μM) and FINO2 (5 μM) were used. For induction of ferroptosis in 786-O cells by RSL3, 3,000 cells per well were seeded in 96-well plates. For cell viability assay of A375 GPX4-KO cells and 786-O GPX4-KO cells, 500 cells per well were plated in 96-well plates and incubated without Lip1. For cell viability assay of H9C2 cells, 4,000 cells per well were plated in 96-well plates with low-glucose DMEM (1.0 g l$^{-1}$ glucose, Gibco) supplemented with 1% FBS to enhance susceptibility to ferroptosis[27]. For apoptosis induction, Jurkat cells (20,000 cells per well) were incubated with soluble human Fas ligand (30 ng ml$^{-1}$, ALX-522-020, Enzo Life Sciences) for 24 h, and Pfa1 cells (1,500 cells per well) were co-incubated with mouse TNF (10 ng ml$^{-1}$) and BV-6 (400 nM) for 24 h. For necroptosis induction, L929 cells (10,000 cells per well) were co-incubated with mouse TNF (10 ng ml$^{-1}$), BV-6 (400 nM) and zVAD-FMK (30 μM) for 24 h. For pyroptosis induction, LPS (1 μg ml$^{-1}$, 4h)-stimulated THP-1 cells (20,000 cells per well) were pretreated with vitamin K or MCC950 for 1 h and then incubated with nigericin (10 μM) for 2 h.

### Preparation of lentiviral particles

Lentiviral packaging system consisting of a transfer plasmid, psPAX2 (12260, Addgene), and pMD2.G (12259, Addgene) was co-lipofected into HEK293T cells using the X-tremeGENE HP agent (Roche). Cell culture supernatants containing viral particles were collected 48 h after the transfection and used to transduce the cell line of interest after filtration using a 0.45 μm low protein binding syringe filter.

### CRISPR–Cas9-mediated gene knockout

Single guide RNAs (sgRNA) were designed to target critical exons of the genes of interest to be inactivated as listed in Supplementary Table 1. The guides were cloned in the $BsmBI$-digested lentiCRISPR v2-blast and lentiCRISPR v2-puro vectors (98293 and 98290, Addgene), or $BbsI$-digested pKLV-U6gRNA($Bbs$I)-PGKpuro2aBFP vector (50946, Addgene).

### Transient expression of the CRISPR–Cas9 system

A375 and B16F10 cells were transiently co-transfected with the desired sgRNA expressing lentiCRISPR v2-blast and lentiCRISPR v2-puro using the X-tremeGENE HP agent (Roche). One day after transfection, cells were selected by incubation with puromycin (1 μg ml$^{-1}$) and blasticidin (10 μg ml$^{-1}$). After selection, single-cell clones were picked and knockout clones were identified by sequencing out-of-frame mutations and immunoblotting.

### Stable expression of CRISPR–Cas9 system

786-O and HepG2 cells were infected with VSV-G pseudotyped lentiviral particles containing the transfer plasmid of lentiCRISPRv2-puro and lentiCRISPRv2-blast. One day after transfection, cells were treated with selection antibiotics (blasticidin 15 μg ml$^{-1}$ and puromycin 1.5 μg ml$^{-1}$ for 786-O, and puromycin 1 μg ml$^{-1}$ for HepG2). After the selection, loss of FSP1 expression in HepG2 cells was confirmed by immunoblotting of batch cultures. Regarding 786-O GPX4-KO cells, single-cell clones were picked and individually expanded. KO clones were identified by immunoblotting.

### Doxycycline-inducible Cas9 expression system

Dox-inducible Cas9 expressing cells were generated by transducing 786-O and HT-1080 cells with VSV-G coated ecotropic lentiviral particles containing pCW-Cas9-Blast (83481, Addgene). After blasticidin selection (15 and 10 μg ml$^{-1}$ for 786-O and HT-1080, respectively), single-cell clones expressing Dox-inducible Cas9 were identified by immunoblotting. pCW-Cas9-Blast expressing 786-O and HT-1080 cells were used to generate FSP1-KO cells by lentiviral infection with particles containing the desired sgRNA expressing pKLV-U6gRNA(sgRNA)-PGKpuro2ABFP. One day after infection, cells were selected with puromycin (1.5 and 1 μg ml$^{-1}$ for 786-O and HT-1080, respectively), and then incubated with doxycycline (Dox) (10 μg ml$^{-1}$) for 5 days to express Cas9. After Cas9 induction, loss of FSP1 expression in the cell pool was confirmed by immunoblotting.

### Overexpression and Dox-inducible expression of FSP1

Codon-optimized human $FSP1$ gene with a C-terminal HA tag was cloned in the expression vector p442-Neo and Dox-inducible lentivirus vector pSLIK-Neo (25735, Addgene). FSP1-KO cells were infected with VSV-G pseudotyped lentiviral particles containing the hFSP1-cloned

transfer plasmids. One day after infection, cells were selected with geneticin (1 mg ml$^{-1}$). Reconstitution of FSP1 expression was verified by immunoblotting. Dox-inducible FSP1 expression was additionally verified by immunoblotting after treatment with increased concentrations of Dox for 24 h. For determining cell viability, cells were treated with increasing concentrations of Dox overnight and maintained in medium containing the same concentration of Dox during the assay period.

## Live-cell imaging
HT-1080 cells (80,000 cells) were seeded on μ-Dish 35 mm low (80136, i-bidi) and incubated overnight. On the next day, the cells were treated with or without 3 μM MK4 1 h before the addition of 0.5 μM RSL3. Live-cell imaging was performed using 3D Cell Explorer and Eve software v1.8.2 (Nanolive). Images were obtained at 1 min intervals. During imaging, the cells were maintained at 37 °C and 5% $CO_2$ by using a temperature-controlled incubation chamber.

## BODIPY 581/591 C11 staining
Pfa1 cells (50,000 cells per well) were seeded on 6-well dishes one day prior to the experiment. On the next day, cells were treated with 0.3 μM RSL3. Phylloquinone (3 μM), MK4 (3 μM), menadione (3 μM) and Lip1 (1 μM) were added 3 h prior to the addition of RSL3. Three hours after the addition of RSL3, cells were incubated with 1.5 μM of BODIPY 581/591 C11 (Thermo Fisher) for 30 min at 37 °C. Subsequently, cells were trypsinized, resuspended in 300 μl of Hanks' balanced salt solution (HBSS, Gibco), strained through a 40 μm cell strainer (Falcon tube with cell strainer CAP), and then analysed using a flow cytometer (CytoFLEX and CytExpert 2.4, Beckman Coulter) with a 488-nm laser paired with a 530/30 nm bandpass filter. Data were analysed using FlowJo Software 10 (Treestar).

## Liperfluo staining
Cellular lipid hydroperoxides were detected using the fluorescent probe Liperfluo (Dojindo). H9C2 cells were plated onto black, clear-bottom μClear 96-well culture plates (Greiner). After removal of the medium, the cells were incubated in HBSS containing 2 μM of Liperfluo. Subsequently, the cells were incubated with 100 μM BSO and the indicated compounds for 40 h. After the incubation, the cells were washed with HBSS and observed using a BZ-X800 fluorescence microscope (Keyence). The signal intensity per cell was measured with ImageJ software v1.53 (NIH).

## Iron-chelating activity assay
Iron-chelating activity was measured by Metalloassay kit Fe (FE02M, Metallogenics). After the addition of the compounds (final concentration, 100 μM) into an iron standard solution of 200 μg dl$^{-1}$ of iron(III) nitrate, free iron levels in the solution were measured according to the protocol.

## Epilipidomics analysis
Lipids from cells were extracted using the methyl-*tert*-butyl ether (MTBE) method[37]. In brief, cell pellets (4 to 6 × 10$^6$ cells) collected in phosphate-buffered saline (PBS) containing dibutylhydroxytoluene (BHT, 100 μM) and diethylenetriamine pentaacetate (100 μM) were washed and centrifuged. Splash Lipidomix (Avanti Polar Lipids) was added (5 μl) and incubated on ice for 15 min. After ice-cold methanol (375 μl) and MTBE (1,250 μl) were added, samples were vortexed and incubated for 1 h at 4 °C (Orbital shaker, 32 rpm). Phase separation was induced by the addition of water (375 μl), vortexed, incubated for 10 min at 4 °C (Orbital shaker, 32 rpm), and centrifuged to separate organic and aqueous phase (10 min, 4 °C, 1,000$g$). The organic phase was collected, dried in the vacuum concentrator and redissolved in 53 μl of isopropanol, centrifuged and 50 μl were transferred in glass vials for LC–MS analysis.

Lipids from mouse livers (approximately 150 mg wet tissue weight) were extracted according to Folch extraction method[38]. SPLASH LIPIDO-MIX (Avanti Polar Lipids, 30 μl) was added. Samples were homogenized in methanol (1 ml) by cryomilling and transferred in 10 ml glass tubes. Lysis beads were washed with methanol (400 μl) and chloroform (1,000 μl). Additional 1.8 ml chloroform was added, samples were vortexed (2 min, 2500 rpm) and incubated for 1 h at 4 °C with rotation (32 rpm). Phase separation was induced by adding water (840 μl). Samples were mixed by vortexing and incubated for 10 min 4 °C with rotation, before centrifugation (10 min, 1,000$g$, 4 °C). The lower, organic phase containing lipids was collected into new glass vials. For re-extraction, chloroform (2.8 ml) was added, samples vortexed, incubated (1 h, 4 °C, and 32 rpm), and centrifuged (10 min, 1,000$g$, 4 °C). The organic phases, combined from both extractions, were dried in a vacuum. Lipids were reconstituted in 300 μl IPA, and transferred in glass vials for LC–MS analysis. To avoid oxidation, all solvents used for lipid extraction were spiked with 0.1% (w/v) BHT and cooled on ice before use.

Reversed phase liquid chromatography (RPLC) was carried out on a Vanquish focused+ (Thermo Fisher Scientific) equipped with an Accucore C30 column (150 × 2.1 mm; 2.6 μm, 150 Å, Thermo Fisher Scientific). Lipids were separated by gradient elution with solvent A (acetonitrile/water, 1:1, v/v) and B (isopropanol/acetonitrile/water, 85:15:5, v/v) both containing 5 mM $NH_4HCO_2$ and 0.1% (v/v) formic acid. Separation was performed at 50 °C with a flow rate of 0.3 ml min$^{-1}$ using the following gradient: 0–20 min, 10 to 86% B (curve 4); 20–22 min, 86 to 95% B (curve 5); 22–26 min, 95% isocratic; 26–26.1 min, 95 to 10% B (curve 5); followed by 5 min re-equilibration at 10% B. Mass spectrometry analysis was performed on Thermo Scientific Q Exactive Plus Quadrupole-Orbitrap (Thermo Fisher Scientific) equipped with a heated electrospray (HESI) source and operated in negative ion mode with the following parameters: sheath gas 40 arbitrary units, auxiliary gas 10 arbitrary units, sweep gas 1 arbitrary units, spray voltage 2.5 kV, capillary temperature 300 °C, S-lens RF level 35%, and aux gas heater temperature 370 °C. For relative quantification of oxidized lipids, retention time scheduled parallel reaction monitoring using elemental composition of previously identified or computationally predicted oxidized lipids as precursors was used in negative ion mode at the resolution of 17,500 at $m/z$ 200, AGC target of 2 × 10$^5$ and a maximum injection time of 200 ms. The isolation window for precursor selection was 1.2 $m/z$, and normalized stepped collision energy (20–30–40 and 30–40–50 eV for phospholipids and neutral lipids, respectively) was used for HCD. Data were acquired in profile mode.

Acquired data were processed by Skyline v. 21.1[39] considering fragment anions of oxidized fatty acyl chains as quantifiers. The obtained peak areas were normalized by appropriate lipid species from SPLASH LIPI-DOMIX Mass Spec Standard (Avanti), e.g. by LPC(18:1(d7)), LPE(18:1(d7)), PC(15:0/18:1(d7)), or phosphatidylethanolamine (15:0/18:1(d7)), and the sample weights. Normalized peak areas were further log-transformed and auto-scaled in MetaboAnalyst online platform v5.0 (https://www.metaboanalyst.ca)[40]. Zero values were replaced by 0.2× the minimum values detected for a given oxidized lipid within the samples. Oxidized lipids showing a significant difference (ANOVA, adjusted $P$-value (false discovery rate (FDR)) cutoff: 0.05) between samples were used for the heat maps. The heat maps were created in Genesis v1.8.1 (Bioinformatics TU-Graz)[41]. The colour scheme corresponds to auto-scaled log fold change relative to the mean log value within the samples. Shorthand notations for oxidized lipids are given using LipidLynxX system v0.9.24[42].

## Western blotting
Cells were lysed in LCW lysis buffer (0.5% Triton X-100, 0.5% sodium deoxycholate salt, 150 mM NaCl, 20 mM Tris-HCl, 10 mM EDTA, 30 mM sodium pyrophosphate tetrabasic decahydrate) containing protease and phosphatase inhibitor mixture (cOmplete and phoSTOP, Roche), and centrifuged at 15,000$g$, 4 °C for 20 min. The supernatant was

collected and used as the protein sample. Western blotting was performed by standard immunoblotting procedure with 12% SDS–PAGE gel, PVDF membrane, and primary antibodies against GPX4 (1:1,000, ab125066, Abcam), 4HNE (1 µg ml⁻¹, MHN-20P, JaICA), human FSP1 (1:1,000, sc-377120, Santa Cruz), mouse FSP1 (1:100, clone 1A1 rat IgG2a; and 1:2, hybridoma supernatant of clone 14D7 rat IgG2bϗ, developed in-house), VKORC1 (1:1,000, ab206656, Abcam), GGCX (1:1,000, ab197982, Abcam), β-actin–HRP (1:5,000, A3854, Sigma-Aldrich) and valosin containing protein (VCP, 1:10,000, ab11433, Abcam). Images were analysed with Image Lab 6.0 software (Bio-Rad).

## Generation of monoclonal antibodies against mouse Fsp1

Female Wistar rats (RjHan:Wi, age 160 days) were immunized subcutaneously and intraperitoneally with a mixture of 70 µg recombinant C-terminal-His tagged full-length mouse Fsp1 protein in 200 µl PBS, 5 nmol CpG2006 (TIB MOLBIOL), and 200 µl Incomplete Freund's adjuvant (Sigma-Aldrich). After 8 weeks, a boost without Freund's adjuvant was given intraperitoneally and subcutaneously 3 days before fusion. Fusion of the myeloma cell line P3X63-Ag8.653 (CRL-1580, ATCC) with the rat immune spleen cells was performed using polyethylene glycol 1500. After fusion, the cells were plated in 96-well plates using RPMI 1640 medium with 20% FBS, glutamine, pyruvate, non-essential amino acids and HAT media supplement (Hybri-Max, Sigma-Aldrich). Hybridoma supernatants were screened 10 days later in a flow cytometry assay for binding to c-His tagged Fsp1 protein captured via biotinylated mouse anti-His antibody (clone HIS 3D5, prepared in-house) to streptavidin beads (PolyAN). Hybridoma supernatant was incubated for 90 min with beads and Atto-488-coupled subclass-specific monoclonal mouse-anti-rat IgG. Antibody binding was analysed using ForeCyt software 8 (Sartorius). Positive supernatants were further validated by Western blotting. Selected hybridoma cells were subcloned by limiting dilution to obtain stable monoclonal cell lines.

## Production of purified recombinant human FSP1

Recombinant human FSP1 protein (rhFSP1) was produced in *Escherichia coli*, and purified by affinity chromatography with a Ni-NTA system as described previously[4].

## FENIX assays

Liposomes were prepared from egg phosphatidylcholine (egg PC, Sigma-Aldrich) in pH 7.4 TBS buffer (25 mM, extruded to 100 nm, Chelex-100 treated) according to our previous report[25,43]. Liposomes (from the above suspension), STY-BODIPY (from a 1.74 mM stock in DMSO) and the test quinone (from appropriate stock solutions in CH₃CN) were combined and diluted to 285 µl with pH 7.4 TBS buffer in the wells of a 96-well plate, such that the concentrations in the well were 1.0526 mM liposomes, 1.0526 µM STY-BODIPY and 2.1053, 4.2105, 8.4210 or 16.8421 µM quinone. This was followed by the addition of 5 µl rhFSP1 at desired concentrations (with 19.2 µM FAD in pH 7.4 TBS buffer). The plate was incubated at 37 °C in a plate reader for 1 min followed by a vigorous mixing protocol for 5 min. The plate was ejected from the plate reader, and 5 µl of NADH (appropriate concentrations in pH 7.4 TBS buffer) and 5 µl of DTUN (12 mM in ethanol) were added such that the final concentrations of reagents were: 1 mM liposomes, 1 µM STY-BODIPY, 2, 4, 8 or 16 µM quinone, 2, 4, 8, 16 or 32 nM rhFSP1, 320 nM FAD, 4, 8, 16, 32 or 64 µM NADH and 200 µM DTUN. The plate was incubated at 37 °C for 1 min followed by another 1 min wherein it was mixed and the fluorescence ($\lambda_{ex}/\lambda_{em}$ = 488/518 nm) recorded every 2 min for the duration of the experiment. For determinations of inhibition rate constants, the rate of initiation ($R_i$) under the exact experimental conditions was first determined from the inhibition period observed upon inclusion of PMC, for which $n$ = 2, as a representative data trace is shown in Extended Data Fig. 7a. The $R_i$ was calculated from the expression below to yield $R_i$ = 7.81 × 10⁻¹⁰ s⁻¹ from $t_{inh}$ = 10,240 s, where $t_{inh}$ is the inhibited period. This $R_i$ was used along with the expression in Extended

Data Fig. 7a to calculate the rate constants shown in Extended Data Fig. 7i.

$$R_i = \frac{[PMC] \times n}{t_{inh}}$$

## Synthesis of 1,4-dimethoxy-2-methylnaphthalene

To a solution of 2-bromo-1,4-dimethoxy-3-methylnaphthalene[44] (dimethylmenadione) (600 mg, 2.13 mmol) in THF (20 ml) was added *n*-butyllithium (1.02 ml, 2.5 M in *n*-hexane) dropwise at −78 °C; the mixture was stirred at −78 °C for 10 min, followed by the addition of 0.5 ml water. The cooling bath was removed and reaction mixture was warmed to room temperature. Solvent was removed, and the mixture was purified by column with ethyl acetate/hexanes as the eluent. 1,4-Dimethoxy-2-methylnaphthalene was obtained as a light-yellow oil (350 mg, 81% yield). 1H NMR (400 MHz, chloroform-*d*) δ 8.20 (*d*, *J* = 8.5 Hz, 1H), 8.03 (*d*, *J* = 8.4 Hz, 1H), 7.54–7.49 (*m*, 1H), 7.45–7.41 (*m*, 1H), 6.61 (*s*, 1H), 3.97 (*s*, 3H), 3.87 (*s*, 3H), 2.45 (*s*, 3H). 13C NMR (101 MHz, chloroform-*d*) δ 151.7, 147.1, 128.8, 126.6, 125.7, 125.4, 124.7, 122.3, 121.6, 106.9, 61.4, 55.7, 16.4.

## Synthesis of vitamin K–coumarin conjugates

2-(Hydroxymethyl)-3-methylnaphthalene-1,4-dione[44] (42 mg, 0.20 mmol) and 7-(diethylamino)-2-oxo-2*H*-chromene-3-carbonyl chloride[4] (67 mg, 0.24 mmol) were dissolved in CH₂Cl₂ (2 ml), followed by the addition of Et₃N (40 µl, 0.003 mol) and *N*,*N*-dimethylpyridin-4-amine (3 mg, 0.024 mmol). The mixture was stirred at room temperature for 12 h. Precipitation was formed during this process. The reaction mixture was filtered and washed with ether. Crude product was purified by recrystallization from ethyl acetate to give vitamin K–coumarin as a yellow solid (55.2 mg, 62% yield). ¹H NMR (600 MHz, chloroform-*d*) δ 8.38 (*s*, 1H), 8.12–8.10 (*m*, 2H), 7.75–7.72 (*m*, 2H), 7.33 (*d*, *J* = 9.1 Hz, 1H), 6.64 (*dd*, *J* = 9.1, 2.4 Hz, 1H), 6.47 (*d*, *J* = 2.4 Hz, 1H), 5.40 (*s*, 2H), 3.44 (*q*, *J* = 7.1 Hz, 4H), 2.37 (*s*, 3H), 1.22 (*t*, *J* = 7.1 Hz, 6H). ¹³C NMR (151 MHz, Chloroform-*d*) δ 185.3, 183.6, 163.7, 158.6, 158.1, 152.8, 149.5, 148.1, 139.6, 133.9, 133.8, 132.3, 132.0, 131.4, 126.7, 126.6, 110.2, 108.4, 108.2, 97.4, 58.2, 45.6, 13.2, 12.5. HRMS–ESI (*m/z*) calculated for C₂₆H₂₃NNaO₆ [M+Na]⁺: 468.1423, found 468.1430.

## Synthesis of CoQ–coumarin conjugates

2-(Hydroxymethyl)-5,6-dimethoxy-3-methylcyclohexa-2,5-diene-1,4-dione[45] (43 mg, 0.20 mmol) and 7-(diethylamino)-2-oxo-2*H*-chromene-3-carbonyl chloride[46] (67 mg, 0.24 mmol) were dissolved in CH₂Cl₂ (2 ml), followed by the addition of Et₃N (40 µl, 0.003 mol) and *N*,*N*-dimethylpyridin-4-amine (3 mg, 0.024 mmol). The mixture was stirred at room temperature for 12 h. Precipitation was formed during this process. The reaction mixture was filtered and washed with ether. Crude product was purified by recrystallization from ethyl acetate to give CoQ–coumarin as a yellow solid (49.1 mg, 54% yield). ¹H NMR (600 MHz, Chloroform-*d*) δ 8.36 (*s*, 1H), 7.34 (*d*, *J* = 8.9 Hz, 1H), 6.61 (*dd*, *J* = 9.0, 2.4 Hz, 1H), 6.45 (*d*, *J* = 2.3 Hz, 1H), 5.20 (*s*, 2H), 4.03 (*s*, 3H), 3.99 (*s*, 3H), 3.44 (*q*, *J* = 7.1 Hz, 4H), 2.19 (*s*, 3H), 1.22 (*t*, *J* = 7.1 Hz, 6H). ¹³C NMR (151 MHz, Chloroform-*d*) δ 184.3, 183.0, 163.6, 158.7, 158.1, 153.1, 149.6, 144.8, 144.8, 143.9, 135.4, 131.4, 109.9, 108.0, 107.9, 97.0, 61.4, 61.4, 57.6, 45.4, 12.5, 12.5. HRMS–ESI (*m/z*) calculated for C₂₄H₂₅NNaO₈ [M+Na]⁺: 478.1478, found 478.1480.

## Monitoring FSP1 activity with quinone–coumarin conjugates

FAD, NADH and rhFSP1 in pH 7.4 TBS buffer were added in succession to varying concentrations of vitamin K–coumarin or CoQ–coumarin conjugate in pH 7.4 TBS buffer at 37 °C (final concentration: 6 nM rhFSP1, 50 nM FAD, 200 µM NADH) and the initial rates of the reaction were obtained by monitoring the increase in the fluorescence upon the reduction of the quinone to the hydroquinone on a plate reader

($\lambda_{ex}/\lambda_{em} = 415/470$ nm). The raw fluorescence data were converted to hydroquinone concentrations using response factors of $4.64 \times 10^9$ RFU $\mu M^{-1}$ (for CoQ–coumarin) and $3.40 \times 10^9$ RFU $\mu M^{-1}$ (for vitamin K–coumarin) which were determined from a standard curve obtained from the maximum fluorescence recorded for various concentrations of the quinones in the presence of massive excess of either rhFSP1, FAD or NADH.

## LipiRADICAL Green assay

LipiRADICAL Green assay was performed according to a previous report[27] using the fluorescence probe LipiRADICAL Green, previously called NBD-Pen (FDV-0042, Funakoshi) with several modifications. Arachidonic acid (10931, Sigma-Aldrich) and soybean lipoxygenase (LOX) from Glycine max (L7395, Sigma-Aldrich) were used in an AA/LOX system. Ninety microlitres of PBS pH 7.4 containing 5 µM LipiRADICAL Green, 100 µg ml$^{-1}$ LOX, and the indicated final concentration of compounds were prepared in black-walled 384 well plates. Immediately after the addition of a 10 µl solution of 5 mM AA (final 500 µM) to the mixture, the fluorescence intensity (ex 470/em 530 nm) was measured every 30 s at 37 °C using a Spectra Max M5 plate reader (Molecular Devices). The intensity before the addition of AA was used as background. For the phosphatidylcholine hydroperoxide (PCOOH)/Fe$^{2+}$ system, PCOOH was enzymatically synthesized from 1-palmitoyl-2-linoleoyl-*sn*-glycero-3 -phosphocholine (16:0-18:2 PC, Avanti Polar Lipids) using soybean lipoxygenase-1 and chromatographically purified[47]. Ninety microlitres of water containing 5 µM LipiRADICAL Green, 10 µM PCOOH and the indicated final concentration of compounds were prepared in black-walled 384 well plates. Immediately after the addition of 10 µl solution of 500 µM of Fe(NH$_4$)$_2$(SO$_4$)$_2$ (final 50 µM) to the mixture, the fluorescence intensity was measured as described above. For LipiRADI-CAL Green assay using rhFSP1, NADH and AA/LOX system, 80 µl of PBS containing 5 µM LipiRADICAL Green, 200 µM NADH, 150 nM hFSP1, and 100 µg ml$^{-1}$ LOX were prepared. After the addition of 10 µl solution of 1 mM phylloquinone or MK4 (final 100 µM) and 10 µl solution of 5 mM AA (final 500 µM) to the mixture, the kinetics of LipiRADICAL Green fluorescence intensity was measured. For the PCOOH/Fe$^{2+}$ system, 80 µl of PBS containing 5 µM LipiRADICAL Green, 200 µM NADH, 150 nM hFSP1, and 20 µM PCOOH were prepared. After the addition of 10 µl solution of 3 mM phylloquinone or MK4 (final 300 µM) and 10 µl solution of 500 µM of Fe(NH$_4$)$_2$(SO$_4$)$_2$ (final 50 µM) to the mixture, the fluorescence intensity was measured.

## FSP1 activity assay by measuring NADH consumption

FSP1 enzymatic assay was performed as described with a minor modification[4]. NADH consumption was measured at 340 nm using 100 µl of enzyme reactions in PBS pH 7.4 on a 96-well plate. Enzyme reactions contained 150 nM rhFSP1, 200 µM NADH (freshly prepared in water) and 300 µM of different substrate candidates (phylloquinone, MK4, and menadione). A Spectra Max M5 Microplate Reader (Molecular devices) was used to determine the absorbance at 340 nm every 30 s. Reactions without NADH or without enzyme were used to normalize the results.

## FSP1 activity inhibitor assay

Enzyme reactions in PBS containing 150 nM rhFSP1, 200 µM NADH and the inhibitors (iFSP1, warfarin and dicoumarol) were prepared. After the addition of 100 µM resazurin sodium salt (Sigma-Aldrich), fluorescent intensity (ex 540/em 590 nm) was measured every 30 s.

## Chemical reduction of menadione

Menadione and menadiol (300 µM) were incubated with 1 mM DTT or 10 mM GSH in water at room temperature for 5 min, and then measured by absorbance spectrum ranging from 200 to 450 nm by using a Spectra Max M2e (Molecular Devices). Background control (in blank well) of absorbance values was subtracted from each individual absorbance value.

## Detection of MK4-H$_2$ by FIA–MS

To detect chemical reduction of MK4, 22.5 mM of MK4 (dissolved in chloroform, 10 µl) was diluted in methanol (190 µl), then 1 mg NaBH$_4$ was added. Reactant solution (10 µl) was collected before and 1, 15, 30, 60, and 120 min after the addition of NaBH$_4$, and dissolved in methanol (990 µl). The sample solution was analysed by flow injection analysis mass spectrometry (FIA-MS) using a LC–MS/MS system consisted of an Exion LC system connected to a QTRAP 6500$^+$ tandem mass spectrometer (SCIEX). To detect enzymatic reduction of MK4, 1 mM MK4 (dissolved in DMSO, 10 µl) was added with 170 µl PBS, 1.5 µM rhFSP1 (dissolved in PBS, 10 µl), and 2 mM NADH (dissolved in DDW, 10 µl). The solution was incubated at 37 °C for 30 min. After incubation, a part of the reactant solution (10 µl) was dissolved in methanol (990 µl) and analysed by FIA-MS. Mass spec parameters are described in Supplementary Table 2.

## Quantification of cellular MK4 and MK4 epoxide levels

HepG2 cells ($1 \times 10^6$ cells per well) were seeded on 6 well plates. On the next day, medium was replaced with fresh medium with or without warfarin (10 µM). On the following day, cells were incubated in the presence or absence of MK4 (3 µM) for 7 h. After washing with PBS three times, cells were trypsinized and collected. Cell pellets were suspended in 400 µl PBS, supplemented with 20 µl of MK4-d$_7$ (2 ng µl$^{-1}$ in ethanol, 25709, Cayman) as internal standard, and sonicated for 30 s with a sonication probe (Bronson Sonifer). In this procedure, 10 µl of cell lysate was analysed for protein determination with a BCA protein assay (Pierce BCA Protein Assay Kit, Thermo Fisher). Extraction of vitamin K and its metabolites from cells was performed as reported[33]. Four-hundred microlitres ethanol and 1.2 ml hexane were added to the cell lysate (in PBS, 400 µl) followed by shaking for 5 min. Samples were centrifuged at 1,000$g$ for 5 min, and the upper organic layer was collected. Re-extraction of the remaining aqueous phase was performed by addition of 150 µl ethanol and 450 µl hexane with subsequent vortexing. Samples were centrifuged at 1,000$g$ for 5 min. Collected organic layers were combined, spiked with 20 µl of phylloquinone (2 ng µl$^{-1}$ in ethanol) as recovery standard and evaporated under reduced pressure. Dried extracts were resuspended in 30 µl ethanol. Quantification of the target analytes (MK4 and MK4 epoxide) was achieved using an Agilent 5890 Series II gas chromatograph (GC) coupled with a Thermo Finnigan SSQ7000 single quadrupole mass spectrometer (MS). Chromatographic separation was carried out on a Restek Rtx-5Sil MS column (30 m × 0.25 mm internal diameter × 0.25 µm film thickness). Two microlitres of each sample was injected in splitless mode using helium as carrier gas at a constant pressure of 16 psi. The injection temperature was 280 °C. Initial column temperature was 90 °C held for 1.5 min, increased to 220 °C at a rate of 20 °C min$^{-1}$, followed by a second ramp to 320 °C at a rate of 10 °C min$^{-1}$ and held for 10 min. The mass spectrometer was operated in negative chemical ionization mode and the masses of the negative molecular ions were registered in single ion monitoring mode.

## Quantification of MK4 and MK4 epoxide in mouse samples

Blood samples of mice were collected by bleeding from the retroorbital plexus into citrate-treated tubes. After centrifugation (3,000$g$ for 10 min), plasma samples were obtained and stored −80 °C until analysis. Liver tissues were collected from mice after transcardiac perfusion with 10 ml PBS, snap-frozen into liquid nitrogen and stored at −80 °C. For sample preparation of plasma, 100 µl aliquots of plasma were transferred into glass tubes, spiked with 20 ng of MK4-d$_7$ and briefly mixed. Next, 2 ml ethanol, 4 ml hexane and 100 µl water containing butylated hydroxytoluene (0.1 %, w/v) were added. After vigorously mixing for 5 min, the samples were centrifuged at 2,200$g$ for 5 min at 4 °C. The upper layer was transferred into a clean glass tube, and the samples were then re-extracted by the addition of an equal volume of hexane. Both

supernatants were collected and evaporated in vacuo. The samples were dissolved in 2 ml hexane and loaded onto silica columns. For sample preparation of tissues, the tissues (kidney and liver) were weighed, transferred to lysing matrix tubes containing stainless steel beads (MP Biomedicals), and then thoroughly homogenized in 1 ml ethanol containing 20 ng of MK4-$d_7$. The tissue homogenates were transferred into glass tubes using glass Pasteur pipettes. Following the addition of 6 ml acetone containing BHT (0.1%, w/v), the homogenates were thoroughly mixed using a Ohaus Multi-Tube Vortex mixer, for 5 min at 2,500 rpm, allowed to stand for 5 min, and centrifuged at 2,200$g$ for 5 min at 4 °C. This procedure was repeated three times. Supernatants were collected and evaporated in vacuo. The samples were dissolved in 2 ml water and 6 ml hexane containing BHT (0.1%, w/v), thoroughly mixed, and centrifuged at 2,200$g$ for 5 min at 4 °C. The samples were evaporated in vacuo, dissolved in 2 ml hexane and loaded onto silica columns. Plasma and tissue extracted samples were applied to silica Sep-Pak extraction cartridges (500 mg per 3 ml, Waters) connected to a Visiprep SPE Vacuum Manifold (Supelco), which were preconditioned prior with 3 ml diethyl ether:hexane (1:1, v/v) and then 3 × 2 ml hexane. After sample application, the cartridges were washed with 2 ml hexane followed by 4 × 2 ml hexane containing BHT (0.1%, w/v) to remove concomitants. The vitamin K-containing fraction was then eluted with 4 ml diethyl ether:hexane (3:97, v/v). The eluate was evaporated in vacuo and the residue reconstituted in 100 μl water:methanol (2:98, v/v) for measurement with LC–MS/MS. As vitamin K is light sensitive, samples were protected from light during preparation and analysis. MK4 and MK4 epoxide were separated by reversed phase liquid chromatography (RPLC) on a Sciex Exion LC System equipped with a Kinetex F5 100 × 2.1 mm, 100 Å, 2.6 μm column (Phenomenex). Analytes were separated by gradient elution with mobile phase A ($H_2O$ containing 5 mM ammonium formate) and B (methanol), both containing 0.1% (v/v) formic acid. Separation was performed at 50 °C with a flow rate of 0.5 ml min$^{-1}$ using the following gradient: 0–1 min, 70 to 98% B; 1–3 min, 98% isocratic; 3–3.1 min, 98 to 70% B; and 3.1–5 min, 70% isocratic. MK4 and MK4 epoxide were quantified by LC–MS/MS electrospray ionization on a Sciex Triple Quad 7500 LC–MS/MS System, operating in positive mode. Settings were as follows: CUR 50 psi, IS 3,500 V, TEM 500 °C, GS1 20 psi, GS2 70 psi, MRM dwell time 55 ms, pause between mass range 5 ms and EP 10 V. The following parent-to-daughter transitions were monitored: $m/z$ 452.4 [M+H]$^+$ to $m/z$ 194.0 for MK4-$d_7$ with CE of 34 V and CXP of 10 V, $m/z$ 445.1 [M+H]$^+$ to $m/z$ 187.0 for MK4 with CE of 31 V and CXP of 10 V, m/z 461.2 [M+H]$^+$ to $m/z$ 161.0 for MK4 epoxide with CE of 34 V and CXP of 14 V. The limits of quantification for MK4 were 0.2 ng mg$^{-1}$ tissue and 0.1 ng ml$^{-1}$ plasma, and those for MK4 epoxide were 2.0 ng mg$^{-1}$ tissue and 1.0 ng ml$^{-1}$ plasma.

## Animal studies

All experiments were performed in compliance with the German Animal Welfare Law and have been approved by the institutional committee on animal experimentation and the government of Upper Bavaria (approved no. ROB-55.2-2532-Vet_02-18-13 and ROB-55.2-2532.Vet_03-17-68) and the State of Bavaria (permission granted by the government of Lower Franconia, approved No. 54-2532.1-19/13), the Landesdirektion Sachsen (TVV07/2021) involving an independent ethics committee and the Animal Committee of Tohoku University (approved No. 2019-BeA012, 2019-BeA014 and 2019PhA-010-01). Mice were kept under standard conditions with water and food ad libitum and in a controlled environment (22 ± 2 °C, 55 ± 5% humidity, 12 h light/dark cycle). For animal studies, mice were randomized into separate cages. Sex-matched littermates were used and experiments were intended to test a single variable.

## Hepatocyte-specific inducible *Gpx4*-KO mice

To generate mice with a TAM-inducible hepatocyte-specific deletion of *Gpx4* (*Alb-creER$^{T2}$;Gpx4$^{fl/fl}$*), *Gpx4$^{fl/fl}$* mice were first crossbred with *Alb-creER$^{T2}$* mice[48] (kindly provided by P. Chambon) to yield *Alb-creER$^{T2}$; Gpx4$^{fl/+}$* mice. These were then crossed with *Gpx4$^{fl/fl}$* mice to generate *Alb-creER$^{T2}$;Gpx4$^{fl/fl}$* mice and respective controls. To achieve inducible disruption of the floxed *Gpx4* alleles, mice were intraperitoneally injected with 2 mg TAM (dissolved in Miglyol 812, Caelo) on two consecutive days. Animals were equally distributed between sex and weight and were typically 8–10 weeks of age. For pharmacological treatment, vehicle or MK4 (100 mg kg day$^{-1}$ dissolved in Miglyol, twice daily) was intraperitoneally administered to the mice each day starting from 2 days before the first TAM injection until the completion of the study. The diet was changed from a standard diet (containing 143 mg kg$^{-1}$ vitamin E, no. 1314 Fortified, Altromin) to a vitamin E-deficient diet (containing <7 mg kg$^{-1}$ vitamin E, E15314-247, ssniff Spezialdiäten) at the timing of the first TAM injection. When animals reached the humane end point, they were immediately euthanized. For the end point analysis, the mice were euthanized 7 days after the first TAM injection, and the plasma and tissues were collected. Serum ALT were measured by AU480 chemistry analyser (Beckman Coulter). For the pharmacokinetic study of MK4, samples of plasma, liver and kidney were collected from *Gpx4$^{fl/fl}$* mice 0, 1, 3, 6 and 24 h after an intraperitoneal injection of MK4 (200 mg kg$^{-1}$ dissolved in Miglyol).

## Liver ischaemia–reperfusion injury model in mice

Eight to 10-week-old male C57BL/6J mice, provided by Charles River (Germany), were fed a standard diet (containing 135 mg kg$^{-1}$ vitamin E, ssniff Spezialdiäten) and underwent liver ischaemia–reperfusion injury as described previously[10]. In brief, mice were aneasthetized with xylazine/ketamine and shaved at their front. After opening the abdominal cavity an atraumatic clip was placed across the portal vein, hepatic artery and bile duct, just above branching to the right lateral lobe. After 90 min of ischaemia, the clamp was removed and the liver was reperfused. Mice were euthanized 24 h following transient ischaemia–reperfusion and blood and tissues were collected. MK4 (200 mg kg$^{-1}$ dissolved in Miglyol 812) or vehicle was injected intraperitoneally 24 h and 1 h before the onset of ischaemia. Serum ALT was measured using a Dimension 1500 Vista Analyzer (Siemens). Calculation of the necrotic/damaged areas (% of the whole section minus the major vessels) in the haematoxylin and eosin-stained sections were performed in a blinded manner using ZEISS Axio Vision software AxioVs v4.9 (Carl Zeiss).

## Kidney ischaemia–reperfusion injury model in mice

Eight- to twelve-week-old male C57BL/6N mice (Charles River), were fed a standard diet (containing 135 mg kg$^{-1}$ vitamin E, V1534-300, ssniff Spezialdiäten) and underwent renal ischaemia–reperfusion injury as described previously[49]. In brief, bilateral renal pedicle clamping was performed via a midline abdominal incision for 36 min. Throughout the surgical procedure, the body temperature was maintained between 36 and 37 °C. After removal of the clamps, the abdomen was closed allowing restoration of blood flow as also visually observed. Sham-operated mice underwent the identical surgical procedures, except clamping of renal pedicles. All mice were killed 48 h after the reperfusion. All ischaemia–reperfusion experiments were performed in a double-blinded manner. MK4 (200 mg kg$^{-1}$ dissolved in corn oil) or vehicle was injected intraperitoneally 1 h before the onset of ischaemia. Serum creatinine and urea were measured in the Institute for clinical chemistry of the University Hospital Dresden (Germany). Kidney tissue damage was quantified by two researchers in a double-blind manner on a scale ranging from 0 (unaffected tissue) to 10 (severe organ damage). The following parameters were chosen as indicative of morphological damage to the kidney after ischaemia–reperfusion injury: brush border loss, red blood cell extravasation, tubule dilatation, tubule degeneration, tubule necrosis, and tubular cast formation. These parameters were evaluated on a scale of 0–10, which ranged from not present (0), mild (1–4), moderate (5 or 6), severe (7 or 8), to very severe (9 or 10). For the scoring system,

tissues were stained with periodic acid–Schiff (PAS), and the degree of morphological involvement in renal failure was determined using light microscopy.

## Generation of *Fsp1*⁻/⁻ mice

*Fsp1*⁻/⁻ mice (that is, *B6.129-Aifm2*^*tm1Marc*^*/Ieg*) were obtained from Infrafrontier (https://www.infrafrontier.eu; EM:05283). In these mice, exons 5 and 6 of the *Aifm2* (also known as *Fsp1*) gene were replaced by a *lacZ-neo* cassette. For genotyping PCR, following primers were used: 5′-GCCTGGTATTCACATTGGAA and 5′-GAGTGGATAAGAGTGACCTG for the wild-type allele; 5′-CCGCTTAAGCTAGCCATGGGTAATTC and 5′-GACAGTATCGGCCTCAGGAA for the KO allele.

## Warfarin treatment of mice

Sex- and age-matched littermates (20–30 g, aged 8–16 weeks) of *Fsp1*⁻/⁻ and *Fsp1*⁺/⁻ mice were used. Mice were orally administered with warfarin sodium through bottled drinking water (0.33 mg ml⁻¹ water) until completion of the study. This dose corresponds to a warfarin uptake of 50 mg kg⁻¹ per mouse for a 24-h feeding period, assuming water consumption is 15 ml per 100 g per 24 h. Prothrombin time was measured using CoaguChek Pro II (Roche Diagnostics), which has a reportable range of 9.6 to 96 s, at the timing of 60 h after the start of warfarin sodium administration and 12 h after the subcutaneous injection of 20 mg kg⁻¹ MK4 (dissolved in Miglyol 812) or vehicle. When the prothrombin time was above the detectable limit, the value was regarded as 96 s for the statistical analysis. For the measurement of MK4 and MK4 epoxide levels, plasma and liver tissues were collected 6 h after injection of MK4 (20 mg kg⁻¹, subcutaneous injection) from *Fsp1*⁺/⁻ and *Fsp1*⁻/⁻ mice treated with warfarin sodium. For the survival study, MK4 (10 mg kg day⁻¹, subcutaneous injection) or vehicle was administrated each day. All mice were monitored twice daily for survival. When animals reached the humane end point, they were immediately euthanized.

## Histology, immunohistochemistry and TUNEL staining

Tissues were fixed in 4% paraformaldehyde and embedded in paraffin. For immunohistochemistry, deparaffinized sections were immunolabeled using antibodies for anti-GPX4 (ab125066, Abcam), anti-4HNE (HNEJ-2, JaICA), anti-KIM-1 (AF1817, R&D) and anti-cleaved caspase-3 (9661, Cell Signaling). For anti-GPX4, KIM-1 and cleaved caspase-3 staining, the sections were heated for antigen retrieval in a microwave oven in 0.01 M citrate buffer pH 6.0 (for anti-GPX4 and KIM-1) or in Target Retrieval Solution (S1699, DAKO; for anti-cleaved caspase-3) for 20 min. After blocking with 5% FBS (for GPX4, KIM-1 and cleaved caspase-3) or 10% FBS (for 4HNE) in Tris-buffered saline, pH 7.4 containing 0.01% Tween-20 for 30 min, the sections were incubated with the primary antibodies (anti-GPX4 1:100; anti-4HNE 0.5 μg ml⁻¹; anti-KIM-1 1:200; and anti-cleaved caspase-3 1:100) overnight at 4 °C. After incubation with 0.3% $H_2O_2$ in methanol for 20 min, the sections were incubated with the following secondary antibodies for 30 min: biotinylated goat anti-rabbit IgG (1:250; BA-1000, Vector Laboratories) for anti-GPX4; biotinylated goat anti-mouse-IgG (1:200; BA-9200, Vector Laboratories) for anti-4HNE; and biotinylated donkey anti-goat-IgG diluted (1:500; 208000, Abcam) for anti-KIM-1, and then incubated with streptavidin–biotin peroxidase complex (VECTASTAIN Elite ABC system, Vector Laboratories). For anti-cleaved caspase-3 staining, Histofine Simple Stain MAX PO (R) Anti-Rabbit (Nichirei) was used as secondary antibody. The sections were visualized with nickel-enhanced 3,3′-diaminobenzidine (DAB, SK-4100, Vector Laboratories) for anti-GPX4 and KIM-1, or DAB and counterstaining with Mayer's Hematoxylin for anti-4HNE and cleaved caspase-3. TUNEL staining was performed using the ApopTag peroxidase in situ apoptosis detection kit (Millipore). To reduce false-positive signals, the TdT enzyme was diluted 1:16 in reaction buffer for preparation of the working solution. Gr-1⁺ cells were immunohistochemically stained on acetone-fixed frozen liver sections. Dried sections were

blocked with 10% goat serum for 1 h, and then incubated with anti-Gr-1-FITC antibody (0.5 mg ml⁻¹, 553127, BD Pharmingen) for 30 min at room temperature. The sections were treated with goat anti-Rat Alexa Fluor 488 IgG (H+L) (1:500, A-11006, Invitrogen) and DAPI (5 mg ml⁻¹) for visualization. Gr-1⁺ cells were counted per high-power field (HPF) (2,000× magnification; five HPF per slide). A blinded scientist received the slides randomly and performed all cell counting procedures.

## Quantification and statistical analysis

Statistical information for individual experiments can be found in the corresponding figure legends. Values are presented as mean ± s.d. Statistical comparisons between groups were analysed for significance by two-tailed Student's *t*-test, one-way ANOVA with Dunnett's post hoc test. Survival analysis was done according to the log-rank test. Results were considered significant at $P < 0.05$. Statistical analyses were conducted using GraphPad Prism 9 (GraphPad Software) and JMP15 (SAS Institute) software.

## Reporting summary

Further information on research design is available in the Nature Research Reporting Summary linked to this paper.

## Data availability

Epilipidomics data are available at MASSIVE (https://massive.ucsd.edu/) under accession number MSV000089489. Gel source images are shown in Supplementary Fig. 1. Other data are available from the corresponding author upon reasonable request. Source data are provided with this paper.

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

**Acknowledgements** We thank Y. Suhara for providing expert advice; A. Sekimoto for supporting the animal experiments; E. Bürkle for preparation of histology sections; K. Steiger for assistance with immunohistochemistry; R. Hoffmann for providing access to his laboratory; and P. Chambon for provision of *Alb-creER*^*T2*^ mice. This work was supported by funding from the Deutsche Forschungsgemeinschaft (DFG) (CO 291/7-1 and the Priority Program SPP 2306 [CO 291/9-1, CO 291/10-1]), the German Federal Ministry of Education and Research (BMBF), the VIP+ programme NEUROPROTEKT (03VP04260), and the European Research Council (ERC) under the European Union's Horizon 2020 research and innovation programme (grant agreement no. GA 884754) to M.C.; the DFG PR 1752/3-1 to B.P.; JSPS KAKENHI (20KK0363), Japan Heart Foundation/Bayer Yakuhin Research Grant Abroad, the Uehara Memorial Foundation, Watanabe foundation, Japan Foundation for Applied Enzymology, and Tohoku University Center for Gender Equality Promotion Support Project (TUMUG) to E.M.; JSPS KAKENHI (20K20604) and AMED grant (JP21zf0127001) to T.A.; Natural Sciences and Engineering Research Council and Canada Foundation for Innovation to D.A.P.; AMED-CREST grant (JP21gm0910013) and JSPS KAKENHI (20H00493) to K.Y.; German Federal Ministry of Education and Research (BMBF) within the framework of the e:Med research and funding concept for SysMedOS project to M.F.

**Author contributions** E.M., S.D., D.A.P. and M.C. conceived the study and wrote the manuscript. E.M. and T.N. performed in vitro experiments. J.I., M.A., and B.H. performed mass spectrometry analysis of vitamin K content. E.M., J.W. and B.P. performed animal experiments using transgenic mice. A.S.D.M. expressed and purified recombinant FSP1. P.N. and M.F. performed lipidomics and data interpretation. W.T., A.L., E.E. and E.K.G. performed and analysed the ischaemia–reperfusion injury experiments. E.M. and E.S. performed the animal experiment using warfarin treatment. Z.W. and O.Z. performed and interpreted data obtained by the RTA and FSP1 activity assays. K.Y. synthesized LipiRADICAL Green. R.F. generated the antibody against mouse FSP1. D.H. and A.M. performed the serum analysis of the mice. A.W., K.N. and T.A. contributed scientific insights and analysed the results. All authors read and agreed on the content of the paper.

**Funding** Open access funding provided by Helmholtz Zentrum München - Deutsches Forschungszentrum für Gesundheit und Umwelt (GmbH).

**Competing interests** M.C. and B.P. hold patents for some of the compounds described herein, and are co-founders and shareholders of ROSCUE Therapeutics GmbH. E.M. has filed a patent related to the treatment of ferroptosis-associated diseases with vitamin K. The other authors declare no competing interests.

**Additional information**
**Correspondence and requests for materials** should be addressed to Eikan Mishima or Marcus Conrad.

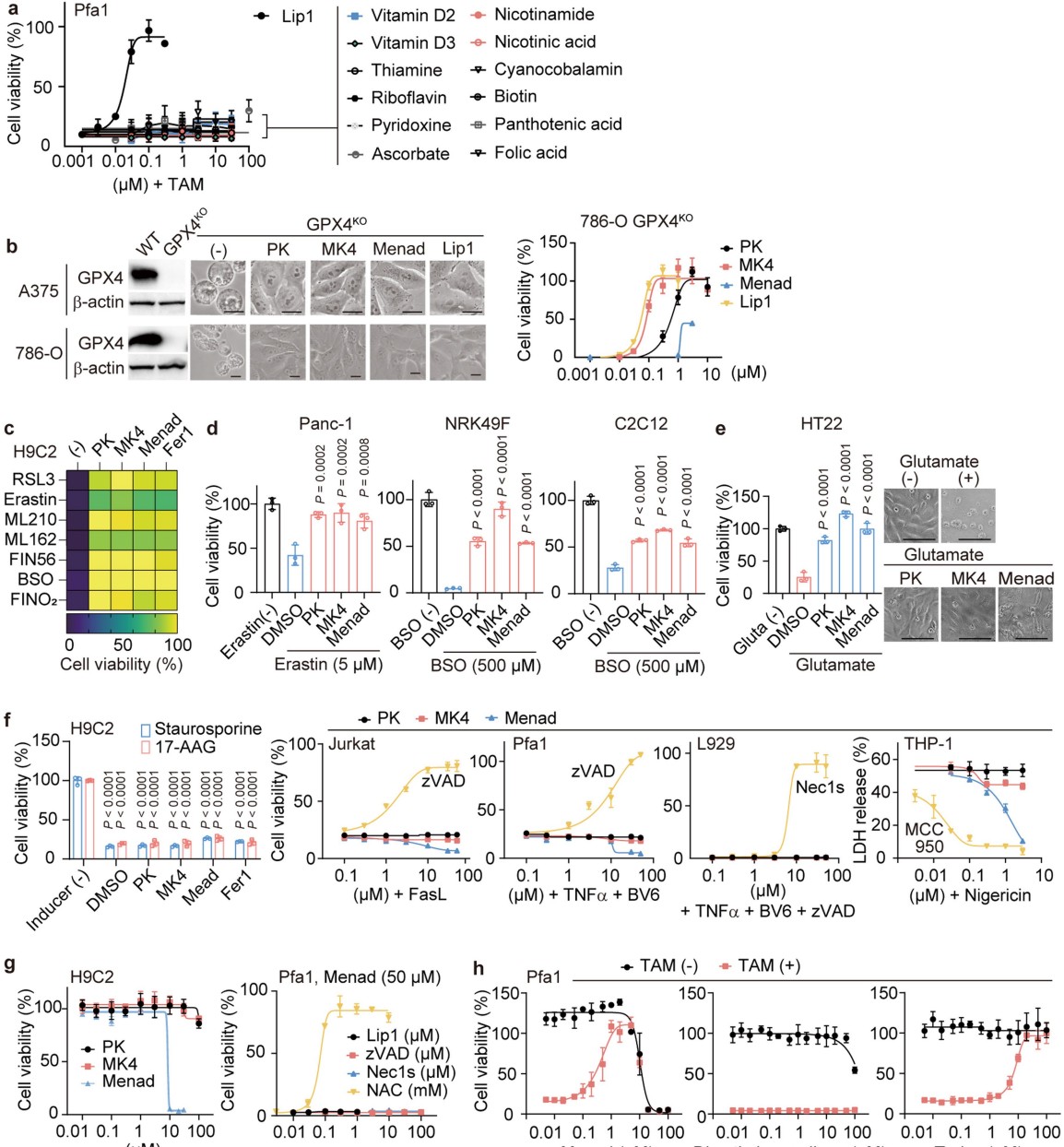

**Extended Data Fig. 1 | General anti-ferroptotic effects of Vitamin K in cells.**
**a**. Viability of TAM-induced *Gpx4*^KO Pfa1 cells treated with different vitamin compounds. Viability was assessed 72 h after addition of 4-OH-TAM. Lip1, Liproxstatin-1 (a ferroptosis inhibitor). **b**. The image of *GPX4*^KO A375 cells treated with indicated compounds 5 days after withdrawal of Lip1. The image and viability of *GPX4*^KO 786-O cells 4 days after withdrawal of Lip1. Scale, 10 μm. **c**. Heatmap showing viability of H9C2 cells treated with the following ferroptosis inducers: RSL3 (50 nM), erastin (2 μM), ML210 (0.3 μM), ML162 (0.3 μM), FIN56 (0.5 μM), BSO (100 μM) and FINO$_2$ (5 μM). PK (10 μM), MK4 (10 μM), Menad (3 μM) and Fer1 (0.3 μM, ferrostatin-1, a ferroptosis inhibitor) were added 1 h prior to the treatment of inducers. **d**. Anti-ferroptotic effect of vitamin K in Panc-1 (human pancreas cancer), NRK49F (rat kidney fibroblast), and C2C12 (mouse myoblast) cells. PK (10 μM), MK4 (10 μM) and Menad (3 μM) were used. **e**. Protective effect of vitamin K against glutamate-induced toxicity in HT-22 mouse hippocampal neuronal cells. Glutamate (4 mM), PK (10 μM), MK4 (10 μM) and Menad (3 μM) were used. Scale, 100 μm. **f**. The effect of

vitamin K on apoptosis, necroptosis and pyroptosis. Apoptosis was induced by staurosporine (50 nM) or 17-AAG (0.5 μM) in H9C2 cells; Fas ligand (30 ng/ml) in Jurkat cells; and mouse TNFα (10 ng/mL) and BV-6 (Smac mimetic, 400 nM) in Pfa1 cells. Necroptosis was induced by mouse TNFα (10 ng/mL), BV-6 (400 nM) and z-VAD-FMK (zVAD, pan-caspase inhibitor, 30 μM) in L929 cells. Pyroptosis was induced by nigericin (10 μM) in LPS-stimulated THP-1 cells. zVAD, Nec1s (necroptosis inhibitor) and MCC950 (NLRP3 inhibitor) were used as control. **g**. Evaluation of cellular toxicity of vitamin K. H9C2 cells were treated with indicated vitamin K concentrations for 48 h. Menadione (50 μM)-treated Pfa1 cells were incubated with the indicated cell death inhibitors for 24 h. N-acetyl-L-cysteine (NAC) prevented menadione-induced cell death. **h**. Viability of Pfa1 cells (TAM−) and 4-OH-TAM-induced Gpx4^KO Pfa1 cells (TAM+) treated with menadione, dimethylmenadione and Trolox. Viability was assessed 72 h after addition of 4-OH-TAM. Data are the mean ± s.d. of n = 3 (a-h). P values, one-way ANOVA (Dunnett's) (vs DMSO in d and e; vs inducer [-] in f).

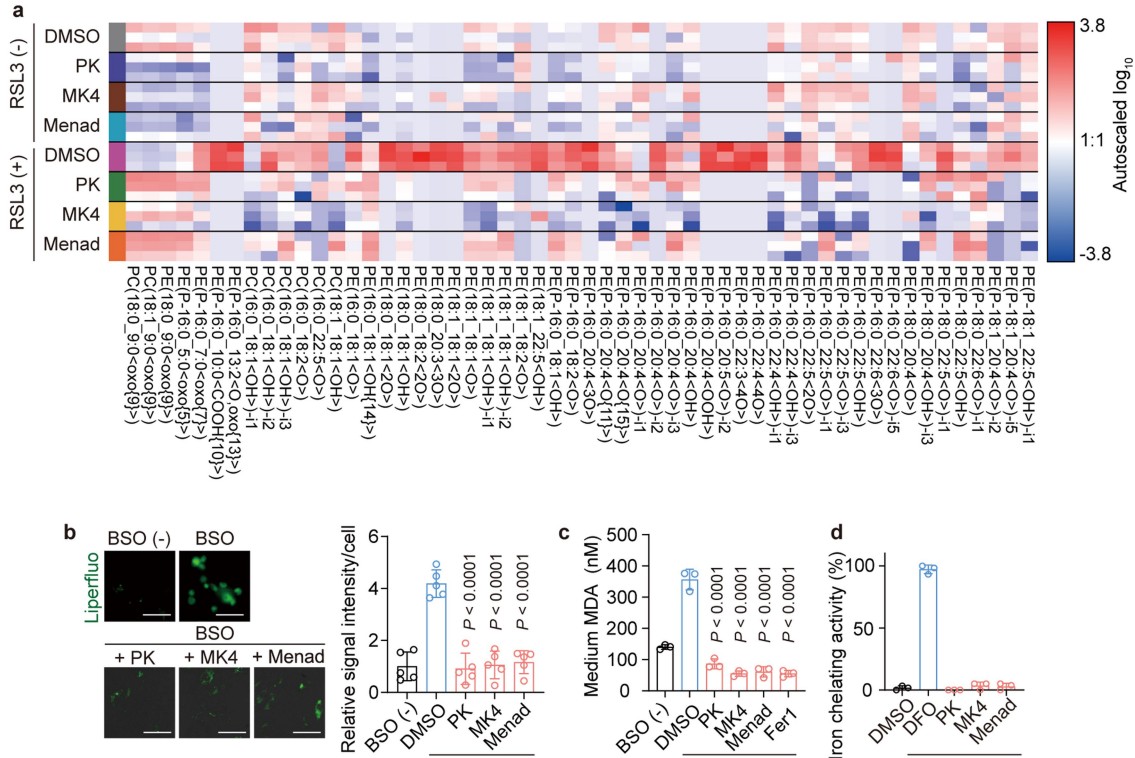

**Extended Data Fig. 2 | Vitamin K prevents lipid peroxidation. a**. Detailed epilipidomics data of Pfa1 cells shown in Fig. 1e. A full heatmap showing relative quantities of oxidized phospholipids in Pfa1 cells treated with RSL3 (0.5 μM) for 8 h. PK, MK4 and Menad (3 μM) were added 6 h before the addition of RSL3. Oxidized lipids were relatively quantified by LC-MS/MS. PE, phosphatidy lethanolamines; PC, phosphatidylcholines. Symbol i indicates isomeric lipids. Normalized peak areas of oxidized lipids showing significant regulation (ANOVA, adjusted p-value [FDR] cutoff: 0.05) were log transformed and

visualized using heatmaps. **b**. Imaging of intracellular lipid hydroperoxides detected by Liperfluo in BSO (100 μM, 40 h)-treated H9C2 cells. Scale, 50 μm. The relative signal intensities were measured. n = 5. **c**. Malondialdehyde (MDA) levels in medium after BSO (100 μM, 60 h) treatment of H9C2 cells. n = 3. **d**. Evaluation of iron chelating activity of vitamin K (each 100 μM). Deferoxamine (DFO) is an iron chelator. n = 3. Data are the mean ± s.d. P values, one-way ANOVA (Dunnett's, vs DMSO) (b, c).

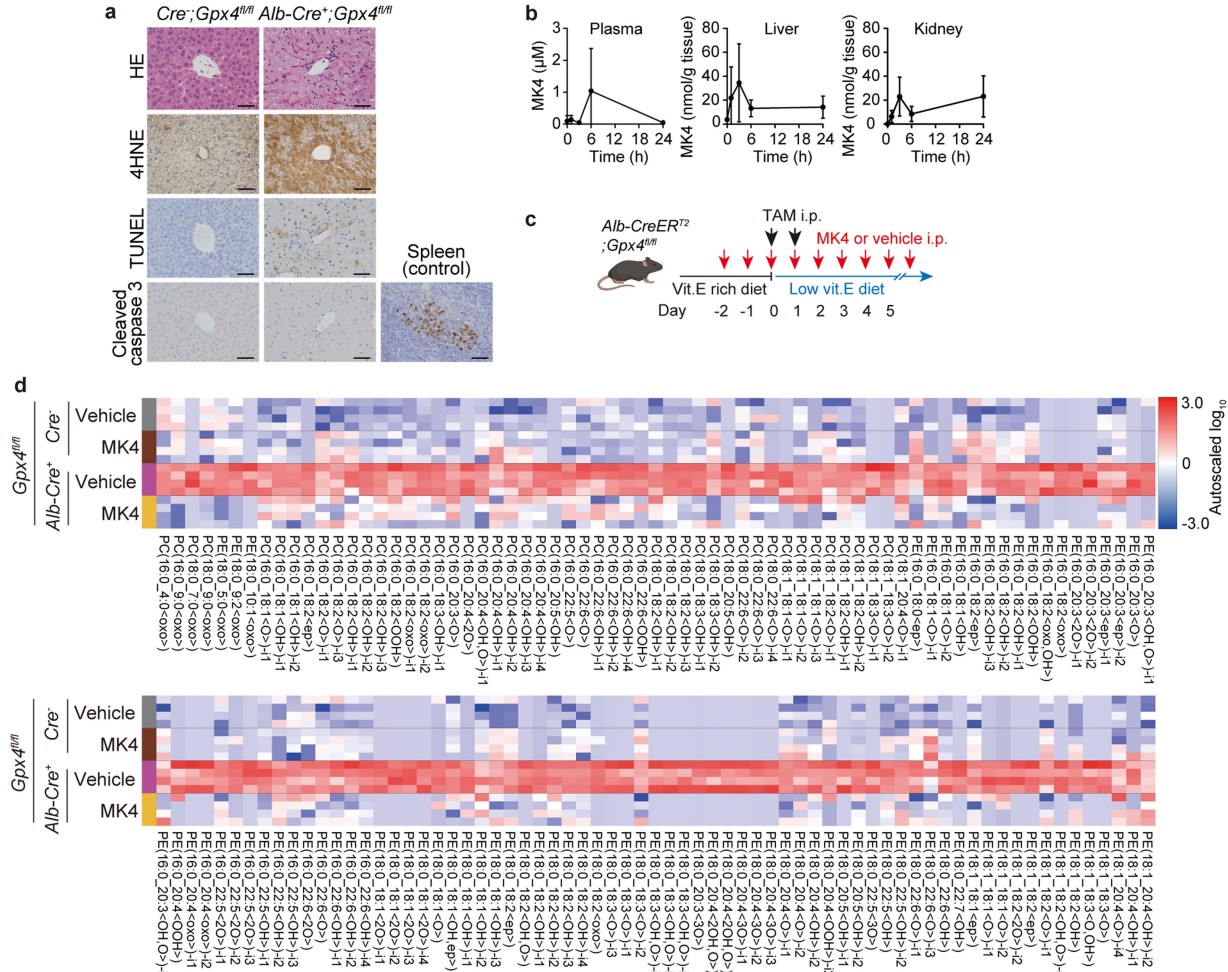

**Extended Data Fig. 3 | MK4 is tissue-protective in hepatocyte-specific *Gpx4* KO mice. a**. Evaluation of the liver in *Cre⁻;Gpx4^{fl/fl}* mice and *Alb-CreER^{T2};Gpx4^{fl/fl}* mice after TAM injection. The tissue samples of *Alb-CreER^{T2};Gpx4^{fl/fl}* mice were collected at the time of humane endpoint after TAM injection. Macroscopic image, H&E staining, immunohistochemistry for GPX4, 4-Hydroxynonenal (4HNE) and cleaved caspase-3 and TUNEL staining are shown. Scale, 50 μm. Livers of induced *Alb-CreER^{T2};Gpx4^{fl/fl}* mice diffusely present TUNEL-positive and cleaved caspase-3-negative sign of cell death. **b**. Time course of the level of MK4 in plasma, kidney and liver after intraperitoneal bolus injection of MK4 (200 mg/kg) to *Gpx4^{fl/fl}* mice. n = 3 in each time point. **c**. Experimental scheme of the studies of *Alb-CreER^{T2};Gpx4^{fl/fl}* mice. The illustration was created using BioRender.com. **d**. Detailed epilipidomics data of the liver shown in Fig. 1j. Heatmap showing relative quantities of oxidized phospholipids in the liver collected 7 days after TAM injection. oxPC, oxidized phosphatidylcholines; oxPE, oxidized phosphatidylethanolamines. Symbol i indicates isomeric lipids. Normalized peak areas of oxidized lipids showing significant regulation (ANOVA, adjusted p-value [FDR] cutoff: 0.05) were log transformed, auto-scaled and visualized using heatmaps.

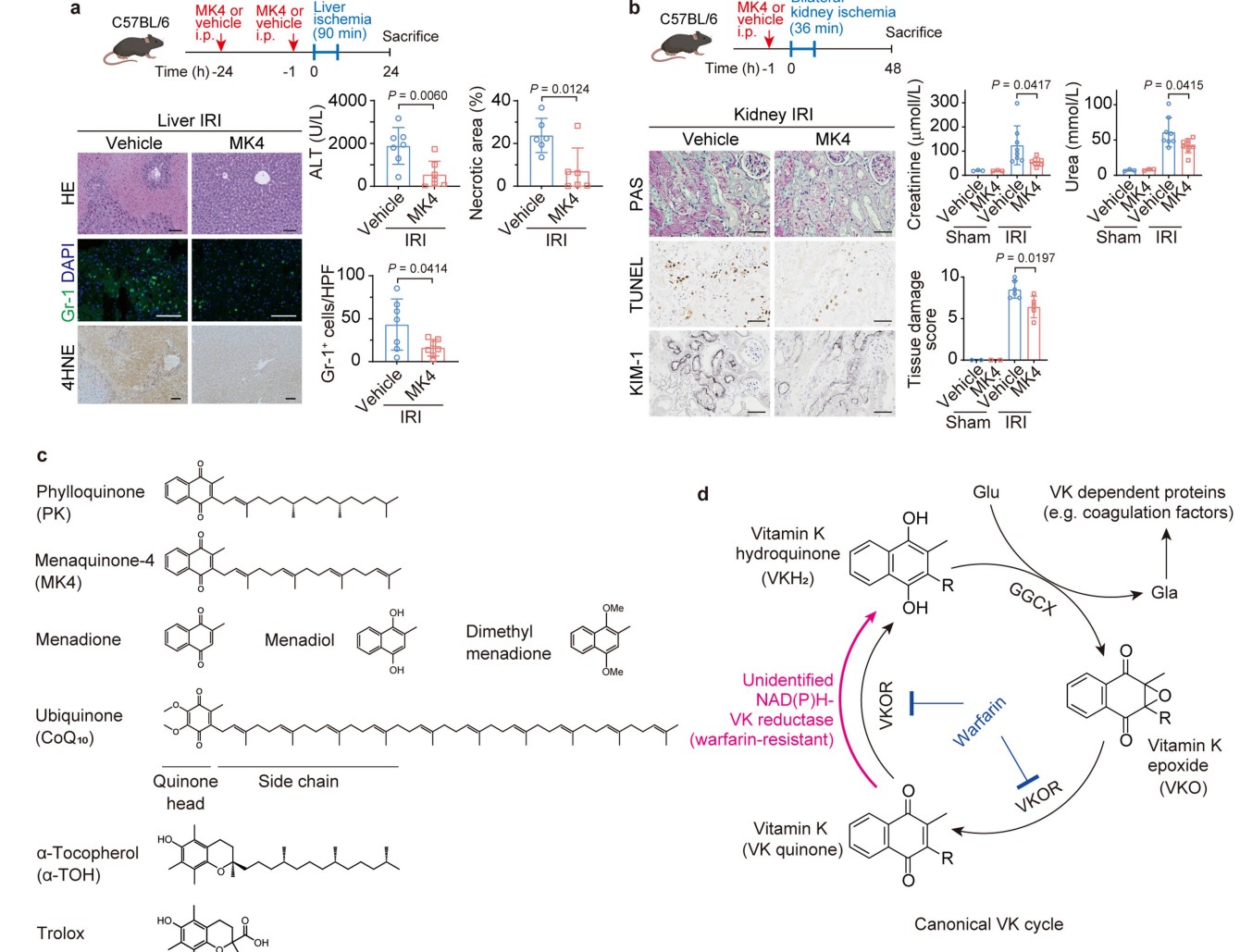

**Extended Data Fig. 4 | Tissue protective effect of MK4 in ischemia-reperfusion injury, chemical structure of vitamin K and the canonical vitamin K cycle. a**. Effect of prophylactic MK4 treatment (200 mg/kg, i.p., 24 h and 1 h before IRI) on liver ischemia-reperfusion injury (IRI) in C57BL/6 mice. Liver histology (H&E staining and immunohistochemistry for Gr-1 and 4HNE), serum ALT level, and quantification of necrotic area and Gr-1+ cells are shown. n = 7 (ALT and %Gr-1+ cells) and n = 6 (necrotic area). Scale, 100 μm. Data are mean ± s.d. two-tailed t-test. The illustration of the mouse was created using BioRender.com. **b**. Effect of prophylactic MK4 treatment (200 mg/kg, i.p., 1 h before IRI) on kidney IRI in C57BL/6 mice. Kidney histology (PAS and TUNEL staining, and immunohistochemistry for kidney injury molecule-1 [KIM-1]), serum creatinine and urea, and tissue damage score are shown. Scale, 50 μm. n = 3 sham, n = 8 IRI (creatinine and urea). n = 2 sham, n = 6 vehicle IRI, n = 5 MK4-IRI (tissue score). Data are mean ± s.d. One-way ANOVA (Dunnett's). **c**. Chemical structures of PK, MK4, menadione, menadiol, dimethylmenadione, ubiquinone (Coenzyme Q10, CoQ10), α-tocopherol and Trolox. **d**. Scheme of the canonical vitamin K cycle. GGCX, γ-glutamyl carboxylase; VKOR, vitamin K epoxide reductase; Glu, glutamate; Gla, γ-carboxyglutamate.

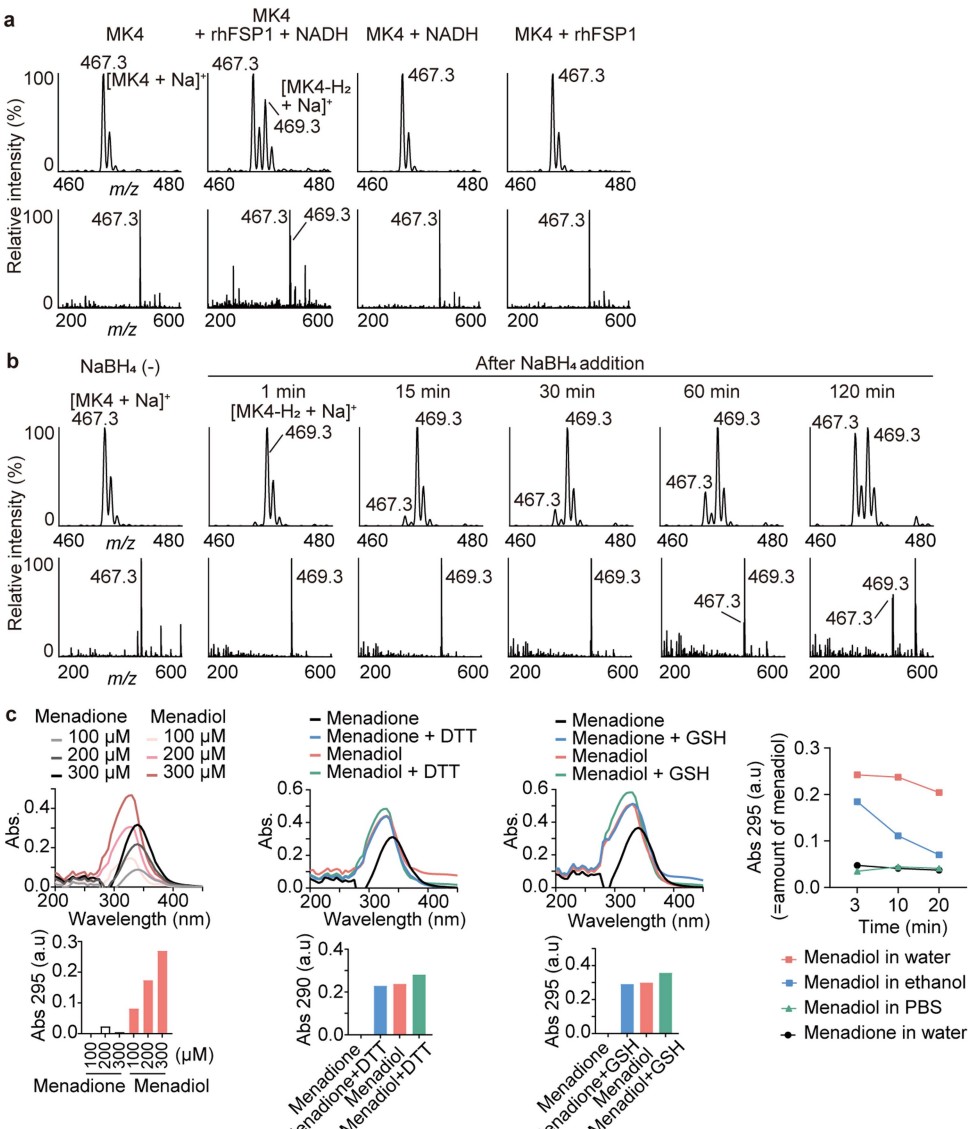

**Extended Data Fig. 5 | Chemical reduction of vitamin K to vitamin K hydroquinone and its autoxidation. a**. MS spectra of MK4 incubated with recombinant human FSP1 (rhFSP1) and/or NADH detected by MS. The data of MK4 only and MK4 + rhFSP1 + NADH (*m/z* 460-480) are used in Fig. 2b. **b**. MS spectra of MK4 before and after the addition of sodium borohydride (NaBH₄). MK4-hydroquinone (MK4-H₂) reduced by NaBH₄ was rapidly, non-catalytically re-oxidized to MK4 under atmospheric conditions. **c**. *(Left)* Absorbance (Abs) spectrum of menadione and menadiol dissolved in water. The value of Abs 290 or 295 nm indicates the amount of menadiol. *(Middle)* Absorbance spectrum

and the value of Abs 295 nm of menadione and menadiol (300 μM) incubated with or without dithiothreitol (DTT, 1 mM) or glutathione (GSH, 10 mM). Menadione was non-enzymatically reduced by GSH as well as by DTT. *(Right)* The time course change of Abs 295 nm of the solution of menadiol (dissolved in water, ethanol or PBS) and menadione (dissolved in water), indicating autoxidation of menadiol to menadione. Without reducing agent, such as DTT, menadiol quickly oxidized to menadione under atmospheric conditions, especially in PBS and ethanol.

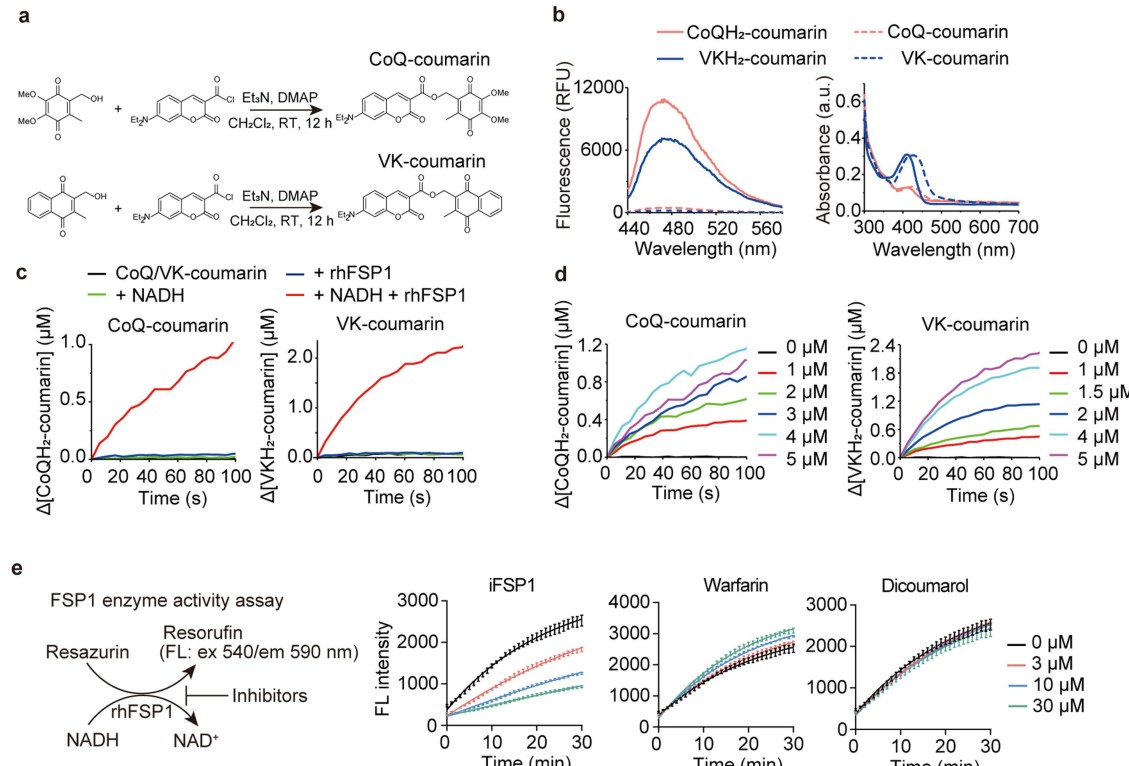

**Extended Data Fig. 6 | Characteristics of the coumarin conjugates, and effect of inhibitors on FSP1 enzymatic activity. a**. The scheme of synthesis of the VK/CoQ-coumarin conjugates. **b**. Fluorescence spectra ($\lambda_{ex}$ = 415 nm) and absorption spectra of 10 μM coumarin conjugates of the VK-coumarin and CoQ-coumarin in TBS buffer (pH 7.4) before and after the reduction with an excess of sodium metabisulfite (250 μM). **c**. Control experiments for the reduction of CoQ-coumarin and VK-coumarin (5 μM) by rhFSP1 (6 nM) and NADH (200 μM) in TBS buffer pH 7.4 at 37 °C. **d**. Representative kinetic traces

for reduction of CoQ-coumarin and VK-coumarin by rhFSP1 (6 nM) in the presence of FAD (50 nM) in TBS buffer at pH 7.4 in the presence of NADH (200 μM) at 37 °C. **e**. Scheme of FSP1 enzyme activity assay. Resazurin (100 μM), a substrate of FSP1, is reduced to resorufin by incubation with rhFSP1 (150 nM) and NADH (200 μM). The amount of resorufin evaluated by fluorescent intensity (ex 540/em 590 nm) indicates FSP1 enzymatic activity. The inhibitory effect of iFSP1, warfarin and dicoumarol on FSP1 enzymatic activity are shown. Data are the mean ± s.d. of n = 3 (e).

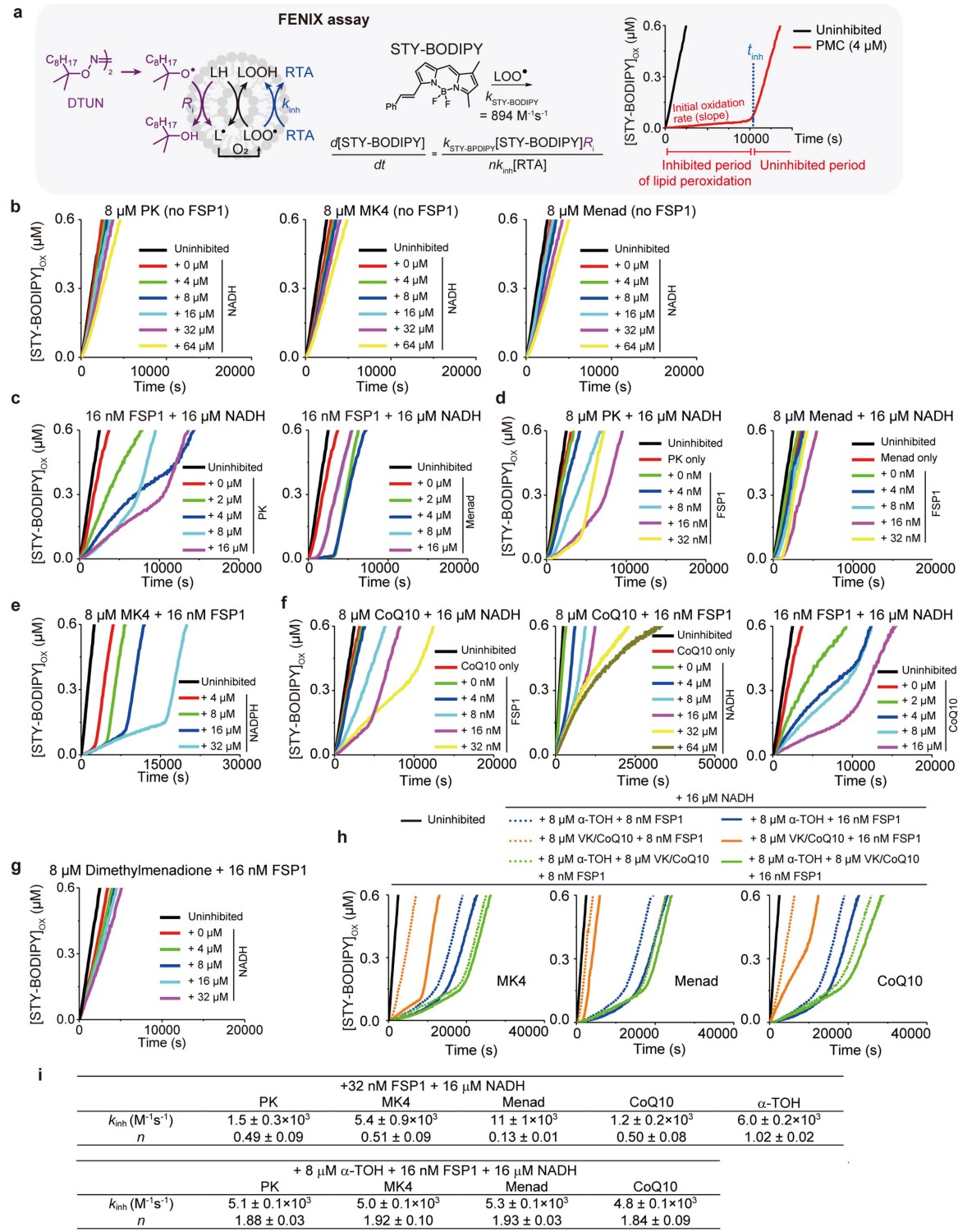

**Extended Data Fig. 7** | See next page for caption.

**Extended Data Fig. 7 | Evaluation of lipid peroxidation-inhibiting activity of vitamin K by FENIX assay. a**. The FENIX assay to determine lipid radical trapping antioxidant (RTA) activity. DTUN, a lipophilic radical initiator, decomposes to form lipophilic alkoxyl radicals at a constant rate $R_i$ to initiate lipid peroxidation in liposomal egg phosphatidylcholine (PC). Reaction progress is monitored by competitive consumption of STY-BODIPY (ex 488/em 518 nm) and the rate is used to calculate the inhibition rate constant ($k_{inh}$) for the reaction of an RTA with lipid peroxyl radicals (LOO•). A representative control reaction of FENIX showing PMC-inhibited co-autoxidations of STY-BODIPY (1 µM) and egg PC liposomes (1 mM) in TBS (pH 7.4) at 37 °C initiated by DTUN (200 µM). The inhibited period ($t_{inh}$) is defined by the intersection of the lines of best fit to the inhibited and uninhibited phases. PMC; 2,2,5,7,8-pentamethyl-6-chromanol, a truncated analog of vitamin E. **b**. The inhibition of lipid peroxidation by PK, MK4 or menadione (8 µM) as a function of NADH (0-64 µM) in the absence of FSP1. **c**. The inhibition of lipid peroxidation in the presence of FSP1 (16 nM) and NADH (16 µM) as a function of PK or menadione (0-16 µM). **d**. The inhibition of lipid peroxidation by PK or menadione (8 µM) with NADH (16 µM) as a function of FSP1 concentration (0-32 nM). **e**. The inhibition of lipid peroxidation by MK4 (8 µM) in the presence of FSP1 (16 nM) and varying NADPH concentrations (0-32 µM). **f**. The inhibition

of lipid peroxidation by coenzyme Q10 (CoQ, 8 µM) with NADH (16 µM) as a function of FSP1 concentration (0-32 nM) *(left)*; by CoQ10 (8 µM) in the presence of FSP1 (16 nM) and varying NADH (0-64 µM) *(middle)*; and by CoQ10 (0-16 µM) in the presence of FSP1 (16 nM) and NADH (16 µM) *(right)*. **g**. The inhibition of lipid peroxidation by dimethylmenadione (8 µM) in the presence of FSP1 (16 nM) and varying NADPH concentrations (0-32 µM). **h**. The inhibition of lipid peroxidation by α-tocopherol (α-TOH, 8 µM) in the presence of FSP1 (8 or 16 nM), NADH (16 µM) and MK4, menadione or CoQ10 (8 µM). **i**. Kinetic parameters determined for the vitamin K derivatives and CoQ10 in the presence of FSP1 (32 nM) and NADH (16 µM). Inhibition rate constants ($k_{inh}$) were determined assuming stoichiometries *(n)* = 2, which corresponds to the maximum amount of hydroquinone being formed and reacting with two peroxyl radical equivalents. It must be noted that these values are underestimates of the rate constant for reaction of the hydroquinone with peroxyl radicals due to the competing autoxidation of the hydroquinones, which leads to far lower stoichiometries. These are reported alongside the $k_{inh}$ values, and clearly reflect the wasting of reducing equivalents by the system. Data shown are representative trials from at least three independent measurements (b-h).

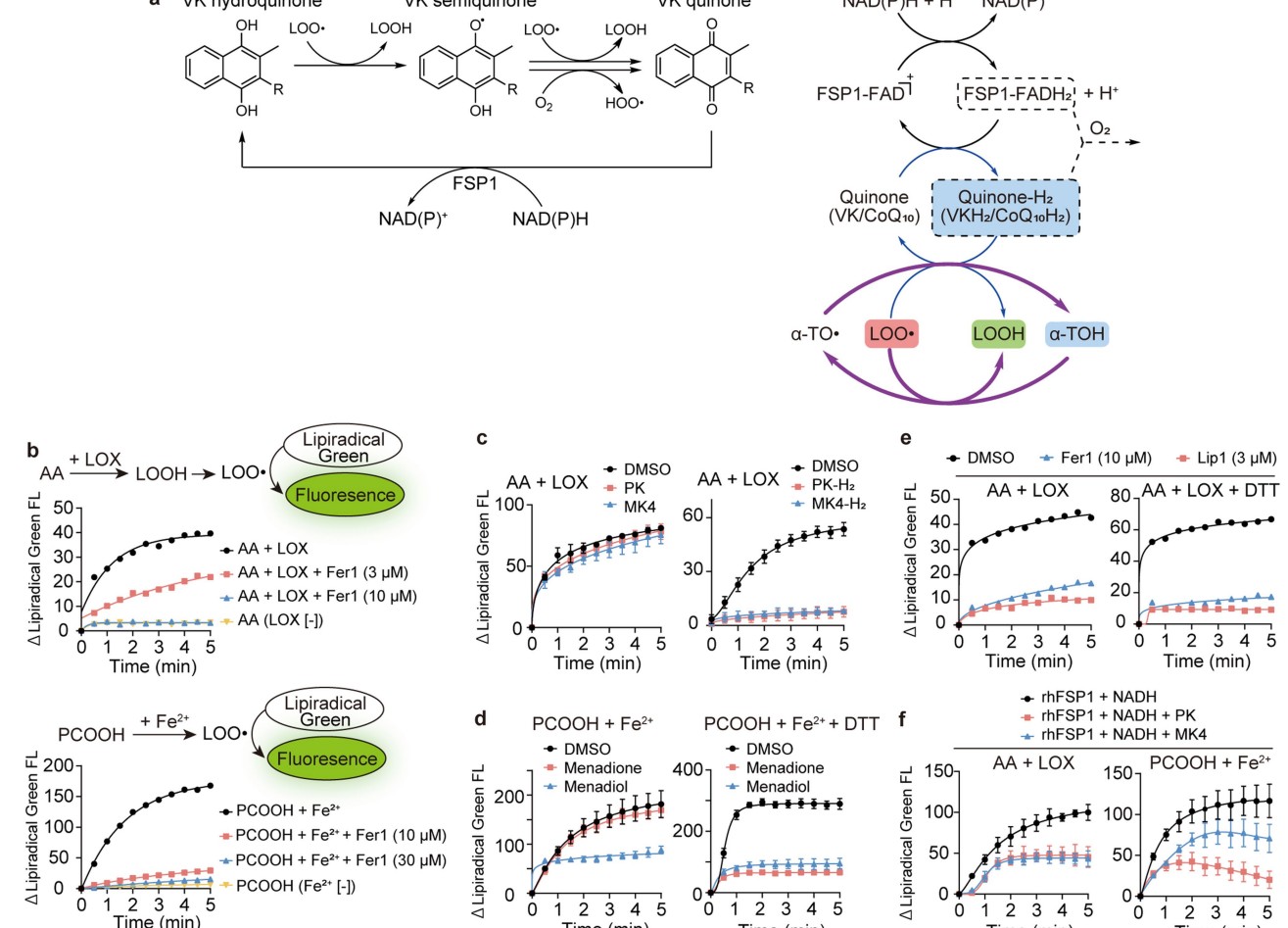

**Extended Data Fig. 8 | RTA reaction of vitamin K and LipiRADICAL Green assay. a**. (*Left*) Scheme of the redox reaction of vitamin K (VK) trapping lipid radicals. Oxidization of VK hydroquinone by trapping lipid radicals generates VK quinone, which is enzymatically reduced to VK hydroquinone by FSP1 consuming NAD(P)H. (*Right*) Diagram of the possible regeneration of α-tocopherol by CoQ10/VK-hydroquinone. α-TOH, α-tocopherol; α-TO•, α-tocopheroxyl radical; VKH$_2$, vitamin K-hydroquinone; LOOH, lipid hydroperoxide; LOO •, lipid peroxyl radical. **b**. Experimental scheme for detection of lipid radicals using LipiRADICAL Green fluorescence probe with the combination of arachidonic acid (AA) and lipoxygenase (LOX), or phosphatidylcholine hydroperoxide (PCOOH) and ferrous ion (Fe$^{2+}$). The increased value of fluorescence intensity (ex 470/em 530 nm) indicates lipid-derived radicals. Fer1, ferrostatin-1; FL, fluorescence intensity.

**c**–**f**. Lipiradical Green assay. **c**. Scavenging activities of 100 μM of PK, MK4, and their hydroquinone forms (PK-H$_2$ and MK4-H$_2$) chemically reduced by sodium borohydride evaluated using the AA + LOX system. **d**. Scavenging activities of menadione and menadiol (100 μM) with or without dithiothreitol (DTT, 1 mM) toward lipid radicals. Lipid radicals were generated from PCOOH and Fe$^{2+}$ in water. **e**. Scavenging activities of Lip1 and Fer1 with or without DTT (1 mM) toward lipid radicals using the AA + LOX system. DTT did not affect to the scavenging activity of Fer1 or Lip1 unlike VK. **f**. Scavenging property of PK and MK4 incubated with recombinant human FSP1 (rhFSP1) and NADH against lipid radicals generated by AA + LOX (PK and MK4, 100 μM) or PCOOH + Fe$^{2+}$ (PK and MK4, 300 μM) system. Data are mean ± s.d. of n = 3 (c, d and f). Data are representative of three independent experiments (c-f).

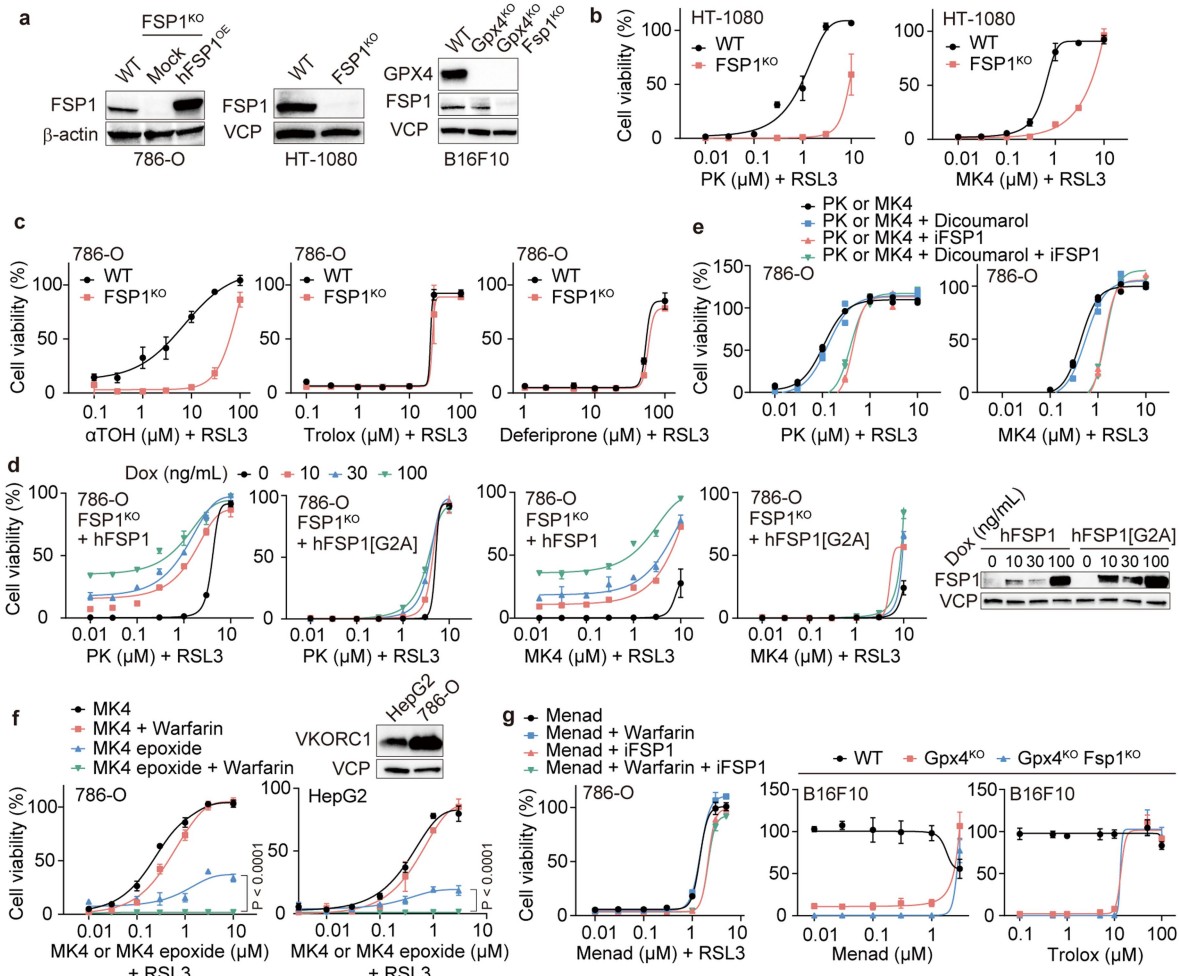

**Extended Data Fig. 9 | Effect of genetic deletion and pharmacological inhibition of FSP1 on the anti-ferroptotic action of vitamin K.**
**a**. Immunoblotting of *FSP1*[KO] 786-O and HT1080 cells, and *Gpx4*[KO] and *Gpx4*[KO]/*Fsp1*[KO] B16F10 cells. hFSP1[OE], overexpression of hFSP1. **b**. Protective effect of PK and MK4 against RSL3 (1 μM)-induced ferroptosis in wild type and *FSP1*[KO] HT-1080 cells. **c**. Protective effects of α-tocopherol (α-TOH), Trolox and deferiprone (an iron chelator) against RSL3 (1.5 μM)-induced ferroptosis in wild type and *FSP1*[KO] 786-O cells. **d**. Effect of doxycycline (Dox)-inducible expression of hFSP1 or hFSP1(G2A) variant in *FSP1*[KO] 786-O cells. Cells were treated with RSL3 (1 μM) and indicated concentrations of PK or MK4. Dox-induced hFSP1 expression was confirmed by immunoblotting. **e**. After pre-treatment of iFSP1 (10 μM) and/or dicoumarol (20 μM), 786-O cells were treated with RSL3

(0.5 μM) and indicated concentrations of PK and MK4. **f**. Effect of warfarin on the anti-ferroptotic effect of MK4 and MK4 epoxide in HepG2 and 786-O cells after pretreatment with warfarin (5 μM). Cells were treated with RSL3 (1 μM) and indicated concentrations of MK4 or MK4 epoxide. Expression of VKORC1 in HepG2 and 786-O cells was confirmed by immunoblotting. *P* values; one-way ANOVA (Dunnett's). **g**. (*Left*) After pretreatment of iFSP1 (10 μM) and/or warfarin (5 μM), 786-O cells were treated with RSL3 (1 μM) and indicated concentrations of menadione. (*Right*) Effect of menadione and Trolox on viability in Gpx4[KO] and Gpx4[KO]/FSP1[KO] B16F10 cells. Viability was assessed 4 days after withdrawal of Lip1. Viability of the cells treated with Lip1 (1 μM) was taken as 100%. Data are the mean ± s.d. of n = 3 (b-d, f and g) and n = 2 (e).

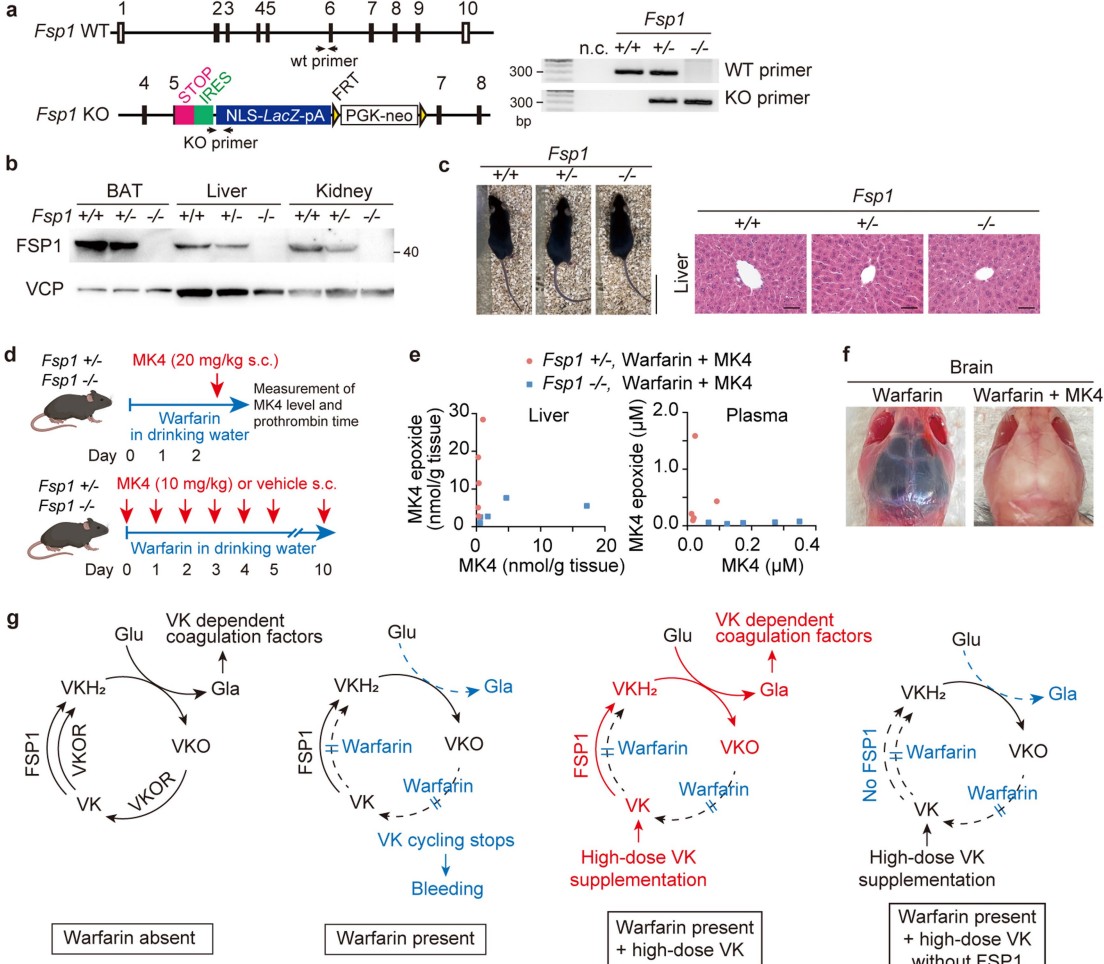

**Extended Data Fig. 10 | Characteristics of *Fsp1* KO mice. a.** Schematic maps of the wild type (WT) and *Fsp1* KO allele, and genotyping PCR of *Fsp1*$^{+/+}$, *Fsp1*$^{+/-}$ and *Fsp1*$^{-/-}$ mice. **b.** Immunoblotting evaluating FSP1 expression in the brown adipose tissue (BAT), liver and kidney of *Fsp1*$^{+/+}$, *Fsp1*$^{+/-}$ and *Fsp1*$^{-/-}$ mice. **c.** Images of 8-week-old male of *Fsp1*$^{+/+}$, *Fsp1*$^{+/-}$ and *Fsp1*$^{-/-}$ mice. Scale, 5 cm. H&E staining of the liver. Scale, 100 μm. *Fsp1*$^{-/-}$ mice showed normal growth and normal liver histology. **d.** Study design for evaluating the level of MK4/MK4 epoxide, prothrombin time and survival rate of *Fsp1*$^{+/-}$ and *Fsp1*$^{-/-}$ mice treated with high-dose warfarin (0.33 mg/ mL in drinking water). The level of MK4/MK4 epoxide and prothrombin time were measured 6 and 12 h after MK4 injection (20 mg/kg, s.c.), respectively. For the study evaluating survival rate, MK4

(10 mg/kg/day, s.c.) or vehicle was daily injected. The illustration was created using BioRender.com. **e.** The level of MK4 and MK4 epoxide in the liver and plasma of *Fsp1*$^{+/-}$ and *Fsp1*$^{-/-}$ mice treated with high-dose warfarin with MK4. n = 5. **f.** Representative brain image of a mouse treated with high-dose warfarin for 5 days, showing massive cerebral hemorrhage. The mouse cotreated with warfarin and MK4 (10 mg/kg/day, s.c.) did not show cerebral hemorrhage. **g.** Diagrams of vitamin K cycle with or without high-dose warfarin treatment in the presence or absence of FSP1 expression. VK, vitamin K quinone; VKH$_2$, vitamin K hydroquinone; VKO, vitamin K epoxide; VKOR, vitamin K epoxide reductase; Glu, glutamate; Gla, γ-carboxyglutamate.

# Reporting Summary

## Statistics

For all statistical analyses, confirm that the following items are present in the figure legend, table legend, main text, or Methods section.

| n/a | Confirmed | |
|---|---|---|
| ☐ | ☒ | The exact sample size (*n*) for each experimental group/condition, given as a discrete number and unit of measurement |
| ☐ | ☒ | A statement on whether measurements were taken from distinct samples or whether the same sample was measured repeatedly |
| ☐ | ☒ | The statistical test(s) used AND whether they are one- or two-sided *Only common tests should be described solely by name; describe more complex techniques in the Methods section.* |
| ☒ | ☐ | A description of all covariates tested |
| ☐ | ☒ | A description of any assumptions or corrections, such as tests of normality and adjustment for multiple comparisons |
| ☐ | ☒ | A full description of the statistical parameters including central tendency (e.g. means) or other basic estimates (e.g. regression coefficient) AND variation (e.g. standard deviation) or associated estimates of uncertainty (e.g. confidence intervals) |
| ☐ | ☒ | For null hypothesis testing, the test statistic (e.g. *F*, *t*, *r*) with confidence intervals, effect sizes, degrees of freedom and *P* value noted *Give P values as exact values whenever suitable.* |
| ☒ | ☐ | For Bayesian analysis, information on the choice of priors and Markov chain Monte Carlo settings |
| ☒ | ☐ | For hierarchical and complex designs, identification of the appropriate level for tests and full reporting of outcomes |
| ☒ | ☐ | Estimates of effect sizes (e.g. Cohen's *d*, Pearson's *r*), indicating how they were calculated |

*Our web collection on statistics for biologists contains articles on many of the points above.*

## Software and code

Policy information about availability of computer code

| | |
|---|---|
| Data collection | CytExpert v2.4 (Beckman Coulter, ), Image Lab v6.0 (Biorad), SoftMax Pro v7 (Molecular Devices), Eve v1.8.2 (Nanolive) |
| Data analysis | GraphPad Prism v9 (GraphPad Software), JMP v15 (SAS Institute Inc.), Flow Jo v10 software (Treestar, Inc), Skyline v21.1 (PMID 31984744), MetaboAnalyst online platform v5.0 (PMID 31756036), Genesis v1.8.1 (Bioinformatics TU-Graz), LipidLynxX system 0.9.24 (doi:10.1101/2020.04.09.033894), Image J v1.53 (NIH), ZEISS Axio Vision software AxioVs v4.9 (Carl Zeiss), ForeCyt software v8 (Sartorius). |

For manuscripts utilizing custom algorithms or software that are central to the research but not yet described in published literature, software must be made available to editors and reviewers. We strongly encourage code deposition in a community repository (e.g. GitHub). See the Nature Portfolio guidelines for submitting code & software for further information.

## Data

Policy information about availability of data

All manuscripts must include a data availability statement. This statement should provide the following information, where applicable:

- Accession codes, unique identifiers, or web links for publicly available datasets
- A description of any restrictions on data availability
- For clinical datasets or third party data, please ensure that the statement adheres to our policy

All data are available within the Article and the Supplementary Information, and from the corresponding author on reasonable request. Gel source images are shown in Supplementary Fig. 1. All source data are provided with this paper. Epilipidomics data are available at MASSIVE (https://massive.ucsd.edu/) under accession number MSV000089489.

# Field-specific reporting

Please select the one below that is the best fit for your research. If you are not sure, read the appropriate sections before making your selection.

☒ Life sciences ☐ Behavioural & social sciences ☐ Ecological, evolutionary & environmental sciences

For a reference copy of the document with all sections, see nature.com/documents/nr-reporting-summary-flat.pdf

# Life sciences study design

All studies must disclose on these points even when the disclosure is negative.

| | |
|---|---|
| Sample size | For in vitro experiments, sample sizes were determined based on previous similar studies that have given statistically significant results (PMID: 31634899). The number of animals studied per treatment group was determined based on our preliminary data and previous similar studies that have given statistically significant results (PMID: 34285231 and 25402683), and respects the limited use of animal models in line with the 3R recommendations: Replacement, Reduction, Refinement. |
| Data exclusions | No data exclusions. |
| Replication | The experimental findings were reproduced as validated by at least three independent experiment in Fig1a-d and f, Fig2a-f, Fig 4a, Extended Fig1b, Extended Fig 5a-c, 6a-e, 7b-h, 9e and 10b; and at least two independent experiments in Fig 1g-i, Fig 3a-d, Fig4c, Extended Fig 1a-h, Extended Fig 8b-f, Fig 9b-g. |
| Randomization | For animal studies, mice were randomized into separate cages. Sex-matched littermates were used and experiments were intended to test a single variable. For in vitro studies, samples were randomized, when possible, prior running. |
| Blinding | For animal study, mice were given a number prior to data collection and analysis. Data was collected and analyzed blindly. For in vitro experiments, investigators were not blinded, as standard in this manner of study, which contained multiple steps requiring distinct operations for accuracy and precision precluding blinding to experimental variables. |

# Reporting for specific materials, systems and methods

We require information from authors about some types of materials, experimental systems and methods used in many studies. Here, indicate whether each material, system or method listed is relevant to your study. If you are not sure if a list item applies to your research, read the appropriate section before selecting a response.

## Materials & experimental systems

| n/a | Involved in the study |
|---|---|
| ☐ | ☒ Antibodies |
| ☐ | ☒ Eukaryotic cell lines |
| ☒ | ☐ Palaeontology and archaeology |
| ☐ | ☒ Animals and other organisms |
| ☒ | ☐ Human research participants |
| ☒ | ☐ Clinical data |
| ☒ | ☐ Dual use research of concern |

## Methods

| n/a | Involved in the study |
|---|---|
| ☒ | ☐ ChIP-seq |
| ☐ | ☒ Flow cytometry |
| ☒ | ☐ MRI-based neuroimaging |

## Antibodies

| | |
|---|---|
| Antibodies used | GPX4 (1:1000 for WB, 1:100 for IHC, ab125066, Abcam), 4-HNE (1 µg/mL for WB and 0.5 µg/mL for IHC, MHN-20P, JaICA), human FSP1 (1:1000, sc-377120, Santa Cruz Biotechnology), mouse FSP1 (1:100, clone AIFM2 1A1 rat IgG2a, and 1:1 clone AIFM2 14D7 IgG2b supernatant of hybridoma, developed in-house), VKORC1 (1:1000, ab206656, Abcam), GGCX (1:1000, ab197982, Abcam), β-actin-HRP (1:5000, A3854, Sigma-Aldrich), valosin containing protein (VCP, 1:10000, ab11433, Abcam), anti-KIM-1 (1:200, AF1817, R&D), anti-cleaved caspase-3 (1:100, 9661, Cell Signaling), anti-Gr1-FITC antibody (0.5mg/mL, 553127, BD Pharmingen), goat anti-Rat Alexa Fluor 488 IgG (H+L) (1:500, A-11006, Invitrogen), biotinylated goat anti-rabbit IgG (1:250; BA-1000, Vector Laboratories), biotinylated goat anti-mouse-IgG (1:200; BA-9200, Vector Laboratories), biotinylated donkey anti-goat-IgG diluted (1:500; 208000, Abcam), Histofine Simple Stain MAX PO (R) Anti-Rabbit (414141F, Nichirei), and anti-His antibody clone 3D5 (prepared in-house as described in a previous publication; PMID 8994661). |
| Validation | GPX4 antibody (ab125066, Abcam) and VCP (Abcam) were validated for WB using mouse and human cells samples in a previous publication (PMID: 31634899). <br> Human FSP1 (sc-377120) was validated for WB using human cells samples in a previous publication (PMID: 31634899). <br> 4-HNE (MHN-20P) was validated for IHC and WB using mouse samples on the manufacturer's website (https://www.jaica.com/e/products_lipid_4hne_ab.html). <br> VKORC1 (ab206656) for WB using human cell samples was validated on the manufacturer's website (https://www.abcam.com/vkorc1-antibody-epr20245-ab206656.html). |

GGCX (ab197982) for WB using human cell samples was validated on the manufacturer's website (https://www.abcam.com/ggcx-antibody-ab197982.html).
β-actin-HRP (A3854) was validated for WB using mouse and human cell samples on the manufacturer's website (https://www.sigmaaldrich.com/DE/en/product/sigma/a3854).
KIM-1 (AF1817) for IHC using mouse samples was validated in a previous publication (PMID: 31767624).
Cleaved caspase-3 (9661) for IHC using mouse samples was validated in a previous publication (PMID: 30672316).
Gr1 antibody (553127) for IHC using mouse samples was validated in a previous publication (PMID: 31718093).
Anti-His antibody (clone 3D5) for capture of his-tagged proteins was validated in a previous publication (PMID: 8994661)
FSP1 antibody (clone AIFM2 1A1 rat IgG2a, and clone AFM2 14D7 IgG2b, developed in-house) has been validated for WB in this study in Extended Data Fig 9a (clone 1A1) and Extended Data Fig 10b (clone 14D7).

# Eukaryotic cell lines

Policy information about cell lines

| | |
|---|---|
| Cell line source(s) | 4-OH-TAM-inducible Gpx4-/- murine immortalized fibroblasts (Pfa1) were reported previously (PMID: 18762024). HT-1080 (CCL-121), 786-O (CRL-1932), A375 (CRL-1619), B16F10 (CRL-6475), H9C2 (CRL-1446), NRK49F (CRL-1570), C2C12 (CRL-1772), HepG2 (HB-8065), Jurkat (TIB-152), L929 (CCL-1), HEK293T (CRL-3216) and P3X63-Ag8.653 (CRL-1580) cells were obtained from ATCC. Panc-1 cells were obtained from Cell Resource Center for Biomedical Research, Institute of Development, Aging and Cancer, Tohoku University (Sendai, Japan). THP-1 cells were obtained from DSMZ (Germany). HT-22 cells were purchased from Millipore (SCC129). |
| Authentication | None of the cell lines used were authenticated. |
| Mycoplasma contamination | All cell lines were tested negative for mycoplasma contamination. |
| Commonly misidentified lines (See ICLAC register) | No commonly misidentified cell lines were used. |

# Animals and other organisms

Policy information about studies involving animals; ARRIVE guidelines recommended for reporting animal research

| | |
|---|---|
| Laboratory animals | C57BL/6J male mice (8 to 10-week old) were obtained from Charles River (Sulzfeld, Germany). C57BL/6N male mice (8 to 12-week old) were obtained from Charles River (Sulzfeld, Germany) and CLEA Japan (Tokyo, Japan). Alb-CreERT2 male mice were provided by Prof. Pierre Chambon (Illkirch, France). Gpx4fl/fl mice were reported previously (PMID: 25402683). Alb-CreERT2;Gpx4fl/+ mice were generated by crossing Gpx4fl/fl mice and Alb-CreERT2 mice in the animal facility in our institute. Alb-CreERT2;Gpx4fl/fl (male and female, 8 to 10-week old) were used in the analysis. Fsp1-/- mice (i.e., B6.129-Aifm2tm1Marc/leg) were obtained from INFRAFRONTIER (https://www.infrafrontier.eu; EM:05283). Fsp1+/+, +/- and -/- mice (male and female, 8 to 16-week old) were used in the analysis. Gpx4 fl/fl and Fsp1-/- mice were on a congenic C57BL/6J background. Wister rats (RjHan:Wi, female, age 160 days) were obtained from Javier Labs (France). Mice were kept under standard conditions with water and food ad libitum and in a controlled environment (22 ± 2°C, 55 ± 5% humidity, 12 h light/dark cycle). |
| Wild animals | The study did not involve wild animals. |
| Field-collected samples | The study did not involve field-collected samples. |
| Ethics oversight | All experiments were performed in compliance with the German Animal Welfare Law and have been approved by the institutional committee on animal experimentation and the government of Upper Bavaria (approved No. ROB-55.2-2532-Vet_02-18-13 and ROB-55.2-2532.Vet_03-17-68) and the State of Bavaria (permission granted by the government of Lower Franconia, approved No. 54-2532.1-19/13), the Landesdirektion Sachsen (TVV07/2021) involving an independent ethics committee, and the Animal Committee of Tohoku University (approved No. No. 2019-BeA012, 2019-BeA014 and 2019PhA-010-01). |

Note that full information on the approval of the study protocol must also be provided in the manuscript.

# Flow Cytometry

## Plots

Confirm that:

☒ The axis labels state the marker and fluorochrome used (e.g. CD4-FITC).

☒ The axis scales are clearly visible. Include numbers along axes only for bottom left plot of group (a 'group' is an analysis of identical markers).

☒ All plots are contour plots with outliers or pseudocolor plots.

☒ A numerical value for number of cells or percentage (with statistics) is provided.

## Methodology

Sample preparation
Pfa1 cells (50,000 cells/well) were seeded on 6-well dishes one day prior to the experiment. On the next day, cells were treated with 0.3 μM RSL3 to induce ferroptosis. Three hours later, cells were incubated with 1.5 μM of BODIPY 581/591 C11 (ThermoFisher) for 30 min at 37°C. Subsequently, cells were trypsinized, resuspended in 300 μL of Hanks' balanced salt solution (HBSS, Gibco), strained through a 40 μm cell strainer (Falcon tube with cell strainer CAP), and then analyzed using a flow cytometer (CytoFLEX, Beckman Coulter) with a 488-nm laser paired with a 530/30 nm bandpass filter.

Instrument
CytoFLEX (Beckman Coulter)

Software
CytExpert v2.4 was used for data collection. FlowJo v10 was used for data analysis.

Cell population abundance
At least 8,000 cells were analyzed for each sample.

Gating strategy
Cell populations were separated from cellular debris using FSC and SSC.

☒ Tick this box to confirm that a figure exemplifying the gating strategy is provided in the Supplementary Information.

