## [Peer Review File · Nature]

Manuscript Title: A non-canonical vitamin K cycle is a potent ferroptosis suppressor

Editorial Notes: Redactions – unpublished data

Reviewer Comments & Author Rebuttals

Reviewer Reports on the Initial Version:

Referees' comments:

Referee #1 (Remarks to the Author):

In their manuscript “A non-canonical vitamin K cycle is a potent ferroptosis suppressor”, Mishima et al. report that vitamin K is a bona fide inhibitor of ferroptosis through the activity of the ferroptosis suppressor protein-1 (FSP1) oxidoreductase. The authors comprehensively demonstrate that each of the 3 vitamin K vitamers are potent inhibitors of lipid peroxidation and suppress ferroptosis in diverse cell and animal models. Further, they elegantly show that the anti-ferroptotic activity of these compounds is associated with the radical trapping antioxidant nature of their hydroquinone forms that are derived in a FSP1-dependent manner. Finally, the authors illustrate that FSP1 is the previously unidentified NAD(P)H-dependent oxidoreductase of the canonical vitamin K cycle that mitigates warfarin poisoning. Once every few years I have the pleasure to review a manuscript that is so excellent I have nothing further to add. This is that manuscript, and is science done at its very finest. In totality, this is a very impressive manuscript with broad and significant impact to both the ferroptosis, ROS and metabolism communities, and I recommend publication in its current form. I have only very minor comments for the authors below.

Minor Comments:

- Throughout the manuscript, exogenous Vitamin K derivatives are used to interrogate their ferroptosis suppressive function. While the authors make no claims regarding the use of Vitamin K vs. CoQ by FSP1 in vivo, I wonder if they have any data regarding the preference of FSP1 for these two cofactors endogenously in various tissues. Related to this question, do the authors know the serum/tissue concentrations of MK4 upon administration in the various models, and how the levels relate to physiological concentrations of Vitamin K?
- Regarding figure 3b: is NADPH an equally efficient reductant for the FSP1-mediated reduction of vitamin K or is NADH superior for this activity?

Referee #2 (Remarks to the Author):

In their manuscript, Mishima et al identify derivatives of Vitamin K (VK) as potent suppressors of ferroptosis initiated by GPX4 inhibition or ischemia reperfusion injury in both cell and animal

models. This effect was almost completely reversed by deletion of FSP1, and the authors propose that FSP1 is responsible for the reduction of VK derivatives. This hypothesis is supported by in vitro data showing that FSP1 can protect from lipid peroxidation in a VK and NADH dependent manner. Remarkably, the authors find that protective effect of VK on warfarin poisoning is FSP1 dependent, further validating this hypothesis. Overall the experiments performed are rigorous and well-controlled, and the findings are of substantial importance both with respect to ferroptosis, vitamin K biology, and our general understanding of physiology. I only have the following minor points:

1. In Figure 3G and 3H, the authors show that FSP1 KO cells are resistant to suppression of ferroptosis by VK derivatives. The authors should confirm that other anti-ferroptotic agents like Vitamin E are capable of rescuing such cells in their model system to further establish that the phenotype is specific to VK.

The authors might also consider the following:

2. The authors should comment more directly on the relative rescue of VK derivatives compared to Vitamin E in the liver GPX4 KO and IRI models.
3. In Figure 3C, it is difficult to distinguish between some of the colors used, especially between 4 and 16 μ M conditions.
4. In Figure 4F, it would be helpful to know if the authors continued the experiment past 10 days, or if the rescue was really 100%.

Referee #3 (Remarks to the Author):

in this manuscript the authors define vitamin K, a group of fat soluble naphthoquinones as strong anti-ferroptotic agents. Indeed, they show that a group of analogs are more potent than vitamin E, making them the most potent of naturally occurring radical trapping agents. Of note, FSP1 (also known as AIFM2 (apoptosis inducing factor mitochondrial protein 2)), appears to recycle vitamin K derivatives to their antioxidant forms. These findings define a novel role for naturally occurring vitamin K and suggest that FSP is an important aspect of vitamin K function in reversing anticoagulation by warfarin. These findings are novel and interesting, and begin to frame understanding of relevant anti-ferroptotic agents in vivo. However, the data in aggregate is quite correlative and does not seem to employ state of the art chemical biology or mutational analysis to establish a causal relationship between vitamin K, fsp1 and ferroptosis. Moreover, other mechanisms that could be protective are not explored. Finally, the robustness of the protection is not supported by physiological or behavioral data. My specific comments about criticisms are below:

- 1.) iron dependent lipid peroxidation may be important for ferroptosis, but other mechanisms by which iron may participate in ferroptosis have been advanced and are not considered. This is a minor point.

- 2.) A detailed structure activity relationship is not developed for any of the vitamin K derivatives, so chemical evidence that the RTA moieties in vitamin K are essential to their death inhibitory role is not provided.

- 3.) The phenoxazines have been shown to be as potent as vitamin K, does the naturally occurring nature of vitamin K represent a conceptual advance for the field.
- 4.) Vitamin E has been disappointing as a therapeutic in a number of fields, How will vitamin K move beyond these prior failures.
- 5.) Roland stocker has shown that vitamin E can be pro-oxidant, is this true for vitamin K. Does the increased potency increase the likelihood that vitamin K derivatives will be toxic in other organs as well as protective. this is not well explored.
- 6.) Mutational analysis of FSP-1 to show that its ubiquinone reducing function is necessary. Reinstatement of the wt but not the mutant should rescue cells doubly deficient in GPX4 and FSP1.
- 7.) Can the quinone reductase that is induced by NRF-2 also reduce vitamin K?
- 8.) Since menadione is used to generate reactive oxygen species, is it possible its effects are hormetic rather than as an RTA. This needs to be investigated.
- 9.) The eplipidomics analysis again is highly correlative. Which of these lipids if any are responsible for death.
- 0.) SAR with distinct analogs of MK4 with and without FSP-1 substrate ability would enhance the interpretation of the in vivo results.
- 1.) No evidence that agents substitute for GPX4 deficiency rather than working downstream of it. Data in the literature suggest that lipid peroxidation may reflect penultimate demise but earlier RLS events that trigger death signaling may not be dependent on lipid peroxidation but be sensitive to GPX4.
- 2.) Not clear from in vivo experiments that the conversion of efficiency of MK4 epoxide is not happening in FSP deficient cells simply because they are sick.
- 3.) Need to exclude a enzymatic mechanism for protection with higher doses of vitamin K derivatives in absence of FSP1.

Overall, the conceptual advance seems limited and the data suggest correlation rather than causation.

Referee #4 (Remarks to the Author):

Mishima and colleagues describe a role for reduced Vitamin K in the control of ferroptosis and describe FSP1 as the NA(P)H-ubiquinone reductase that is required for the actions of Vitamin K. These are novel observations and are described in a series of technically well performed experiments that include both in vitro and in vivo studies. The senior author has previously

described FSP1 as a negative regulator of ferroptosis and therefore this paper provides a potential mechanism for its anti-ferroptotic activities. But a number of questions are raised upon careful reading of the paper that temper enthusiasm for publication:

Major points:

1. The survival graph in Figure 2a suggests only a short-term protective effect of MK4 on mice lacking Gpx4, this being selected as “the most efficacious derivative” of Vitamin K. Clearly then the Vitamin K-regulated protective mechanism uncovered by the authors is not as efficacious as use of high dose Vitamin E supplementation which is used to maintain Gpx4^{-/-} mice. This then draws into question the physiological importance of Vitamin K and its derivatives in the control of ferroptosis. A few additional points that are related to this experiment. What is the cause of mortality in the MK4 treated mice and what is the condition of the livers of these mice at later time points than day 7? TUNEL (Fig 2b) is used to show ferroptosis, this should be in conjunction with 4HNE co-staining rather than determining the latter by Western, but where is the proof that ferroptosis is the mechanism of cell death that is responsible for mortality in this model? It is also intriguing that the survival plots for vehicle and MK4 treated mice look very similar apart from the latter being shifted to the right, why is this? Similar in the IRI experiments (Fig 2c and d), where is the evidence that the MK4 is protecting the tissues through prevention of ferroptosis? Have protective effects of MK4 on other mechanisms of cell death such as apoptosis and necroptosis etc been formally ruled out? Similar for the LPS + dGal experiment (extended data 3c), this model is more associated with classic TNF-dependent Fas/FasL hepatotoxicity rather than ferroptosis. Regarding the kidney IRI experiments (Fig 2d), the effects of MK4 are at best modest and especially in the case of the tissue damage score, this surely brings into question the physiological importance of Vitamin K derivatives in protection from IRI induced and ferroptosis-regulated damage in this particular experimental setting. It is essential that the authors formally prove that ferroptosis is a major contributor to cell death in the various in vivo models and for them to convincingly demonstrate that MK4 (and I would suggest at least one other Vitamin K derivative) is predominantly protecting the organs by preventing ferroptosis.

2. Is FSP1 the only enzyme that mediates NAD(P)H reduction of Vitamin K and is it essential for this chemical process in vivo? The warfarin experiments are useful towards this question, but there is a lack of analysis of Vitamin K derivatives in FSP-1 knockout cells or organs. Additionally, if FSP1 brings about its anti-ferroptotic effects via reduction of Vitamin K then it should be possible to show that supplementing FSP1 deficient mice with MK4 would prevent ferroptosis and tissue damage in the IRI injury models.

Minor points:

- 1) Abstract/introduction: at the first glance, it seems that is the vitamin K that prevents ferroptosis; probably rephrasing a couple of sentences could be better.
- 2) Better contextualise the importance of the discovery of Vitamin K derivatives in a bigger picture, what are the clinical benefits (i.e. organ damage/failure, inflammation, fibrosis, cancer etc).
- 3) Figure 1b, the bar on the left should be divided in several segments to show that there is no release of LDH once the VK were added. Figure 1B: there is a release in LDH after adding Menadione.

Explain.

- 4) Page 4: missed reference on the ferroptosis induced by RSL3 and other synthetic molecules.
- 5) Figure 1d: Missing references on the several ferroptosis inducers mentioned
- 6) Figure 1 extended: It is not mention in the text or in the legends that lip1 is a ferroptosis protector and which are the ferroptosis inducers for readers that are not in the field
- 7) Figure 2, better to show 4HNE in IHC instead of WB as it is more informative.
- 8) Extended data Figure 1h: Can you extend on that? At 10 uM 100% of cells are dead, is only because there is ROS release?....Can you provide a reference for that?
- 9) Extended data Figure 2A: Quantify Liperfluo signal
- 10) Extended data 3A: Validation of GPX4 Fl/Fl is good by WB, but it is total liver. The data would benefit if supported by an IHC to show a specific deletion in hepatocytes or a western blot of isolated hepatocytes.
- 11) Fig 4e – the authors must realise that comparing just 3 mice in FSP+/+ against 7 mice in -/+ and -/- is not robust.
- 12) Can you extend on Ext. Figure 6,. The results are important as they support what is showed in Figure 3
- 13) Line 178 to 181: Which other mechanism could contribute to the reduction of VK, if it is the not the only one?
- 14) Line 210 show the genotyping? It is a novel mouse....
- 15) The liver FSP1-/- looks very blurry. Is it possible to change to something more defined.

Author Rebuttals to Initial Comments:

Nature manuscript 2021-10-15779

We thank all Reviewers and the Editor for the critical assessment of our manuscript and the highly appreciative comments made by the reviewers. Please find below our response to each comment on a point-by-point basis.

Referee #1

In their manuscript "A non-canonical vitamin K cycle is a potent ferroptosis suppressor", Mishima et al. report that vitamin K is a bona fide inhibitor of ferroptosis through the activity of the ferroptosis suppressor protein-1 (FSP1) oxidoreductase. The authors comprehensively demonstrate that each of the 3 vitamin K vitamers are potent inhibitors of lipid peroxidation and suppress ferroptosis in diverse cell and animal models. Further, they elegantly show that the anti-ferroptotic activity of these compounds is associated with the radical trapping antioxidant nature of their hydroquinone forms that are derived in a FSP1-dependent manner. Finally, the authors illustrate that FSP1 is the previously unidentified NAD(P)H-dependent oxidoreductase of the canonical vitamin K cycle that mitigates warfarin poisoning. Once every few years I have the pleasure to review a manuscript that is so excellent I have nothing further to add. This is that manuscript, and is science done at its very finest. In totality, this is a very impressive manuscript with broad and significant impact to both the ferroptosis, ROS and metabolism communities, and I recommend publication in its current form. I have only very minor comments for the authors below.

We are incredibly thankful for the highly appreciative comments and are very much pleased that the Reviewer shares the enthusiasm for our work.

Minor Comments:

1 Throughout the manuscript, exogenous Vitamin K derivatives are used to interrogate their ferroptosis suppressive function. While the authors make no claims regarding the use of Vitamin K vs. CoQ by FSP1 *in vivo*, I wonder if they have any data regarding the preference of FSP1 for these two cofactors endogenously in various tissues.

This is a very good point which has also puzzled us while performing the study and which remains unclear at present. This is further complicated by the tissue/cell-type specific distribution of extra-mitochondrial CoQ10, which we previously showed to be an excellent substrate of FSP1 (Doll et al., Nature 2019). Nonetheless, addressing the substrate preference of FSP1 *in vivo* is technically very challenging (if not impossible) because the reduced form of vitamin K is very unstable *in vivo* and as such might be extremely difficult to assess. However,

the CoQ concentrations in the body are much higher than those of vitamin K (plasma level of CoQ10 is 0.88 $\mu\text{g}/\text{mL}$, as compared to vitamin Ks that are in the range of 200 - 600 pg/mL [PMID: 27128225 and 8256661]), indicating that CoQ10 is the main substrate of FSP1 in terms of availability. We have added a half-sentence to address this point, as follows:

Page 10, line 253 "Considering the evolution of life, when environmental oxygen concentrations increased after the great oxidation event in primordial Earth, it appears that MK was substituted by ubiquinone as an electron carrier due to its higher redox potential and increased abundance as compared to VKs".

Related to this question, do the authors know the serum/tissue concentrations of MK4 upon administration in the various models, and how the levels relate to physiological concentrations of Vitamin K?

We fully agree that we are using supra-nutritional concentrations of vitamin K. However, since there is no known *in vivo* toxicity of vitamin K (see also comment to Referee 3, point #5), we first wanted to make the point that vitamin K in conjunction with FSP1 can be a powerful ferroptosis suppressing mechanism, beyond its canonical function in blood clotting, bone metabolism etc.

We have since performed pharmacokinetic analyses to determine the levels of MK4 in plasma and tissue of *Gpx4 fl/fl* mice after bolus i.p. injection of MK4 (200 mg/kg ; 0, 1, 3, 6 and 24 h after injection; Extended Data Fig. 3b). As expected, the tissue concentrations of MK4 were exceedingly high as compared to physiological levels of vitamin K, which are known to range from 0.7 pmol/mL (plasma) to 520 pmol/g tissue (pancreas) in mice (PMID: 18083713). Regarding the potential physiological role of an anti-ferroptotic action of vitamin K, please see the comment to Referee 4, major point 1.

2 Regarding figure 3b: is NADPH an equally efficient reductant for the FSP1-mediated reduction of vitamin K or is NADH superior for this activity?

We additionally examined the efficacy of NADPH on the reducing activity of FSP1 towards MK4 using the FENIX assay (Extended Data Fig. 5e), showing that the efficacy of NADPH as a cofactor

was comparable to NADH. This result is consistent with the reducing activity of FSP1 towards CoQ10 (Doll et al., Nature 2019).

Referee #2

In their manuscript, Mishima et al identify derivatives of Vitamin K (VK) as potent suppressors of ferroptosis initiated by GPX4 inhibition or ischemia reperfusion injury in both cell and animal models. This effect was almost completely reversed by deletion of FSP1, and the authors propose that FSP1 is responsible for the reduction of VK derivatives. This hypothesis is supported by in vitro data showing that FSP1 can protect from lipid peroxidation in a VK and NADH dependent manner. Remarkably, the authors find that protective effect of VK on warfarin poisoning is FSP1 dependent, further validating this hypothesis. Overall, the experiments performed are rigorous and well-controlled, and the findings are of substantial importance both with respect to ferroptosis, vitamin K biology, and our general understanding of physiology. I only have the following minor points:

We greatly appreciate the very kind words and highly supportive comments made by the Reviewer, as well as for recognizing the importance and relevance of our study.

1. In Figure 3G and 3H, the authors show that FSP1 KO cells are resistant to suppression of ferroptosis by VK derivatives. The authors should confirm that other anti-ferroptotic agents like Vitamin E are capable of rescuing such cells in their model system to further establish that the phenotype is specific to VK.

We extended the results of Fig. 3f and Fig. 3g to confirm that the phenotype observed in FSP1 KO cells is specific to VK (see Extended Data Fig. 7c and 7g, and below).

In 786O-FSP1 KO cells treated with RSL3, the anti-ferroptotic effects of Trolox (a water-soluble analogue of vitamin E) and deferiprone (an iron chelator) were the same as in the control WT cells (Extended Data Fig. 7c). In contrast, α -TOH (α -tocopherol, a vitamin E) showed a diminished anti-ferroptotic effect against RSL3 in FSP1 KO cells (Extended Data Fig. 7c), akin to

PK and MK4 (Fig. 3f), because FSP1 can also regenerate α -TOH following its reaction with lipid peroxyl radicals (see the scheme in Extended Data Fig. 6e and Ref. Doll et al., Nature 2019, PMID: 31634899).

The anti-ferroptotic effect of Trolox, whose antioxidant mechanism is independent of FSP1, was also comparable between B16F10 *Gpx4* KO cells and that of *Gpx4/Fsp1* double KO cells (Extended Data Fig. 7g), like menadione (which also can be reduced independently on FSP1). Taken together, the diminished anti-ferroptotic effect in FSP1 KO cells is specific to PK, MK4 and α -TOH. Regarding these additional results, we described in the main text, as follows:

Page 8, line 190: "PK and MK4 showed a diminished anti-ferroptotic effect against RSL3 in FSP1 KO cells similar to α -TOH, which can also be regenerated by FSP1⁴ (Fig. 3f, Extended Data Fig. 7a-c)."

Page 8, line 208: "Genetic deletion and pharmacological inhibition of FSP1 did not significantly influence the anti-ferroptotic effect of menadione, akin to other FSP1-independent ferroptosis inhibitors (Extended Data Fig. 7c, g)."

The authors might also consider the following:

2. The authors should comment more directly on the relative rescue of VK derivatives compared to Vitamin E in the liver GPX4 KO and IRI models.

We agree that there are tissue-specific protective mechanisms at play, and that the reasons for differences in rescue might be manifold:

For instance, we know that the diet used in the genetic liver-specific GPX4 null model and the ones used in IRI studies have different contents of vitamin E. While better controlled in the genetic liver-specific *Gpx4* KO setting (approx. 7 mg/kg in vitamin E deficient diet in chow pellet), in the IRI models the animals received a diet containing "normal" concentrations of vitamin E (130 mg/kg in pellet) according to guidelines provided by Charles River in the 1960s (which are also too high when considering the daily intake of vitamin E in humans - for a calculation and extended discussion see Friedmann Angeli et al., Trends in Pharmacol Sci 2017; PMID: 28363764).

Another very important point to consider is the following, which is directly related to differences in vitamin E administration in food vs i.p. injections of MK4: Successive or repeated dosing of a certain drug will result in increasing concentrations of the drug in the body until a plateau is reached (i.e., the "steady state level"). At steady state, the amount of drug administered on each dosing occasion is matched by an equivalent amount of drug leaving the body between each dose. In addition, at steady state, plasma concentrations of the drug will increase and decrease

according to the dosing interval as long as the drug is administered at the same dose level and same time period between doses. Hereby, vitamin E is administered with the mouse diet. As mice eat food *ad libitum* continuously throughout the day, less fluctuations in vitamin E plasma concentrations are anticipated compared to MK4, which is administered daily by i.p. injection. In addition, for most drugs, it takes roughly five half-lives to reach an approximate steady state level. In the case of Vit E, mice already have reached a steady state level at the time of tamoxifen induction and the start of the experiment. For MK4, however, it presumably takes some time to reach its steady state after knockout induction. Hence, we hypothesize that for ferroptosis inhibition and prevention of organ failure in this model, a continuous exposure of drugs with minimal fluctuations in plasma concentration in the steady state phase is likely preferred. Experimental validation using a comprehensive pharmacokinetic analysis in combination with *in vivo* efficacy studies using both vitamin E and vitamin K at the same route of administration and dosing intervals would be necessary to verify this hypothesis. Moreover, tissue- and cell type-specific expression levels of FSP1 in the different organs will also have a major impact on the outcome on the dependence on vitamin E and vitamin K. [REDACTED]

All these different possibilities should not, however, distract from the overall message that the herein described “non-canonical vitamin K cycle” is a novel cycle that should be extensively studied in the future following our proof-of-concept. Of note, the protective effect of vitamin K against IRI was found in the condition with sufficient vitamin E. We thus additionally addressed these points in the main text, which now reads as follows

Page 5, line 131: “Although the anti-ferroptotic *in vivo* efficacy of MK4 was not directly compared to that of vitamin E, these results nonetheless demonstrate that the tissue-protective effects of MK4 were evident in animal models either being deprived of, or having sufficient, vitamin E in the diet.

Page 5, line 124: “To address whether MK4 might also be tissue-protective in a ferroptosis model of ischemia-reperfusion injury (IRI), C57BL/6 mice maintained on a normal diet were treated with MK4 prior to liver or kidney IRI.”

3. In Figure 3C, it is difficult to distinguish between some of the colors used, especially between 4 and 16 uM conditions.

We have changed the colors in Fig. 3c, which are hopefully now easier to follow.

4. In Figure 4F, it would be helpful to know if the authors continued the experiment past 10 days, or if the rescue was really 100%.

Multiple previous papers have shown that high doses of vitamin K can completely rescue coumarin intoxication including warfarin poisoning over a long period of time (it is the actual antidote of warfarin poisoning). That is the reason why we discontinued the study; and yes, the rescue is really 100%.

Referee #3

in this manuscript the authors define vitamin K, a group of fat soluble naphthoquinones as strong anti-ferroptotic agents. Indeed, they show that a group of analogs are more potent than vitamin E, making them the most potent of naturally occurring radical trapping agents. Of note, FSP1 (also known as AIFM2 (apoptosis inducing factor mitochondrial protein 2), appears to recycle vitamin K derivatives to their antioxidant forms. These findings define a novel role for naturally occurring vitamin K and suggest that FSP is an important aspect of vitamin K function in reversing anticoagulation by warfarin. These findings are novel and interesting, and begin to frame understanding of relevant anti-ferroptotic agents in vivo. However, the data in aggregate is quite correlative and does not seem to employ state of the art chemical biology or mutational analysis to establish a causal relationship between vitamin K, fsp1 and ferroptosis. Moreover, other mechanisms that could be protective are not explored. Finally, the robustness of the protection is not supported by physiological or behavioral data. My specific comments about criticisms are below:

We thank the Reviewer for the comments and for appreciating the novelty and the interest of our study.

1. iron dependent lipid peroxidation may be important for ferroptosis, but other mechanisms by which iron may participate in ferroptosis have been advanced and are not considered. This is a minor point.

There is overwhelming consensus in the literature that iron-dependent lipid peroxidation is the hallmark of ferroptosis. Of course, there are other mechanisms where iron could play a role (e.g., iron-sulfur cluster biogenesis, see Alvarez et al. "NFS1 undergoes positive selection in lung tumours and protects cells from ferroptosis" Nature 2017), but ultimately free (i.e., unchaperoned) iron in conjunction with peroxides incites lipid peroxidation, and if not counteracted by GPX4, FSP1 etc., will ultimately lead to cell death by ferroptosis. To provide further mechanistic clarity, we confirmed that all forms of vitamin K did not show any iron chelating properties (Extended Data Fig. 2c), a point which we now include as follows:

Page 4, line 103: "These studies showed that all three forms of VK efficiently averted lipid

peroxidation in a mechanism independent of any iron chelating effect (Extended Data Fig. 2c)."

2. A detailed structure activity relationship is not developed for any of the vitamin K derivatives, so chemical evidence that the RTA moieties in vitamin K are essential to their death inhibitory role is not provided.

Thank you for your suggestion. Since the key role of the phenolic O-H group of α -TOH in both its anti-ferroptosis and RTA activity has been illustrated by showing that its methylated analog is completely devoid of both (see Shah et al. *ACS Central Science* 2018, 4, 387-396), we have done the same here with menadione. Similarly, following methylation of the naphthoquinone moiety, dimethylmenadione was devoid of both RTA activity in the FENIX assay and anti-ferroptotic activity in Pfa1 cells (Extended Data Fig. 1h and Extended Data Fig 5g; also shown below). Thus, the redox-active naphthoquinone headgroup is the functional group in vitamin K conferring RTA activity. In addition, we provide a scheme depicting the radical trapping reaction of VK in Extended Data Fig. 6f:

3.) The phenoxazines have been shown to be as potent as vitamin K, does the naturally occurring nature of vitamin K represent a conceptual advance for the field?

As the reviewer mentioned, phenoxazines, spiroquinoxalines and phenothiazines (and many others) have been shown to be anti-ferroptotic compounds by acting as RTAs. However, of note, vitamin K itself is not an RTA, and must first be reduced to its hydroquinone form by a specific enzyme, i.e. FSP1 (as unequivocally shown here for the first time), to display RTA activity. We now provide strong evidence for this unidentified mechanism of this naturally occurring vitamin.

This is clearly a conceptual advance, since scientists working on the function of vitamin K in the

canonical vitamin K cycle would not have otherwise linked vitamin K with ferroptosis. Beyond this, our findings will have immediate impact on clinical settings as warfarin is one of most widely prescribed drugs and as clinical trials are ongoing that put forward the concept that vitamin K supplementation might be a new strategy to ameliorate degenerative diseases (see comments below & Reviewer 4, minor point #2).

4.) Vitamin E has been disappointing as a therapeutic in a number of fields, How will vitamin K move beyond these prior failures?

The clinical efficacy of vitamin E has been proven in the treatment for nonalcoholic steatohepatitis (NEJM 2010, PMID 20427778), and provided encouraging results even in Alzheimer's disease studies (although statistical proof-of-concept has not been reached likely also due to poor stratification of enrolled patients and lack of knowledge on the general vitamin E status). As our data now shows that vitamin K at supra-nutritional concentrations is tissue-protective in certain pathological conditions (i.e., IRI) associated with ferroptosis even in animals that were kept at standard vitamin E concentrations contained in normal animal chow. Since vitamin K is extremely well-tolerated and does not cause any reported side effects *in vivo*, yet is physiologically only present at low concentrations in the human body, we believe that it will be worth to put forward the concept that vitamin K might confer robust tissue-protective activity in certain disease settings.

5.) Roland stocker has shown that vitamin E can be pro-oxidant, is this true for vitamin K. Does the increased potency increase the likelihood that vitamin K derivatives will be toxic in other organs as well as protective. this is not well explored.

Indeed, Roland Stocker and Keith Ingold showed that vitamin E can be a pro-oxidant, but only in isolated lipoprotein particles where vitamin E-derived radicals can have sufficiently long lifetimes to enable it. As far as we are aware, vitamin E does not possess this 'tocopherol-mediated peroxidation' (TMP) activity in phospholipid bilayers. Moreover, CoQ10 has been shown to subvert this activity due to its ability to donate a H-atom to the vitamin E-derived radical (see PMID: 8419943). The latter reactivity is also behind the FSP1-dependent synergy between the VKs and α -TOH which we report here. Menadione can exhibit pro-oxidant behavior due to the rapid autoxidation of its reduced form in the cytosol, but the more lipophilic VK derivatives (PK and MK4) are less prone to this redox cycling due to their partitioning to the lipid bilayer. Clinically, there are no reports of PK or MK4 overdose toxicity (PMID: 19874942).

However, we additionally extended the result of menadione toxicity (Extended Data Fig. 1g). The cellular toxicity by high concentration of menadione was not rescued by Lip1 (ferroptosis inhibitor), Z-VAD-FMK (pan-caspase inhibitor) or Nec1s (necroptosis inhibitor), indicating that menadione induces a ROS-dependent cell death without caspase-dependent apoptosis,

necroptosis or ferroptosis (Extended Data Fig. 1g). In contrast, N-acetylcysteine (NAC), which is often employed as cysteine source and glutathione (GSH) precursor, prevented the menadione-induced cell death, as described previously (PMID: 7728902). Mechanistically, GSH conjugates with menadione, and the GSH-conjugated menadione can be quickly transported across the cell membrane and pumped out via efflux transporters such as MRP1 (PMID: 22679290).

6.) Mutational analysis of FSP-1 to show that its ubiquinone reducing function is necessary. Reinstatement of the wt but not the mutant should rescue cells doubly deficient in GPX4 and FSP1.

In principle, this is a valid point. Unfortunately, the three-dimensional structure of FSP1 is not known, although several labs are relentlessly working on it. Therefore, this experiment is currently technically very challenging; only the NAD(P)H binding domain can be predicted due to amino acid homology and some mutations in this domain are deleterious. [REDACTED]

[REDACTED]

7.) Can the quinone reductase that is induced by Nrf-2 also reduce vitamin K?

As already described in the text on page 8 (line 206) of the submitted manuscript (Indeed, menadione can be reduced non-enzymatically by glutathione (Extended Data Fig. 8a), and enzymatically by NQO1 and/or thioredoxin reductase in addition to FSP1^{28,29}), the Nrf2 target NQO1 can reduce menadione, but cannot reduce PK/MK4. A previous study already showed that NQO1 (alias DT-diaphorase) cannot reduce PK/MK4 (PMID: 2113031).

8.) Since menadione is used to generate reactive oxygen species, is it possible its effects are hormetic rather than as an RTA. This needs to be investigated.

The reviewer likely refers to the idea that low levels of ROS would be beneficial (“hormetic effects”), while high concentrations are cytotoxic. To exclude the possibility of a hormetic effect resulting from the stress response to low levels of ROS, we examined the viability of Pfa1 cells \pm TAM \pm H₂O₂, as shown below. The low concentration of H₂O₂ did not prevent ferroptosis, in contrast to menadione. Since we think that this does not add any new insights, we would rather refrain from including this data in the revised version.

6.) The eplipidomics analysis again is highly correlative. Which of these lipids if any are responsible for death.

This data illustrates how vitamin K prevents unrestrained (per)oxidation of a series of phospholipids (the hallmark of ferroptosis) that we have previously described along with Kagan's laboratory to be preferentially oxidized in cells and tissues undergoing ferroptosis and which could be blunted by the *in vivo* efficacious ferroptosis inhibitor liproxstatin-1 (see Friedmann Angeli et al., Nat Cell Biol 2014). These include the w-6 PUFAs arachidonic (C20:4) and adrenic (C22:4) acids esterified preferably in phosphatidyl ethanolamine (PE) (Kagan et al., Nat Chem Biol 2017; Doll et al., Nat Chem Biol 2017), which was repeatedly confirmed by many groups thereafter. The (per)oxidation of the species shown in Figure 1f can be inhibited by co-treatment with vitamin K. In addition, eplipidomics analysis of liver-specific *Gpx4* knockout mice revealed widespread oxidation of PUFAs contained in PE and phosphatidyl choline (PC), which could be fully blunted by MK4 treatment of liver-specific *Gpx4* knockout mice (see new Fig. 2c and Extended Data Fig. 3d).

In case the Reviewer is interested in the cellular events downstream of lipid peroxidation, the Reviewer is referred to a recent preprint article by the Friedmann Angeli group, where they provide insights on how lipid peroxidation products cause membrane disintegration and ferroptotic cell death (DOI: 10.21203/rs.3.rs-943221/v1).

0.) SAR with distinct analogs of MK4 with and without FSP-1 substrate ability would enhance the interpretation of the *in vivo* results.

As mentioned in comment #6, the 3D-structure of FSP1 has not been published, thus precluding extensive structure-activity relationships (SAR) with MK4 and its analogs. Moreover, the purpose of the current work is to elucidate the mechanism of these natural products, not the design/optimization of new chemical entities.

We already tested MK4 in FSP1 WT and KO animals, the reciprocal experiment. We deem this experiment more meaningful, as MK4 analogs might not only interfere with FSP1 substrate

recognition, but also with the canonical vitamin K cycle.

1.) No evidence that agents substitute for GPX4 deficiency rather than working downstream of it. Data in the literature suggest that lipid peroxidation may reflect penultimate demise but earlier RLS events that trigger death signaling may not be dependent on lipid peroxidation but be sensitive to GPX4.

Indeed, we have no evidence that the VK derivatives do anything other than act as RTAs upon FSP1-mediated reduction. Since there is no reasonable chemical mechanism by which the VK derivatives would possess peroxidase-like activity, thereby enabling them to substitute directly for GPX4 deficiency, it isn't clear where we would even begin to consider other 'possible' mechanisms. On the other hand, there is ample evidence supporting a role for RTAs, including those formed when FSP1 reduces the VK derivatives, in the suppression of ferroptosis. Thus, it seems more appropriate to invoke Occam's *pluralitas non est ponenda sine necessitate*.

We have interpreted RLS to mean "Rate-Limiting Step", which would mean that it and 'events' are both redundant and ambiguous since there is generally only one RLS in a mechanism. Moreover, as far as we are aware, the kinetics of ferroptotic cell death have not been systematically studied, precluding the characterization of a RLS for this process.

2.) Not clear from *in vivo* experiments that the conversion efficiency of MK4 epoxide is not happening in FSP deficient cells simply because they are sick.

As suggested (a similar comment has been raised by Referee 4, point #2), to evaluate the conversion efficiency of MK4 into MK4-epoxide *in vivo*, we measured the levels of MK4 and MK4-epoxide in the liver and plasma of *Fsp1*^{+/-} and *Fsp1*^{-/-} mice treated with warfarin + MK4 (see Fig. 4e and Extended Fig. 10e; also shown below). The ratio of MK4-epoxide/MK4 in *Fsp1*^{-/-} mice was much less than that in *Fsp1*^{+/-} mice, indicating the poor conversion of administered MK4 to MK4-epoxide (via MK4-hydroquinone). This result replicates the *in vitro* data (Fig 4d), and strongly supports the conclusion that FSP1 is the responsible warfarin-resistant VK reductase in the canonical VK cycle.

13.) Need to exclude an enzymatic mechanism for protection with higher doses of vitamin K derivatives in absence of FSP1.

As already described in the submitted manuscript (page 8, lines 204), other mechanisms (though less efficient) may contribute to the reduction of vitamin K quinone independent of FSP1, yet we do not know whether there is any enzymatic mechanism beyond the ones discussed.

To identify a putative alternative enzymatic mechanism of vitamin K quinone reduction beyond FSP1, one would need to setup a genome-wide CRISPR/Cas9 drop-out genetic screen in multiple (cancer) cell lines engineered to lack FSP1 expression, whereby high vitamin K concentrations would then fail to rescue (see Figure 3f). However, we firmly believe that this comprehensive set of experiments would go by far beyond the scope of the present study, whereby we really like to make the point and report on the "non-canonical vitamin K cycle" as a novel ferroptosis suppressing mechanism maintained by FSP1 (besides identifying the long-sought after enzyme of the canonical vitamin K cycle!). As such, these studies may be subject of follow-up work.

Referee #4

Mishima and colleagues describe a role for reduced Vitamin K in the control of ferroptosis and describe FSP1 as the NA(P)H-ubiquinone reductase that is required for the actions of Vitamin K. These are novel observations and are described in a series of technically well performed experiments that include both *in vitro* and *in vivo* studies. The senior author has previously described FSP1 as a negative regulator of ferroptosis and therefore this paper provides a potential mechanism for its anti-ferroptotic activities. But a number of questions are raised upon careful reading of the paper that temper enthusiasm for publication:

We thank the reviewer for his comments and his appreciation of the novelty and robustness of our work with several single and compound mutant cell lines and a number of *in vivo* (genetic) models associated with ferroptosis.

Major points:

1. The survival graph in Figure 2a suggests only a short-term protective effect of MK4 on mice lacking Gpx4, this being selected as "the most efficacious derivative" of Vitamin K. Clearly then the Vitamin K-regulated protective mechanism uncovered by the authors is not as efficacious as use of high dose Vitamin E supplementation which is used to maintain Gpx4^{-/-} mice. This then draws into question the physiological importance of Vitamin K and its derivatives in the control of ferroptosis.

A similar point was raised by Referee 1, comment #2. We concur that there are tissue-specific protective mechanisms at play the reasons therefore are likely manifold.

For instance, we know that the diet used in the genetic liver-specific GPX4 null model and the ones used in IRI studies have different contents of vitamin E. While better controlled in the genetic liver-specific *Gpx4* KO setting (approx. 7 mg/kg in vitamin E deficient diet in chow pellet), in the IRI models the animals received a diet containing "normal" concentrations of vitamin E (130 mg/kg in pellet) according to guidelines provided by Charles River in the 1960s (which are also too high when considering the daily intake of vitamin E in humans - for a calculation and extended discussion see Friedmann Angeli et al., Trends in Pharmacol Sci 2017; PMID: 28363764).

Another very important point to consider is the following which is directly related to differences in vitamin E administration in food vs *i.p.* injections of MK4: Successive or repeated dosing of a certain drug will result in increasing concentrations of the drug in the body until a plateau is reached (i.e., "steady state level"). At steady state, the amount of drug administered on each dosing occasion is matched by an equivalent amount of drug leaving the body between each dose. In addition, at steady state, plasma concentrations of the drug will increase and decrease

according to the dosing interval as long as the drug is administered at the same dose level and same time period between doses. Hereby, vitamin E is administered with the mouse diet. As mice eat food *ad libitum* continuously throughout the day, less fluctuations in vitamin E plasma concentrations are anticipated compared to MK4, which is administered daily by i.p. injection. In addition, for most drugs, it takes roughly five half-lives to reach an approximate steady state level. In the case of Vit E, mice already have reached a steady state level at the time of tamoxifen induction and the start of the experiment. For MK4, however, it presumably takes some time to reach its steady state after knockout induction. Hence, we hypothesize, that for ferroptosis inhibition and prevention of organ failure in this model, a continuous exposure of drugs with minimal fluctuations in plasma concentration in the steady state phase is necessary. Experimental validation using a comprehensive pharmacokinetic analysis in combination with *in vivo* efficacy studies using both vitamin E and vitamin K at the same route of administration and dosing intervals would be necessary to verify this hypothesis.

Moreover, tissue- and cell type-specific expression levels of FSP1 in the different organs will also have a major impact on the outcome on the dependence on vitamin E and vitamin K. [REDACTED]

All these different possibilities will, however, not distract from the overall message that the herein described “non-canonical vitamin K cycle” is a novel cycle that should be extensively studied in the future after providing proof-of-concept. However, of note, the protective effect of vitamin K against IRI was found in the condition with sufficient vitamin E. We thus additionally addressed these points in the main text, which now reads as follows:

Page 5, line 131: “Although the anti-ferroptotic *in vivo* efficacy of MK4 was not directly compared to that of vitamin E, these results nonetheless demonstrate that the tissue-protective effects of MK4 were evident in animal models either being deprived of or having sufficient vitamin E in the diet.”

[REDACTED]

[REDACTED]

A few additional points that are related to this experiment. What is the cause of mortality in the MK4 treated mice and what is the condition of the livers of these mice at later time points than day 7?

The cause of death in MK4-treated liver-specific GPX4 null mice is also liver failure similarly to vehicle-treated mice. Indeed, ALT and AST levels in the serum collected at humane endpoint of MK4-treated animals (n=2) was extensively high (AST 58,180 and 52,895 IU/L; ALT 42,825 and 53,070 IU/L). The general condition of the liver-specific *Gpx4* knockout mice deteriorates rapidly during the last 24 h before reaching humane endpoint or death, indicating progression of massive liver damage in this period. MK4 treatment extended the survival period of the KO mice by slowing down the progression of liver damage via preventing lipid peroxidation.

TUNEL (Fig 2b) is used to show ferroptosis, this should be in conjunction with 4HNE co-staining rather than determining the latter by Western, but where is the proof that ferroptosis is the mechanism of cell death that is responsible for mortality in this model?

We agree, TUNEL is a general marker for cell death as currently there is no direct robust marker to detect ferroptosis in tissues unlike for apoptosis and necroptosis, to some extent. We added corresponding IHC staining for 4HNE as suggested as well as staining against cleaved caspase-3 to exclude the contribution of apoptosis in the liver-specific *Gpx4* null mice (Fig. 2b and extended fig 3a). The liver-specific *Gpx4* knockout mice showed strong positivity for 4HNE as well as TUNEL staining, but were negative in staining against cleaved caspase-3, indicating no contribution of apoptosis to liver failure in these mice. In addition, to add further evidence of lipid peroxidation in liver tissue of *Gpx4* null mice, we performed epilipidomics of the affected liver tissue (new Fig. 2c and Extended Data Fig. 3d). The results clearly showed the progression of lipid peroxidation in the liver-specific *Gpx4* knockout mice and its suppression by MK4 treatment.

Furthermore, we refer the Reviewer to a recent paper by the Vanden Berghe lab that independently from our genetic *Gpx4* KO model showed that both the ferroptosis inhibitor UAMC-3203, an improved ferrostatin-1 analogue, and liproxstatin-1 prevent liver damage in the same model (<https://www.researchsquare.com/article/rs-310675/v1>; *accepted in Nature Communications*), akin to hepatic IRI as reported by our group earlier (Friedmann Angeli et al Nat Cell Biol 2014).

It is also intriguing that the survival plots for vehicle and MK4 treated mice look very similar apart from the latter being shifted to the right, why is this?

Yes, we agree that the shape of the survival curve coincidentally looks similar when comparing the two cohorts. However, this represents the data that we obtained during the experiment. The raw data used for the survival period of the individual animals is given below:

Fig 2a	Survival days
vehicle	9
vehicle	10
vehicle	12
vehicle	9
vehicle	6
MK4	17
MK4	18
MK4	17
MK4	21
MK4	6

p=0.0277
Log-rank, vehicle vs MK4

Similar in the IRI experiments (Fig 2c and d), where is the evidence that the MK4 is protecting the tissues through prevention of ferroptosis? Have protective effects of MK4 on other mechanisms of cell death such as apoptosis and necroptosis etc been formally ruled out? Similar for the LPS + dGal experiment (extended data 3c), this model is more associated with classic TNF-dependent Fas/FasL hepatotoxicity rather than ferroptosis. Regarding the kidney IRI experiments (Fig 2d), the effects of MK4 are at best modest and especially in the case of the tissue damage score, this surely brings into question the physiological importance of Vitamin K derivatives in protection from IRI induced and ferroptosis-regulated damage in this particular experimental setting. It is essential that the authors formally prove that ferroptosis is a major contributor to cell death in the various in vivo models and for them to convincingly demonstrate that MK4 (and I would suggest at least one other Vitamin K derivative) is predominantly protecting the organs by preventing ferroptosis.

As mentioned above, a direct and well-established biomarker for ferroptosis is still lacking. We are currently working on the identification of a (lipid-based) robust ferroptosis-specific biomarker (that can be used in conjunction with TUNEL staining – see comment above), but this will likely take another couple of years.

To show that vitamin K indeed specifically protects against ferroptosis, we additionally performed several assays using a number of *bona fide* cell death models for major cell death pathways, such as apoptosis, pyroptosis and necroptosis (new Extended Data Fig. 1f). All three forms of vitamin K failed to protect against apoptosis induced by i) the apoptosis-inducing compounds staurosporine and 17AAG, ii) FasL or iii) TNF α + BV6 (a Smac mimetic). Similarly, vitamin K did not prevent necroptosis induced by TNF α + BV6 + zVAD-FMK in L929 cells. Although menadione prevented pyroptosis induced by nigericin in THP-1 cells, as previously reported (PMID: 32917982), PK and MK4 failed to protect against pyroptosis.

Since only ferroptosis (but not other forms of cell death) is marked by excessive lipid peroxidation, which has been nicely shown by a recent paper from the Vandenabeele/Vanden Berghe/Kagan labs (Ref: Wiernicki B *et al.* Cell Death Dis. 2020 PMID: 33110056), we additionally performed epilipidomics analysis of liver-specific *Gpx4* null mice to unequivocally show that the

cell death modality at play is indeed ferroptosis (new Fig. 2c and Extended Data Fig. 3d). The results showed excessive lipid peroxidation in liver tissue of liver-specific *Gpx4* null mice that could be blunted by MK4 treatment. This set of data convincingly shows that MK4 protected organ damage by preventing lipid peroxidation and ferroptosis.

We also additionally showed that MK4 treatment decreased tissue lipid peroxidation, indicated by 4HNE staining, in the liver IRI model (new Extended Data Fig. 3e). In addition, several previous reports showed that IRI causes ferroptosis in diverse tissues including kidney, liver, lung, intestine and brain which we deem sufficient to firmly make this point (PMID: 25402683, 30737476 and 31056284).

Regarding the LPS + dGalN liver injury model, we previously showed that ferroptosis inhibitor prevented LPS + dGalN-induced liver damage (PMID: 31767624), indicating that ferroptosis is involved in the pathophysiology. Nonetheless, as the reviewer mentioned, other cell death modalities, such as apoptosis, may also contribute to the pathophysiology. Thus, the model would not be suitable to show the anti-ferroptotic effect of MK4. Therefore, we removed the data of LPS-dGal model in the revised version.

The less protective effect in kidney IRI as compared to liver IRI can be manifold and might also depend on the tissue context itself and tissue-specific distribution of vitamin K. The experimental protocol was different between each IRI study. We used 90 min-ischemia and twice MK4 injection for pretreatment in the liver IRI; in contrast, 36 min-ischemia and single MK4 injection were used for the kidney IRI (Extended Fig 3c). In addition, we measured tissue MK4 level in the kidney and liver (new Extended Data Fig. 3b; also shown below). The peak MK4 level after i.p. administration was higher in the liver than the kidney. The difference of MK4 distribution might contribute to the therapeutic efficacy in each organ.

Furthermore, we performed additional *in vivo* studies using phylloquinone (vitamin K1) in both IRI models (we did not use menadione because high-doses of menadione are known to be hepatocytotoxic like in cellular systems; see PMID: 3824412). Phylloquinone treatment showed partial (yet not significant) protection against kidney IRI, and no protective effects in the liver IRI model. As already presumed from the anti-ferroptotic efficacy of phylloquinone compared to MK4 (see Fig. 1a and in text on page 4), these findings might be somehow expected, which may also depend on a different tissue distribution/pharmacokinetics compared to MK4.

2. Is FSP1 the only enzyme that mediates NAD(P)H reduction of Vitamin K and is it essential for this chemical process *in vivo*? The warfarin experiments are useful towards this question, but there is a lack of analysis of Vitamin K derivatives in FSP-1 knockout cells or organs. Additionally, if FSP1 brings about its anti-ferroptotic effects via reduction of Vitamin K then it should be possible to show that supplementing FSP1 deficient mice with MK4 would prevent ferroptosis and tissue damage in the IRI injury models.

Thank you for the great suggestion. As suggested (a similar comment has been raised by Referee 3, point #12), to evaluate the conversion efficiency of MK4 into MK4-epoxide *in vivo*, we measured the levels of MK4 and MK4-epoxide in the liver and plasma of *Fsp1*^{+/+} and *Fsp1*^{-/-} mice treated with warfarin + MK4 (see Fig. 4e and Extended Fig. 10e. also shown below). The ratio of MK4-epoxide/MK4 in *Fsp1*^{-/-} mice was much less than that in *Fsp1*^{+/+} control mice, indicating the poor conversion of administered MK4 to MK4-epoxide (via MK4-hydroquinone).

This result replicates the *in vitro* data (Fig 4d), and strongly supports the conclusion that FSP1 is the responsible warfarin-resistant VK reductase in the canonical VK cycle.

Fsp1^{-/-} mice are only marginally more sensitive to IRI in kidney which is likely due to the presence of GPX4 (Tonnus et al., Nat Commun 2021, PMID: 34285231). Due to this rather marginal difference, it will be hard to detect some differences in *Fsp1*^{-/-} mice treated with vitamin K as also other non-enzymatic (see Fig. 3f and comments on page 8 lines 206) systems are at play that can partially reduce vitamin K.

Minor points:

- 1) Abstract/introduction: at the first glance, it seems that is the vitamin K that prevents ferroptosis; probably rephrasing a couple of sentences could be better.

We corrected this and now say:

Page 2, line 46: "Here, we show that the fully reduced forms of vitamin K, a group of fat-soluble naphthoquinones including menaquinone and plant-derived phylloquinone3, confers a yet-unrecognized strong anti-ferroptotic function, ...".

- 2) Better contextualise the importance of the discovery of Vitamin K derivatives in a bigger picture, what are the clinical benefits (i.e. organ damage/failure, inflammation, fibrosis, cancer etc).

Thank you for your advice. We added the sentence mentioning the contextualize the importance of the discovery of vitamin K derivatives in therapy for ferroptosis-related diseases, such as organ injury and degenerative diseases, which now reads as follows:

Page 10, line 259: "The present findings put forward the concept that vitamin K treatment might be a new powerful strategy to ameliorate ferroptosis-related diseases, such as organ injury and degenerative diseases."

- 3) Figure 1b, the bar on the left should be divided in several segments to show that there is

no release of LDH once the VK were added. Figure 1B: there is a release in LDH after adding Menadione. Explain.

We modified the graph in Fig. 1b (LDH release results). The lines indicating PK, MK4 and Lip1 are shown with a slight shift to avoid overlapping (see below). Extended exposure of cells to menadione (3 days) shows some toxicity, thereby increasing LDH release (this relates to comment #8 describing menadione toxicity).

0) Page 4: missed reference on the ferroptosis induced by RSL3 and other synthetic molecules.

1) Figure 1d: Missing references on the several ferroptosis inducers mentioned

Regarding comments 4) and 5), we now include reference #17 providing an overview about RSL3 and other synthetic molecules. Due to the limitation of the number of references, we just provide one article, which now reads as follows:

Page 3, line 86: "PK, MK4 and menadione also efficiently rescued cells from ferroptosis triggered by well-established ferroptosis inducers¹⁷ including RSL3 in fibrosarcoma HT-1080 cells."

2) Figure 1 extended: It is not mention in the text or in the legends that lip1 is a ferroptosis protector and which are the ferroptosis inducers for readers that are not in the field

Thank you for your suggestion. We now mention in the figure legend that Lip1 is a well-established ferroptosis inhibitor, which now reads as follows:

In Fig. 1a legend "a. Cell viability in 4-hydroxytamoxifen (4-OH-TAM)-induced *Gpx4*^{KO} mouse embryonic fibroblasts (MEFs) (Pfa1 cells) treated with phylloquinone (PK), menaquinone-4 (MK4), menadione (Menad), α -tocopherol (α -TOH) and liproxstatin-1 (Lip1, a well-established ferroptosis inhibitor)"

3) Figure 2, better to show 4HNE in IHC instead of WB as it is more informative. Thank you

for your suggestion. We now show IHC against 4HNE in Fig. 2b and Extended Data

Fig. 3a.

8) Extended data Figure 1h: Can you extend on that? At 10 μM 100% of cells are dead, is only because there is ROS release?....Can you provide a reference for that?

We additionally performed several experiments to extend the data of menadione-induced cell death (Extended Data Fig. 1g and 1h, which you also find below). The cellular toxicity induced by high concentrations of menadione was not rescued by Lip1 (a ferroptosis inhibitor), Z-VAD-FMK (a pan-caspase inhibitor) or Nec1s (a necroptosis inhibitor), indicating that menadione at high concentrations induces a ROS-dependent form of cell death without involving caspase-dependent apoptosis, necroptosis or ferroptosis (Extended Data Fig. 1g). In contrast, N-acetylcysteine (NAC), which is a frequently used cysteine source and glutathione (GSH) precursor, can prevent menadione-induced cell death (Extended Data Fig. 1g). Mechanistically, GSH conjugates with menadione in cells, and the GSH-conjugated menadione can be quickly transported across the cell membrane and pumped out via efflux transporters such as MRP1 (PMID: 22679290). Through this mechanism, NAC prevents menadione toxicity. These findings have been already reported in previous reports (PMID: 20937380). In addition, we now also show that dimethylmenadione (a redox-inactive derivative of menadione) is less toxic, but lacks anti-ferroptotic activity (Extended Data Fig. 1h), supporting the notion that toxicity of menadione and its anti-ferroptotic activity is dependent on its redox activity.

9) Extended data Figure 2A: Quantify Liperfluo signal

We provided the quantitative result of Liperfluo signal intensity in Extended Data Fig. 2a, which now looks as follows:

4) Extended data 3A: Validation of GPX4 Fl/Fl is good by WB, but it is total liver. The data would benefit if supported by an IHC to show a specific deletion in hepatocytes or a western blot of isolated hepatocytes.

Thank you for the good suggestion. We additionally performed IHC against GPX4 of liver tissue in the *Alb-CreERT²⁺;Gpx4^{fl/fl}* mice in Extended Data Fig. 3a. The IHC image clearly shows the disappearance of GPX4 in hepatocytes with remaining positive signals in other types of liver cells, such as stellate cells, Kupffer cells and endothelium:

5) Fig 4e – the authors must realise that comparing just 3 mice in FSP^{+/+} against 7 mice in -/+ and -/- is not robust.

As the Reviewer indicated, the comparison between 3 mice of *Fsp1^{+/+}* and 6 mice in *-/+* and *-/-* is statistically not robust. While we did not show any statistical test on the *Fsp1^{+/+}* versus other groups, our purpose was to illustrate how *Fsp1^{+/-}* mice behaved in comparison to *Fsp1^{-/-}* mice. For the purpose of this comparison (*Fsp1^{+/-}* vs *Fsp1^{-/-}*), 6 mice per groups is statistically sufficient because 6 animals per group were required in the sample size calculation.

Consequently, the P value calculated was as significant (P=0.002, Fig. 4e). In light of tight regulations regarding the use of transgenic animals for *in vivo* experiments, these calculated sample sizes are discussed not only with a biostatistician, but also present a strict requirement when receiving permission from the respective authorities. Nonetheless, as the *Fsp1^{#/#}* and *Fsp1^{#/-}* mice are phenotypically indistinguishable, in accordance to the Reviewer's suggestion, we removed the data of *Fsp1^{#/#}* mice in Fig. 4e:

6) Can you extend on Ext. Figure 6,. The results are important as they support what is showed in Figure 3

We added the data of Lipiradical Green assay in extended fig 6d (as below). We measured the lipid radical scavenging activities of other ferroptosis inhibitors (Fer1 and Lip1) with or without a reducing agent, DTT. These RTAs scavenged lipid radicals without DTT, and DTT did not affect to the scavenging activity of these ferroptosis inhibitors unlike vitamin K. This finding supports the notion that these ferroptosis inhibitors themselves are RTAs; in contrast, vitamin K shows RTA activity only in the reduced form.

7) Line 178 to 181: Which other mechanism could contribute to the reduction of VK, if it is the not the only one?

This point directly relates to Referee 3, comment #13. As described in the manuscript (lines

182), other mechanisms (though less efficient) may contribute to the reduction of vitamin K quinone independent of FSP1, yet we do not know whether there is an enzymatic mechanism beyond the ones discussed.

However, to identify a yet-unrecognized enzymatic mechanism of vitamin K quinone reduction beyond FSP1 (and likely less efficient), one would need to setup a genome-wide CRISPR/Cas9 drop-out screen in multiple (cancer) cell lines engineered to lack FSP1 expression whereby high vitamin K concentrations would then fail to rescue (see Figure 3f). Yet, we firmly believe that this comprehensive set of experiments would go by far beyond the scope of the present study, which is focused on the “non-canonical vitamin K cycle” as a novel ferroptosis suppressing mechanism maintained by FSP1 (besides identifying the long-sought after enzyme of the canonical vitamin K cycle). As such, these studies may be pursued in follow-up studies.

0) Line 210 show the genotyping? It is a novel mouse...

The mouse line was first reported in Tonnus et al. Nat Commun. 2021 Jul 20;12(1):4402. doi: 10.1038/s41467-021-24712-6., although not described in detail. Therefore, we now provide information on the gene targeting strategy applied and a genotyping PCR image in Extended Data Fig 10:

1) The liver FSP1^{-/-} looks very blurry. Is it possible to change to something more defined.

We agree and replaced the image of H&E-stained liver of *Fsp1*^{-/-} mice in Extended Data Fig. 10c:

Reviewer Reports on the First Revision:

Referees' comments:

Referee #1 (Remarks to the Author):

The authors have done an excellent job addressing my minor points, and responding to the critiques raised by the other reviewers. The addition of new data to support their claims has made a very strong manuscript even stronger. This study will open up new avenues of research into FSP1-mediated Vitamin K reduction and its roles in physiology, including Vitamin K as a ferroptosis protective molecule. These highly novel and significant findings should be published immediately.

Referee #2 (Remarks to the Author):

The authors have more than adequately address my comments and present an outstanding study.

Referee #3 (Remarks to the Author):

The authors clearly are thoughtful and understand ferroptosis well, but a number of the aspects of the revised manuscript still leave me unconvinced in the model presented. Accordingly, I do not think that the manuscript represents the kind of conceptual advance suitable for Nature.

- 1.) Vitamin E has been a failure clinically largely because no one has figured out to determine where it is deficient and what is the right dose to ameliorate pathology in humans. I fear Vitamin K metabolites as RTAs may suffer the same fate.
- 2.) The inability to mutate domains of FSP-1 is a major hurdle that needs to be solved before this paper can be seen as more than correlative. It is not really adequate to simply say the full structure has not been completely elucidated. Without this data most of the data again seems highly correlative.
- 3.) The inability to definitively exclude enzymatic mechanisms of higher doses of vitamin K remains a weakness of the study.
- 4.) The authors did not address the issue of whether protection by vitamin K in vivo is associated with physiological or behavioral improvements.
- 5.) The authors did not address the issue of whether vitamin K works up stream of GPX4 deficiency or downstream of it.

Overall, the revision seemed more perfunctory than a rigorous attempt to meet the highest standards of publication. Overall an interesting and potentially important phenomenon that needs

more experimentation to begin to convince one of the relationship between vitamin K derivatives and ferroptosis protection.

Referee #4 (Remarks to the Author):

The authors have not yet provided robust physiological evidence for an impressive protective effect of Vitamin K or for its most potent derivative MK4 in an organ injury model, as such the relevance of the data to a medical need is questionable. In the response to reviewers the authors are not correct in stating that Vitamin E is proven in the treatment of NASH. Clinical trials have shown at best modest biochemical and histological improvements and the duration of any beneficial effects are only short term, so it is unlikely that Vitamin K would be any more efficacious.

The question of whether ferroptosis or another mechanism of cell death is implicated was addressed, however the lack of an unequivocal marker for ferroptosis remains a concern since the authors cannot conclusively prove this mechanism and its relationship with Vitamin K.

Author Rebuttals to First Revision

Author's Response to Editorial Comments and Suggestions:

We greatly thank the Editor for the continued discussions, and the careful assessment of the revised version, as well as for the opportunity to submit a revised version of our manuscript. As suggested by the Editor, we have now avoided the direct comparison between Vit E and Vit K and have reformulated sentences in the manuscript toning down any statements overestimating the translational value of our finding, thereby purely focusing on the novelty of our basic biological findings. Additionally – as suggested by the Editor – we incorporated an additional set of control stainings, excluding a major contribution of apoptosis to the overall damage in our rodent IRI models. We would also like to draw your attention to the series of immunohistochemical stains from our liver IRI experiments, as suggested by you. As you will notice this recapitulates the findings provided in the previous version of the

manuscript, and if you deem them to be sufficiently complimentary, we would be happy to include them side-by-side to the final version of the manuscript.

Figure for response letter.

Liver histology and related immunohistochemical stainings (H&E and TUNEL staining and immunohistochemistry for 4HNE and cleaved-caspase 3) are shown. Livers of mice treated with IR diffusely present TUNEL-positive cell death along with lipid peroxidation as indicated by 4HNE staining. Cleaved caspase-3 positive cells were not detected in the IRI tissue, while cleaved-caspase 3 positive signals were abundantly detected in control tissue (i.e., murine spleen).

Response to Referees:

Referee #1: The authors have done an excellent job addressing my minor points, and responding to the critiques raised by the other reviewers. The addition of new data to support their claims has made a very strong manuscript even stronger. This study will open up new avenues of research into FSP1-mediated Vitamin K reduction and its roles in physiology, including Vitamin K as a ferroptosis protective molecule. These highly novel and significant findings should be published immediately.

Referee #2: The authors have more than adequately address my comments and present an outstanding study.

Authors' Response to referees #1 and #2: We are grateful for Referees #1 and #2 for their thorough assessments of our revised manuscript and their highly appreciative words and comments. We completely agree with both referees considering the novelty and significance of our findings further highlighting our expectation that they will be instrumental in our quest to better understand ferroptosis and vitamin K biology – a link this paper makes for the first time.

Referee #3: The authors clearly are thoughtful and understand ferroptosis well, but a number of the aspects of the revised manuscript still leave me unconvinced in the model presented. Accordingly, I do not think that the manuscript represents the kind of conceptual advance suitable for Nature.

1. Vitamin E has been a failure clinically largely because no one has figured out to determine where it is deficient and what is the right dose to ameliorate pathology in humans. I fear Vitamin K metabolites as RTAs may suffer the same fate.
2. The inability to mutate domains of FSP-1 is a major hurdle that needs to be solved before this paper can be seen as more than correlative. It is not really adequate to simply say the full structure has not been completely elucidated. Without this data most of the data again seems highly correlative.
3. The inability to definitively exclude enzymatic mechanisms of higher doses of vitamin K remains a weakness of the study.
4. The authors did not address the issue of whether protection by vitamin K in vivo is associated with physiological or behavioral improvements.
5. The authors did not address the issue of whether vitamin K works up stream of GPX4 deficiency or downstream of it.

Overall, the revision seemed more perfunctory than a rigorous attempt to meet the highest standards of publication. Overall an interesting and potentially important phenomenon that needs more experimentation to begin to convince one of the relationship between vitamin K derivatives and ferroptosis protection.

Authors' Response to Referee #3: We thank Referee #3 for the comments on our revised manuscript. However, we must respectfully disagree on a few of the points above.

Reply to (1); The translation of the preclinical findings regarding vitamin E supplementation into clinical settings has indeed been arduous. While we are well aware that there are some structural similarities between both naturally occurring lipophilic vitamins, their chemical structures – particularly the RTA-active headgroups – are clearly distinct. That being said, we never intended to provide a new treatment paradigm and/or explain previous clinical negative results for vitamin E or any other vitamin beyond vitamin K. Our work simply does not deal with vitamin E biology or its potential clinical applications. Instead, we deliberately chose to include this class of naturally occurring antioxidants as well-accepted ferroptosis inhibitors and therefore appropriate controls in ferroptosis research. With our studies using vitamin K (MK4) we solely intended to provide unambiguous proof-of-concept for the novel non-canonical vitamin K cycle – driven by FSP1 and being operational *in vivo* – by acting as an anti-ferroptotic mechanism in diverse (rodent) experimental disease models. In fact, we can faithfully demonstrate in several independent and widely used (rodent) models of ferroptosis research that vitamin K – beyond its canonical function in blood coagulation and bone metabolism – conferred a robust anti-ferroptotic effect (as summarized in the graphical abstract in Fig 4h).

Reply to (2): As mentioned in our previous point-by-point response, we and other groups have been working on the 3D structure of FSP1 for several years. And although we can successfully express functionally active FSP1 enzyme to homogeneity (see Doll *et al.* Nature 2019 and current work), the crystals do not diffract sufficiently well for us – or others – to obtain a structure. As such, we have teamed up with dedicated NMR/Cryo-EM experts and have started related experiments recently. What we unequivocally do show in our present study is that in cell-free systems MK4 is an excellent substrate of FSP1. At the same time, we have further increased our efforts regarding the underlying issue and have teamed up with experts in nanobody (VHHs) research. We were able to generate highly specific nanobodies, which will help us to generate better crystals as so-called crystallization chaperones.

Evidently, inclusion of results obtained from these independent and large-scale projects clearly extends the scope of our current study as also discussed with the Editor.

Reply to (3): We demonstrate in various *in vitro* (cell-free and cell-based) models that vitamin K acts as a *bona fide* RTA requiring FSP1-mediated reduction with electrons coming from either NADH or NADPH as demonstrated in the revised manuscript (such as in Fig 3b). We are unaware of any enzyme-mediated process, whereby vitamin K could exert an analogous effect, but if the reviewer has something plausible in mind, we would be very happy to consider it.

Reply to (4): We have shown that MK4 supplementation prolongs the survival of hepatocyte-specific *Gpx4* KO mice and mitigates tissue damage inflicted by ischemia-reperfusion injury (IRI) in the liver and kidney (in Fig. 2). We did not apply MK4 in other models, such as stroke or neurodegeneration, where – at least to our understanding – behavioral assessment would be an appropriate and meaningful readout. Thus, it remains unclear to us how behavioral experiments would bear relevance to renal or hepatic IRI. Nonetheless, we are convinced that our findings will stimulate researchers in the specialized field(s) such as stroke, traumatic brain injury and neurodegeneration to pursue our concept in experimental models thereof.

Reply to (5): We apologize that we have not made this point explicitly clear. The variety of complementary genetic cellular systems used throughout the manuscript as well as the *in vivo* evidence clearly suggested that vitamin K phenotypically acted “downstream” of GPX4. If vitamin K were to act “upstream” of GPX4 (for instance by upregulating its expression or enhancing its activity), then one would have never observed such a strong rescue in cells and mice lacking GPX4 expression (as in Fig. 1 and 2). If the reviewer feels further elaboration would improve the message of our manuscript, we would be happy to do so.

Authors' Response to Referee #4:

The authors have not yet provided robust physiological evidence for an impressive protective effect of Vitamin K or for its most potent derivative MK4 in an organ injury model, as such the relevance of the data to a medical need is questionable. In the response to reviewers the authors are not correct in stating that Vitamin E is proven in the treatment of NASH. Clinical trials have shown at best modest biochemical and histological improvements and the duration of any beneficial effects are only short term, so it is unlikely that Vitamin K would be any more efficacious.

The question of whether ferroptosis or another mechanism of cell death is implicated was addressed, however the lack of an unequivocal marker for ferroptosis remains a concern since the authors cannot conclusively prove this mechanism and its relationship with Vitamin K.

Authors' Response to referee #4: We thank Referee #4 for their thoughtful comments on the robustness of our proposed concept in a clinical setting. After comparing the main concerns raised during the first round of reviews and the ones provided to us after revision, we are concerned that there may be a misunderstanding of the implications of our *in vivo* experiments.

Our preclinical *in vivo* studies were intended solely to provide proof-of-concept for our cell-free and cell-based findings in well-accepted and ferroptosis-relevant rodent models of disease. We are well aware of the potential shortcomings of these rodent models, yet we considered them very important since ferroptosis research may be prone to artifacts given the varying redox conditions between cultured cells and a complex biological organism. Thus, on the basis of the *in vivo* data, it seems fair to suggest that our conclusions should extend beyond our *in vitro* findings. Whether or not the results obtained in preclinical models may guide future translational efforts led by clinicians in clinical settings for human diseases is certainly beyond the scope of our study, where we solely aimed to provide biological proof-of-concept for both the non-canonical vitamin K cycle, as well as the insofar unknown enzyme essential for driving the full canonical vitamin K cycle (an unsolved mystery since roughly half a century) (as shown in the graphical abstract in Fig. 4h). Nonetheless, beyond our considerations and implications in ferroptosis, we think that the translational potential may exceed the models we leveraged in our current study, but which should (and certainly will be) addressed by researchers in related fields.

Concerning a ferroptosis-specific marker, we fully understand the issue raised by referee #4: "... the lack of an unequivocal marker for ferroptosis remains a concern...". The current gold-standard in ferroptosis research is the characterization of oxidized phospholipids by high-resolution mass spectrometry (as also discussed with the Editor), which we have carried out and which provides definitive proof that MK4 suppresses signatures observed after genetic loss of *Gpx4* (i.e., ferroptosis) in both cells and tissues (as shown in Fig. 1f and 2c, Extended fig 2d, 3d). We additionally included data on 4-HNE staining of tissues (in Fig. 2b), which is regarded as an "indirect marker" for lipid peroxidation. As such, we think that the statement on the lack of a robust biomarker in ferroptosis research is not entirely correct. Of course, it would be preferable to have one or more protein-based biomarker on a number of levels, but its identification remains a huge challenge facing the entire ferroptosis research community.

Reviewer Reports on the Second Revision:

Referees' comments:

Referee #3 (Remarks to the Author):

Thank you to authors for taking my queries seriously, but I remain unconvinced on a number of issues.

1.) This is a discussion point, but the authors must provide a perspective on how the distinct head group of MSK4 will allow them ultimately to identify the target and dose of this compound in humans. Target engagement and optimal dose remain a huge challenge in RTA therapeutics.

2.) I would like a structure activity relationship of substrates of fsp-1 tested and the potency and effectiveness of these substrates for inhibiting ferroptosis. This table could make a clear argument for fsp1 being the critical enzyme for rescuing ferroptosis.

3.) Thank you for your thoughtful response.

4.) The response was frankly a bit pedantic and insulting. I asked for physiology and behavior. As the authors know liver failure if fully protected should result in normalization of a variety physiological measures of hepatic function-this was not provided. Also, liver failure is associated with decreased activity of animals and activity monitoring is routinely done in studies outside the nervous system. This will help the authors know whether they are preventing undead cells or fully functional ones.

5.) Again, perplexing response. The experiment I am asking for does GPX4 overexpression abrogate deleterious lipid formation in the absence of fsp-1.

If these issues can be addressed, the paper seems like it could meet the standards of Nature.

Author Rebuttals to Second Revision:

Manuscript ID 2021-10-15779C

Referees' comments:

Referee #3 (Remarks to the Author):

Thank you to authors for taking my queries seriously, but I remain unconvinced on a number of issues.

1.) This is a discussion point, but the authors must provide a perspective on how the distinct head group of MSK4 will allow them ultimately to identify the target and dose of this compound in humans. Target engagement and optimal dose remain a huge challenge in RTA therapeutics.

2.) I would like a structure activity relationship of substrates of fsp-1 tested and the potency and effectiveness of these substrates for inhibiting ferroptosis. This table could make a clear argument for fsp1 being the critical enzyme for rescuing ferroptosis.

3.) Thank you for your thoughtful response.

4.) The response was frankly a bit pedantic and insulting. I asked for physiology and behavior. As the authors know liver failure if fully protected should result in normalization of a variety physiological measures of hepatic function-this was not provided. Also, liver failure is associated with decreased activity of animals and activity monitoring is routinely done in studies outside the nervous system. This will help the authors know whether they are preventing undead cells or fully functional ones.

5.) Again, perplexing response. The experiment I am asking for does GPX4 overexpression abrogate deleterious lipid formation in the absence of fsp-1.

If these issues can be addressed, the paper seems like it could meet the standards of Nature.

Reply to point #1: In the previous round of revisions, we were asked to restrain the discussion surrounding clinical applications of vitamin K as an anti-ferroptotic drug (as also repeatedly discussed with the Editor and Editorial team). Thus, any discussion of target engagement and dosage requirements does not seem appropriate. Nonetheless, we fully agree with the referee that the translation of the anti-ferroptotic function of RTAs from *in vitro* to *in vivo* settings has, to date, been met with little success. We understand that the lack of a conventional (i.e. protein-based) target has, so far, limited medicinal chemistry campaigns aimed at addressing this point. This has been exacerbated by disappointing clinical trials of vitamin E and other naturally occurring RTAs as potential therapeutics. Indeed, our paper does not deal with therapeutic aspects of RTAs. We have worked diligently to translate our *in vitro* findings to an *in vivo* context as a readout for phenotypes provoked by the genetic loss of *Gpx4*, i.e., ferroptosis, in an effort to affirm its biological relevance and significance. Experimentally, we know that *in vitro* there is a clear dose-effect relationship for vitamin K and its anti-ferroptotic function (see Fig. 1a); however, we cannot infer its bioavailability from analyzing its structure as suggested. Since vitamin K cannot be overdosed and no toxicity is clinically observed – even at exceedingly high dosages - we used a high vitamin K dosage in our experiments and phenotypic readouts merely as a proof-of-concept for our cellular and biochemical findings. Regardless, we included one sentence on page 7 in the revised manuscript saying that “*the naphthoquinone head group is sufficient to prevent lipid peroxidation and subsequent ferroptosis*”.

Reply to point #2: As per the reviewer’s suggestion, we have added a supplemental table (included in supplemental data file) summarizing the results of our structure-activity relationship of substrates of FSP1 as a direct comparison of the anti-ferroptotic activity in the different cell lines and under the different conditions considered in the manuscript. It includes both mouse and human cell lines for the sake of comprehensiveness.

In supplemental table 2

Summary of the potency of vitamin K for ferroptosis inhibition *in vitro*

Cells	Induction of ferroptosis	IC ₅₀ (μM) for inhibiting ferroptosis			
		PK	MK4	Menad	Dimethyl-menad
Pfa1	Gpx4 deletion by TAM	0.18	0.074	1.4	No inhibition
A375 GPX4 ^{KO}	Lip1 withdrawal	0.061	0.016	0.47	
786O GPX4 ^{KO}	Lip1 withdrawal	0.49	0.078	1.3	
786O wt	RSL3	0.49	0.39	1.1	
786O FSP1 ^{KO}	RSL3	6.6	5.5	3.7	
786O FSP1 ^{KO} + hFSP1 ^{OE}	RSL3	0.50	0.56	1.5	

FSP1-mediated RTA activity evaluated by FENIX (shown in Extended fig 5i)

	+32 nM FSP1 + 16 μM NADH		
	PK	MK4	Menad
$k_{inh} (M^{-1}s^{-1})$	$1.5 \pm 0.3 \times 10^3$	$5.4 \pm 0.9 \times 10^3$	$11 \pm 1 \times 10^3$
n	0.49 ± 0.09	0.51 ± 0.09	0.13 ± 0.01

*Dimethyl-menad showed no FSP1-mediated RTA activity (Ex fig 5g)

Reply to point #5: Ferroptosis is a widely accepted mode of cell death associated with unrestrained lipid peroxidation in cellular membranes. Of note, the predominant system preventing ferroptosis both *in vitro* and *in vivo* is the enzymatic activity of GPX4 (ref. Friedmann Angeli JP., Nat Cell Biol. 2014, PMID: 25402683; ref. Ingold et al., Cell 2018; PMID: 29290465). Next to GPX4, we identified FSP1 as the second mainstay preventing ferroptosis – acting independently from GPX4 (see Doll et al. Nature, 2019 and below).

It is important to note that, in contrast to GPX4, the genetic deletion of *FSP1* does not lead to ferroptotic cell death or any other overt phenotype in gene-targeted mice, as shown in Extended Fig. 10 in the present manuscript. Likewise, cells deficient for FSP1 are fully viable *in vitro* under steady-state conditions without induction of lipid peroxidation (see Doll et al., Nature 2019). The reason for this is based on the activity of endogenous GPX4, which is potent enough to prevent lipid peroxidation and ferroptosis at steady-state conditions. This

is further illustrated by an immunoblot analysis of GPX4 expression in tissues of *Fsp1*^{-/-} animals and cells (additional data provided below), showing there is no compensatory upregulation of GPX4 in the absence of FSP1.

Regardless, to accommodate the suggestions of the referee, we performed the suggested experiment, where GPX4 was overexpressed in FSP1 KO cells and the extent of lipid peroxidation was determined by Bodipy-C11, the standard lipid peroxidation probe. As expected, FSP1 KO alone did not induce any lipid peroxidation in the steady state, supporting the notion that endogenous GPX4 is sufficient to prevent lipid peroxidation. Accordingly, overexpression of GPX4 also showed no significant effect in the FSP1 KO cells in the steady state (see below).

Reviewer Reports on the Third Revision:

Referees' comments:

Referee #3 (Remarks to the Author):

The authors have done an excellent job of addressing my concerns. The paper appears rigorous and complete and extremely responsive to my concerns.